# Polyamine-mediated ferroptosis amplification acts as a targetable vulnerability in cancer

Guoshu Bi[1,3], Jiaqi Liang[1,3], Yunyi Bian[1,3], Guangyao Shan[1], Yiwei Huang[1], Tao Lu[1], Huan Zhang[1], Xing Jin[1], Zhencong Chen[1], Mengnan Zhao[1], Hong Fan[1], Qun Wang [1], Boyi Gan [2] ✉ & Cheng Zhan [1] ✉

Targeting ferroptosis, an iron-dependent form of regulated cell death triggered by the lethal overload of lipid peroxides, in cancer therapy is impeded by our limited understanding of the intersection of tumour's metabolic feature and ferroptosis vulnerability. In the present study, arginine is identified as a ferroptotic promoter using a metabolites library. This effect is mainly achieved through arginine's conversion to polyamines, which exerts their potent ferroptosis-promoting property in an $H_2O_2$-dependent manner. Notably, the expression of ornithine decarboxylase 1 (ODC1), the critical enzyme catalysing polyamine synthesis, is significantly activated by the ferroptosis signal——iron overload——through WNT/MYC signalling, as well as the subsequent elevated polyamine synthesis, thus forming a ferroptosis-iron overload-WNT/MYC-ODC1-polyamine-$H_2O_2$ positive feedback loop that amplifies ferroptosis. Meanwhile, we notice that ferroptotic cells release enhanced polyamine-containing extracellular vesicles into the microenvironment, thereby further sensitizing neighbouring cells to ferroptosis and accelerating the "spread" of ferroptosis in the tumour region. Besides, polyamine supplementation also sensitizes cancer cells or xenograft tumours to radiotherapy or chemotherapy through inducing ferroptosis. Considering that cancer cells are often characterized by elevated intracellular polyamine pools, our results indicate that polyamine metabolism exposes a targetable vulnerability to ferroptosis and represents an exciting opportunity for therapeutic strategies for cancer.

Ferroptosis refers to a form of iron-dependent cell death initiated by unrestricted lipid peroxidation in the plasma membrane or membrane organelles, which differs from other forms of regulated cell death like apoptosis due to its unique mechanistic and morphological features[1,2]. Briefly, in an iron-rich microenvironment, excessive $H_2O_2$ reacts with iron in a Fenton reaction to produce highly reactive hydroxyl radicals, which attack and peroxidize the bis-allylic moieties of polyunsaturated fatty acid (PUFA)-containing phospholipids located on the cellular membrane, thus leading to membrane rupture and ferroptotic cell death. Meanwhile, to counteract this effect, cells have evolved diverse ferroptosis defence mechanisms mainly consisting of SLC7A11/GPX4-dependent glutathione (GSH) system, FSP1-, DHODH- or GPD2-dependent ubiquinol system, and GCH1-dependent tetra-hydrobiopterin ($BH_4$) system[3,4]. The imbalance between the two

[1]Department of Thoracic Surgery, Zhongshan Hospital, Fudan University, Shanghai, China. [2]Department of Experimental Radiation Oncology, The University of Texas MD Anderson Cancer Center, Houston, TX, USA. [3]These authors contributed equally: Guoshu Bi, Jiaqi Liang, Yunyi Bian. ✉e-mail: bgan@mdanderson.org; czhan10@fudan.edu.cn

processes, such as the inactivation of the defence pathways by ferroptosis inducers (FINs), would lead to a rapid accumulation of lipid peroxides and triggers potent ferroptotic cell death[1].

Accumulating evidence reveals that ferroptosis plays a key role in tumour biology and cancer therapy. On the one hand, ferroptosis could be triggered by multiple canonical cancer therapeutic strategies, including radiotherapy (RT), chemotherapy, and immunotherapy[5–7]. On the other hand, ferroptosis engages in a set of cancer-associated signalling pathways. For example, the oncogenic forms of KRAS-mutant desensitised cells to ferroptosis through NRF2-mediated FSP1 upregulation[8], whereas p53, the tumour suppressor, promotes ferroptosis by both transcriptionally repressing SLC7A11 and releasing the lipoxygenase activity of ALOX12[9]. Therefore, these studies establish ferroptosis as a natural barrier to cancer development, and suggest that altering tumour cells' response to ferroptosis might provide new avenue for treating cancers that are refractory to conventional therapies.

Importantly, cancer cells acquire metabolic adaptions that support their enhanced rates of growth and proliferation[10]. However, these features, such as the high load of ROS and enhanced PUFA synthesis, render some cancer cells intrinsically susceptible to ferroptosis[3]. It is therefore of interest to better understand the interaction between tumoral metabolic characteristics and ferroptosis modulation, and to identify potential therapeutical targets for clinical practice. Although increasing studies have revealed some enzymes and their metabolic products/metabolites and corresponding enzymes participate in ferroptosis regulation, a systematic analysis of endogenous metabolites' pro- or anti-ferroptotic properties is still lacking.

In this study, using a metabolite compound library, we find that arginine and its downstream metabolic product polyamines significantly sensitise cancer cells to ferroptosis in an $H_2O_2$-dependent manner. Polyamine synthesis is induced by the important ferroptosis signal−iron overload through WNT/MYC signalling, thus forming a positive-feedback axis that amplifies ferroptosis. Consistently, polyamine also sensitises cancer cells to radiotherapy or chemotherapy through inducing ferroptosis. Given that cancer cells frequently possess elevated intracellular polyamine abundance, our results establish polyamine metabolism as a targetable vulnerability to ferroptosis in cancer treatment.

## Results

### Identification and validation of arginine as an endogenous ferroptosis mediator

To discover metabolites potentially involved in the process of ferroptosis, an RSL3-induced A549 cell ferroptosis model was adopted to screen a library containing 889 human endogenous metabolites (Fig. 1a). As one of the most commonly used class II FINs, RSL3 directly inhibits the catalytic activity of GPX4 by covalently binding to the selenocysteine residue of GPX4 through their electrophile chloroacetamide moiety[3]. Screening of this library identified arginine, one of the most versatile non-essential amino acids involved in multiple biological processes[11], significantly promoted RSL3-induced ferroptosis (Fig. 1b). We further validated this finding in both A549 and HT1080 cells, as well as three non-small cell lung cancer cell lines including H1299, H23 and PC9, by deleting or exogenously supplementing arginine into the culture medium (Fig. 1c, d and Supplementary Fig. S1a). Since lipid peroxidation is a major hallmark of ferroptosis, we stained cells with BODIPY-C11 581/591 and found that exogenous arginine significantly amplified lipid peroxidation induced by RSL3 (Fig. 1e). Upregulation of prostaglandin-endoperoxide synthase 2 (PTGS2, a biomarker of ferroptosis mRNA was also detected in arginine-treated cells (Supplementary Fig. 1b)[12]. Moreover, arginine-caused RSL3 and IKE sensitisation could be fully rescued by the ferroptosis inhibitor ferrostatin-1 (Fer-1, lipid peroxidation scavenger)

and deferoxamine (DFO, iron chelator), but not by apoptosis inhibitor z-VAD(OMe)-FMK or necroptosis inhibitor (necrosulfonamide), confirming the specific promoting role of arginine in ferroptosis (Supplementary Fig. 1c). These data indicate that arginine specifically enhances cells' vulnerability to ferroptosis.

### The pro-ferroptotic role of arginine is mediated by its downstream metabolite ornithine and polyamine

We next focused on exploring the mechanism accounting for arginine's ferroptosis mediating effect. As one of the most versatile amino acids in animal cells, arginine serves as not only a precursor for protein synthesis but also a key metabolite in the ornithine cycle and citrulline cycle. The former pathway is catalysed by arginase, in which arginine is directly converted to ornithine/urea. The type I isoenzyme (ARG1) is exclusively expressed in the liver, while the type II (ARG2) is widely expressed in extrahepatic tissues[11,13]. For the latter pathway, nitric oxide synthase (NOS) converts arginine to citrulline and NO. Data mined from the Cancer Cell Line Encyclopedia (CCLE) database revealed that ARG2, rather than ARG1 or NOS, is dominantly expressed in A549 and HT1080 used in this study, suggesting ornithine and urea tend to be the major metabolic product of arginine[14]. To verify this issue, we labelled the arginine in DMEM medium with $^{15}N$ isotope ($^{15}N_4$-arginine) and traced its metabolic fate in A549 with liquid chromatography-mass spectrometry (LC-MS) (Fig. 1f). Consistently, ornithine (M + 2) and urea (M + 2), but not citrulline, emerged as the dominant metabolic product of $^{15}N_4$-arginine (M + 4), suggesting arginine is mainly metabolised to ornithine (Fig. 1g).

To understand whether arginine impairs cells' resistance to ferroptosis by converting to ornithine, we performed cytotoxicity assay and observed that exogenously supplemented ornithine generated a similar ferroptosis-sensitising effect to arginine, whereas the by-product urea failed to do so. In addition, citrulline also slightly promoted ferroptosis (Fig. 1h and Supplementary Fig. S1d, e). This phenomenon might be due to its two-step transition to arginine as a part of the ornithine cycle (Fig. 1f)[15]. Furthermore, CRISPR/Cas9-mediated ARG2 knockout (KO) not only led to a detectable accumulation of arginine and decreased formation of ornithine but also inhibited FINs-induced cell death, as well as abolished the ferroptosis-promoting effect conferred by arginine supplementation (Fig. 1i–k and Supplementary Fig. S1f). These findings reveal that ornithine, the metabolic product of arginine, mediates arginine's ferroptosis effect.

Generally, ornithine is widely thought to be metabolised to proline, citrulline, and polyamines by ornithine aminotransferase (OAT), ornithine transcarbamylase (OTC), and ornithine decarboxylase 1 (ODC1) respectively[16]. The absence of $^{15}N$-labelled citrulline in the metabolic tracing analysis mentioned above was in agreement with the absence of OTC expression in the cell lines we used[14]. We next tested whether these metabolites explained the pro-ferroptotic role of arginine and ornithine. Notably, proline did not affect cells' response to FINs, whereas polyamines displayed a stronger ferroptosis-sensitising effect than arginine even in a low concentration without markedly impacting cell's viability when solely administered (spermine > spermidine > putrescine), which was consistent with our initial metabolites screening results (Figs. 1b and 2a and Supplementary Fig. S2a, b). The broad-spectrum sensitising activity of polyamines was also corroborated in FIN56, another FIN-depleting both GPX4 and CoQ, but not in FINO₂, the class IV FIN that directly oxidises iron (Supplementary Fig. 2c)[17]. Interestingly, single use of polyamines decreased the level of lipid peroxidation, but when combined with RSL3, on the contrary, polyamines significantly amplified RSL3-induced lipid peroxidation level (Fig. 2b and Supplementary Fig. S2d). Besides, pre-treatment with polyamine transport inhibitor AMXT-1501 enhanced cells' resistance to RSL3/IKE and blocked polyamines' ferroptosis-sensitising effect (Fig. 2c, d and Supplementary Fig. S2e). Moreover, transmission electron microscopy (TEM) revealed that RSL3 treatment resulted in a

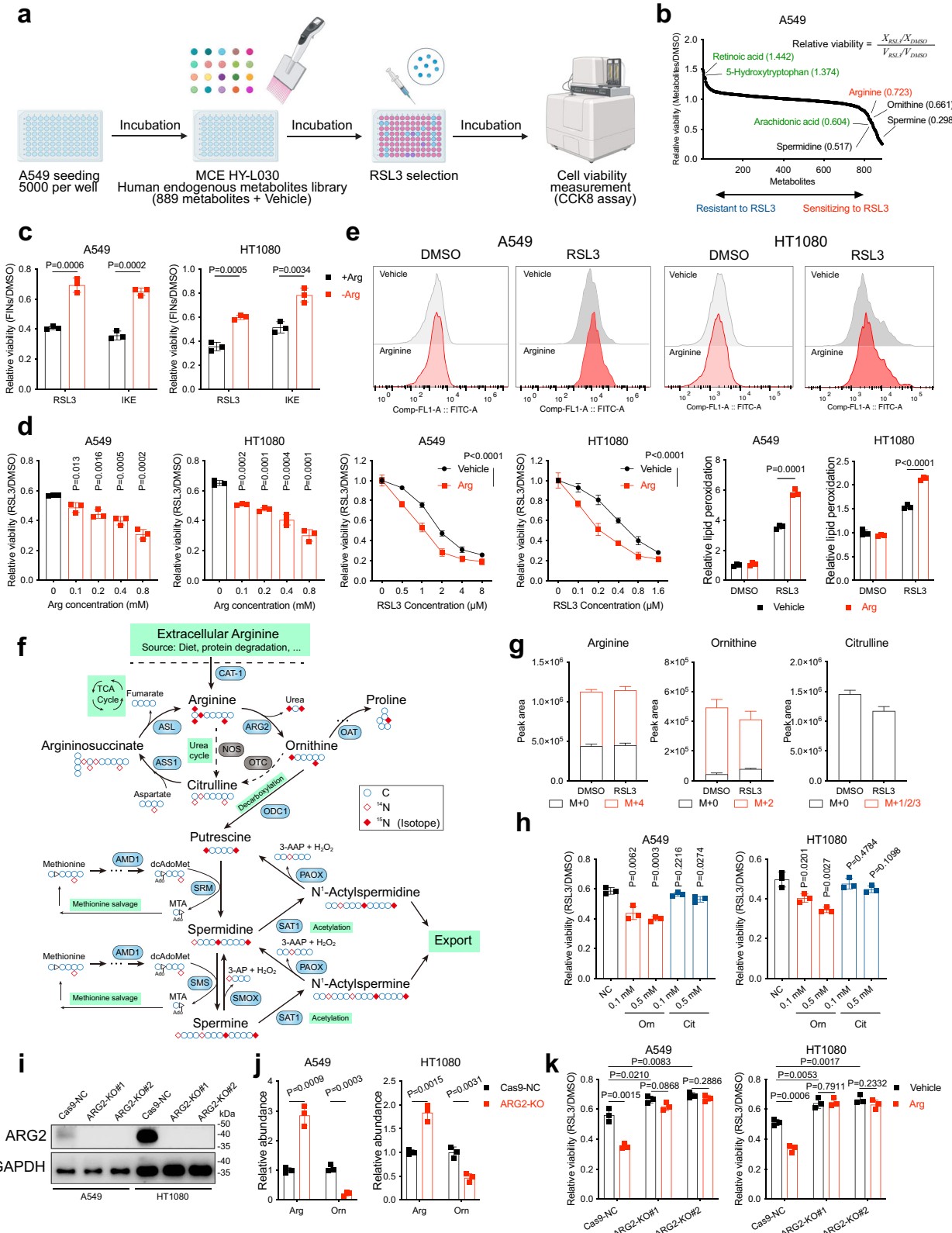

typical morphological feature of ferroptosis characterised by shrunken mitochondria and loss of cristae in A549, and pre-treatment with polyamines further amplified this effect (Fig. 2e). The ferroptosis-sensitising effect of polyamines was fully rescued by Fer-1 and DFO, but not by z-VAD(OMe)-FMK or necrosulfonamide, confirming that polyamine specifically promotes ferroptosis (Fig. 2f and Supplementary Fig. S2f). These findings firmly demonstrate the role of endogenous

metabolite, polyamines, in promoting lipid peroxidation and associated ferroptotic cell death. Previous researchers have demonstrated that polyamines could be catalysed into ammonia and aldehyde, which are toxic to cells, by the amine oxidase contained in bovine serum[18]. However, on the one hand, at the low concentration (5–10 μM) commonly used in this study, polyamines did not exert a significant negative impact on the cell viability when administered alone; on the

**Fig. 1 | Metabolite library screening links arginine and downstream ornithine to ferroptosis. a** Schematic of the metabolite library screening strategy. **b** The metabolite library screening results were exhibited as the relative viability of metabolite-treated cells versus vehicle-treated cells. After cell seeding, the A549 cells were pre-incubated with the metabolites contained in the library (MCE HY-L030) or Vehicle (DMSO) for 12 h. Then, the cells were further treated with 4 μM RSL3 or DMSO for 8 h and the cell viability was measured using CCK8. In the algorithm for the "Relative viability," "X" represents the pre-treatment of a specific metabolite, and "V" represents the pre-treatment of Vehicle DMSO. **c** Cell viability in A549 or HT1080 cells treated with RSL3 (A549: 4 μM for 8 h; HT1080: 0.5 μM for 6 h) following pre-incubation in arginine-supplemented or -depleted medium for 16 h. **d** Cell viability in A549 or HT1080 cells treated with RSL3 following pre-treatment with arginine (left: as indicated, right: 0.5 mM) for 4 h. **e** Lipid peroxidation in A549 or HT1080 cells treated with RSL3 (A549: 2 μM; HT1080: 0.25 μM) for 3 h following pre-treatment with arginine. **f** Schematic depiction of $^{15}N_4$-arginine tracing into arginine and polyamine metabolism. **g** Quantification of $^{15}N_4$ abundance in the indicated metabolites in DMSO or RSL3 (0.5 μM, 24 h) treated A549 using LC-MS. **h** Cell viability in A549 or HT1080 cells treated with RSL3 following pre-treatment with ornithine or citrulline as indicated. **i** ARG2 protein levels in Cas9-NC and ARG2-KO (#1 and #2) cancer cell lines determined by western blotting. **j** Relative abundance of arginine and ornithine in Cas9-NC and ARG2-KO cancer cells determined by LC-MS. **k** Cell viability in A549 or HT1080 cells with indicated genotypes treated with RSL3 following pre-treatment with arginine. **a** is generated using Biorender. Data are presented as the mean ± SD, *n* = 3 independent experiments. Unpaired two-tailed Student's *t* tests are used. Arg arginine, Orn ornithine, Cit citrulline. Source data are provided as a Source Data file.

other hand, we also ran the experiments with aminoguanidine, an inhibitor of the serum amine oxidase, and found that the ferroptosis-sensitising effect of polyamines was identical to the results above (Supplementary Fig. 2g). Therefore, we believed that polyamine's ferroptosis-sensitising could not be explained by the exogenous catalysis from amine oxidase.

We then further investigated the role of polyamines in arginine/ornithine's ferroptosis-promoting effect. The Cancer Therapeutics Response Portal (CTRP) reports correlations between gene expression and drug resistance for more than 900 cancer cell lines[19]. Data mined from the CTRP indicated that ODC1 expression significantly negatively correlates with resistance to multiple FINs, including RSL3, ML210, and erastin (Fig. 2g). In agreement with the CTRP results, the depletion of ODC1 not only protected cells from FINs-induced cell death but also completely abrogated arginine's pro-ferroptotic effect (Fig. 2h, i and Supplementary Fig. S2h). Sensitivity to FINs in ODC1-KO cells was restored by its re-expression, thus excluding the off-target effects of CRISPR/Cas9-mediated ODC1 knockout (Fig. 2j, k and Supplementary Fig. S2i). Meanwhile, pharmacological inhibition of ODC1 using difluoromethylornithe (DFMO) generated similar effect with ODC1 knockout (Fig. 2l and Supplementary Fig. S2j). Supportively, we further validated ODC1-KO caused polyamine reduction in A549 by LC-MS (Supplementary Fig. 2k). Taken together, these findings certify that polyamines, the final products of arginine/ornithine, mediate their pro-ferroptotic effect.

## Polyamine promotes ferroptosis in an $H_2O_2$-dependent manner

Spermidine is well-known as the sole substrate for DHPS/DOHH-catalysed two-step hypusination of eIF5A, a critical regulator of protein synthesis[15,20]. Therefore, we supposed whether polyamines promote ferroptosis by interfering with the translation of classic ferroptosis-associated factors such as ACSL4, SLC7A11 and GPX4. However, on the one hand, none of these proteins' expressions was significantly altered by polyamine treatment (Supplementary Fig. 3a). On the other hand, we treated cells with GC7, a specific inhibitor of DHPS, to block eIF5A, but the ferroptosis-sensitising effect of polyamines was not abolished, suggesting this signalling is dispensable for polyamines' pro-ferroptotic phenotype (Supplementary Fig. 3b). In addition, polyamines did not deplete intracellular GSH like IKE did (Supplementary Fig. 3c).

Having excluded canonical ferroptosis-regulating pathways, we then turned our attention to polyamines metabolism. As depicted in Fig. 1f, putrescine is synthesised by ODC1 from ornithine. S-Adenosylmethionine decarboxylase (AdoMetDC) provides putrescine with one or two aminopropyl groups and converts it to spermidine or spermine with the assistance of spermidine synthase (SRM) and spermine synthase (SMS) respectively[21]. This process is to some extent reversible, since spermidine/spermine $N^1$-acetyltransferase (SSAT) acetylates spermidine or spermine, enabling their oxidative back-conversion by peroxisomal acetylpolyamine oxidase (PAOX). Besides, spermine could also be directly catabolized to spermidine by spermine

oxidase (SMOX). Therefore, the metabolic pattern of polyamines is somewhat like a cycle as the three polyamine subtypes could mutually transform into each other. To understand whether polyamine sensitises cells to ferroptosis via its metabolic process, we designed a series of siRNAs targeting these genes and found that knockdown of these polyamine metabolism-related genes all more or less protected A549 and HT1080 from FINs-induced cell death, among which knockdown of PAOX and SMOX exhibited most potent protective effect (Fig. 3a and Supplementary Fig. S3d, e). Considering that the by-products of PAOX and SMOX activity include $H_2O_2$, which reacts with iron in a Fenton reaction to produce reactive hydroxyl radicals that carry out lipid peroxidation and subsequent ferroptotic cell death, it is conceivable that PAOX/SMOX-catalysed spermine/spermidine oxidation and $H_2O_2$ production accounts for polyamines' pro-ferroptotic role.

To verify this hypothesis, we generated PAOX/SMOX double-knockout cell lines (Fig. 3b). As expected, PAOX/SMOX-KO completely abrogated polyamines-induced ferroptosis sensitisation (Fig. 3c and Supplementary Fig. S3f). Furthermore, we depleted intracellular polyamines by simultaneously targeting their synthesis by DFMO and uptake by AMXT-1501. In these polyamine-depleted cells, PAOX/SMOX-KO failed to confer resistance to RSL3/IKE any more (Fig. 3d and Supplementary Fig. S3g). These results indicated that PAOX- and SMOX-catalysed metabolic processes are required for polyamines' ferroptosis-promoting effect.

Furthermore, we detected polyamine-dependent intracellular $H_2O_2$ generation by H2DCFDA. As expected, ODC1-KO decreased the $H_2O_2$ level (Fig. 3e). Conversely, polyamine supplementation led to $H_2O_2$ accumulation in both the presence and absence of RSL3, whereas PAOX/SMOX depletion largely abolished this effect (Fig. 3f, g). We further ectopically expressed catalase with its peroxisome location signal deleted (△catalase) in A549 and HT1080 (Supplementary Fig. 3h)[22]. The expressed △catalase significantly desensitised cells to RSL3 and IKE, and restored polyamine-treated cells' resistance (Fig. 3h and Supplementary Fig. S3i). Meanwhile, the ferroptosis-inhibiting effect conferred by PAOX/SMOX-KO was also abrogated by △catalase expression (Fig. 3i and Supplementary Fig. S3j). Taken together, polyamines' ferroptosis mediating effect is dependent on PAOX/SMOX-catalysed $H_2O_2$ generation. This mechanism explains why polyamines failed to sensitise cells to FINO2, an endoperoxide which directly initiates Fenton reaction by oxidising $Fe^{2+}$ to $Fe^{3+}$ and generating alkoxyl radical inducing lipid peroxidation[23]. That is, when FINO2 replaced $H_2O_2$ and induced ferroptosis independent of it, polyamine's ferroptosis-promoting effect afforded by $H_2O_2$ production was largely abolished. Among the three types of polyamines, spermine, whose conversion to putrescine leads to the production of two $H_2O_2$ molecules, exhibited the strongest ferroptosis-promoting property. Spermidine's conversion to putrescine generates one $H_2O_2$ molecule, making it the second strongest ferroptosis promoter. As for putrescine, only after being transformed to spermine or spermidine again can it exhibit a pro-ferroptotic role. Notably, polyamine's $H_2O_2$-dependent ferroptosis-sensitising effect also explains the interesting

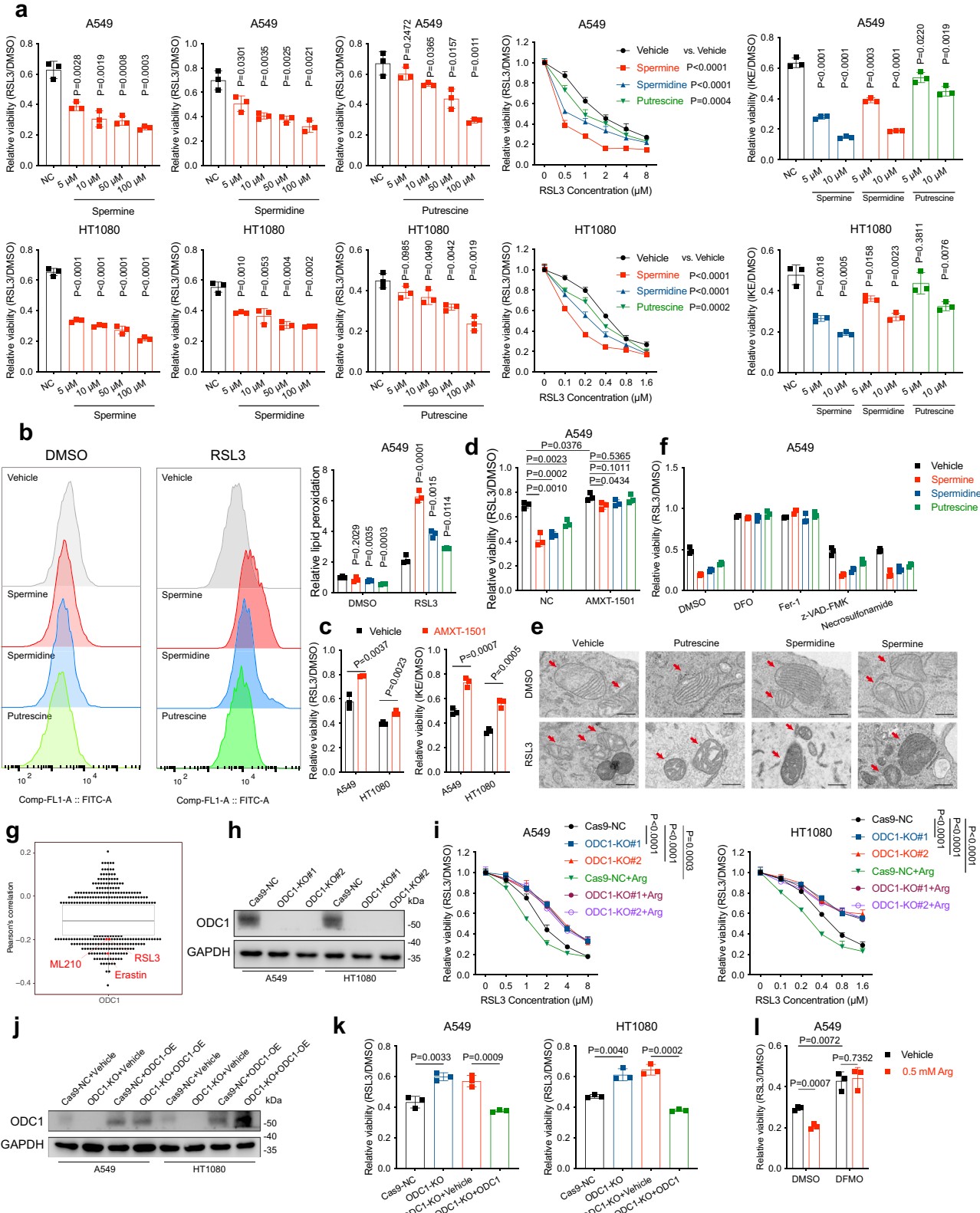

phenomenon that spermine and spermidine at 1 mM caused significant cell death in HT1080. This type of cell death could be partially reversed by DFO but not by Fer-1, which is consistent with $H_2O_2$'s iron-dependent cytotoxicity (Supplementary Fig. 3k)[24].

**Polyamine synthesis is activated during the ferroptosis process**
We next sought to determine whether polyamine metabolism is impacted by ferroptosis process. We performed RNA-Seq for A549 and

HT1080 treated with RSL3 and DMSO, and noticed that the expression of multiple polyamine metabolism-related genes was altered (Fig. 4a). Especially, of which ODC1, the rate-limiting enzyme of polyamine synthesis and an indicator of cell's response to ferroptosis mentioned above, was significantly upregulated after RSL3 treatment in both cell lines (A549: LogFC = 0.974, adjusted $P$ value < 0.001; HT1080: LogFC = 0.873, adjusted $P$ value < 0.001). We validated this finding by qPCR and western blotting, and the results suggested that RSL3 stimulated ODC1

**Fig. 2 | Polyamines mediate arginine and ornithine's pro-ferroptotic role. a** Cell viability in A549 or HT1080 cells treated with RSL3 or IKE (A549: 40 μM for 48 h; HT1080: 4 μM for 12 h) following pre-treatment with 10 μM polyamines (or as indicated) for 4 h. **b** Lipid peroxidation in A549 cells treated with RSL3 following pre-treatment with polyamines. **c** Cell viability in A549 or HT1080 cells treated with RSL3 or IKE following pre-treatment with 0.2 μM AMXT-1501 for 24 h. **d** Cell viability in A549 cells treated with 4 μM RSL3 for 8 h following pre-treatment with AMXT-1501 and polyamines. **e** Transmission electron microscopy images of A549 cells treated with 2 μM RSL3 for 4 h following pre-treatment with 10 μM polyamines for 4 h. Scale bars: 4 μm. **f** Cell viability A549 cells treated with RSL3 combined with or without DFO (100 μM), Fer-1 (10 μM), z-VAD-FMK (10 μM), or necrosulfonamide (0.5 μM) for 8 h following pre-treatment with polyamines. **g** Correlation between ODC1 expression and cancer cells' sensitivity to ferroptosis inducers (RSL3, Erastin,

ML210). Plotted data were mined from the CTRP database[19]. Plotted values are Pearson's correlation coefficients. Box plot indicates median, 25th and 75th percentiles, and minima and maxima of the distributions. n = 545 drugs. P value for correlation test: RSL3: $9.52 \times 10^{-8}$; Erastin: $2.29 \times 10^{-14}$; ML210: $1.04 \times 10^{-7}$. **h** ODC1 protein levels in Cas9-NC and ODC1-KO (#1 and #2) cancer cell lines determined by western blotting. **i** Cell viability in A549 or HT1080 cells with indicated genotypes treated with RSL3 following pre-treatment with arginine. **j** ODC1 protein levels in A549 or HT1080 cells with indicated genotypes determined by western blotting. **k** Cell viability in A549 or HT1080 cells with indicated genotypes treated with RSL3. **l** Cell viability in A549 or HT1080 cells treated with RSL3 following pre-treatment with 500 μM DFMO for 16 h and arginine. Data are presented as the mean ± SD, n = 3 independent experiments. Unpaired two-tailed Student's *t* tests are used. Source data are provided as a Source Data file.

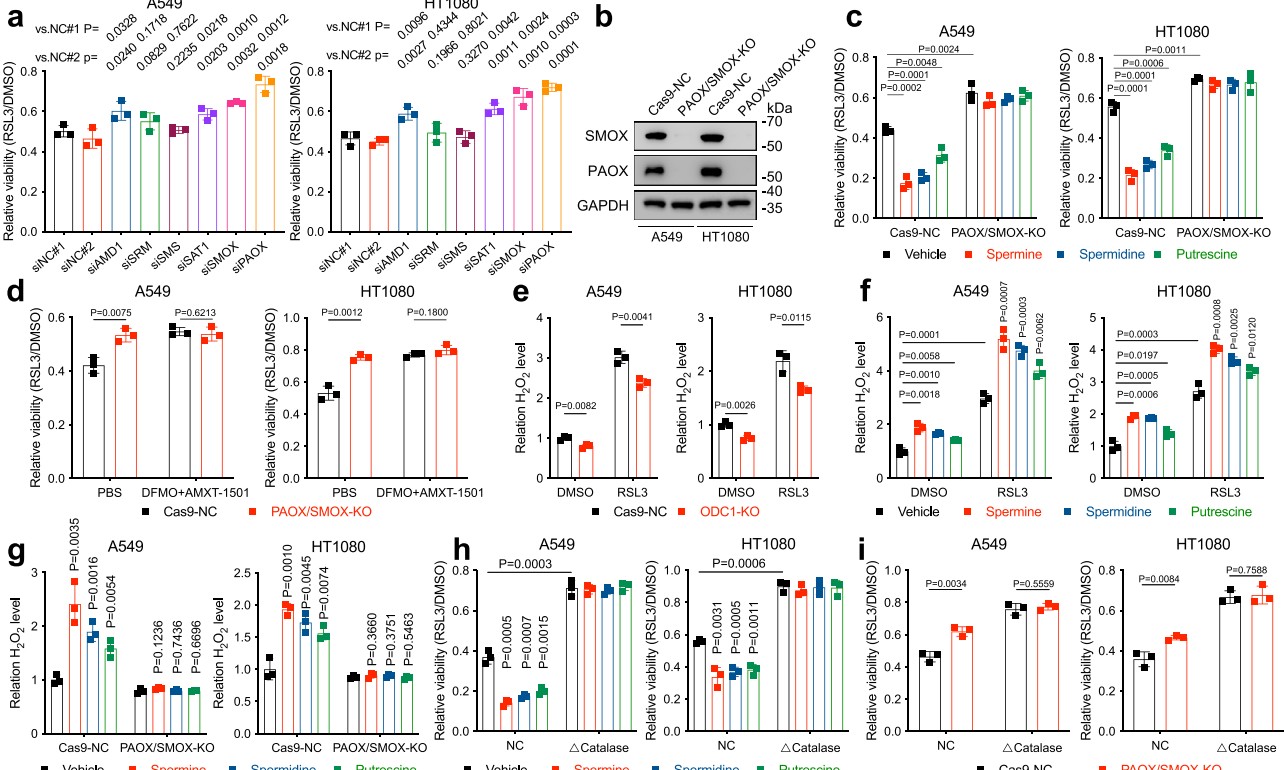

**Fig. 3 | Polyamines' ferroptosis-promoting property depends on PAOX/SMOX mediated H$_2$O$_2$ production. a** Cell viability in A549 or HT1080 cells treated with RSL3 following siRNAs transfection for 48 h. **b** The protein levels of PAOX and SMOX in cancer cell lines with indicated genotypes determined by western blotting. **c** Cell viability in Cas9-NC and PAOX/SMOX-KO A549 or HT1080 cells treated with RSL3 following pre-treatment with polyamines. **d** Cell viability in Cas9-NC and PAOX/SMOX-KO A549 or HT1080 cells treated with RSL3 following pre-treatment with AMXT-1501 and DFMO. **e** Relative H$_2$O$_2$ level in Cas9-NC and ODC1-KO A549 or HT1080 cells treated with RSL3 (A549: 2 μM; HT1080: 0.25 μM) for 3 h, determined

by H$_2$DCFDA. **f** Relative H$_2$O$_2$ level in A549 or HT1080 cells treated with RSL3 following pre-treatment with polyamines. **g** Relative H$_2$O$_2$ level in Cas9-NC and PAOX/SMOX-KO A549 or HT1080 cells treated with RSL3. **h** Cell viability in vehicle- and △catalase-overexpressed A549 or HT1080 cells treated with RSL3 following pre-treatment with polyamines. **i** Cell viability in A549 or HT1080 cells with indicated genotypes treated with RSL3. Data are presented as the mean ± SD, *n* = 3 independent experiments. Unpaired two-tailed Student's *t* tests are used. Source data are provided as a Source Data file.

expression in a concentration-dependent manner (Fig. 4b, c and Supplementary Fig. S3l). Furthermore, we measured the abundance of [15]N-labelled polyamines generated in A549 after 9-hour treatment with RSL3 or DMSO in [15]N$_4$-arginine-containing DMEM medium using an LC-MS method. In support of RSL3-induced ODC1 activation, in DMSO-treated A549, only 4.1%, 2% and 16.8% of the putrescine, spermidine, and spermine acquired [15]N label, while in RSL3-treated cells the numbers were 16.7%, 3.2% and 24.2%, suggesting enhanced transition of arginine to polyamines after RSL3 treatment (Fig. 4d). Similar results were obtained in classic metabolomics (Fig. 4e). Together, these findings reveal that ODC1 expression is activated by RSL3 treatment, leading up to upregulated polyamine synthesis in ferroptosis process.

## Iron overload occurred during ferroptosis process triggers ODC1 upregulation

Here we noticed an interesting phenomenon that treating cells with ferroptosis inhibitor Fer-1 and DFO all completely inhibited RSL3-induced cell death, nevertheless, only DFO reversed RSL3-induced ODC1 upregulation (Fig. 4b, c). This unexpected finding prompted us to associate ODC1 regulation with the difference between the mechanisms of DFO and Fer-1's ferroptotic inhibiting function. DFO, an iron chelator, inhibits ferroptosis by blocking liable iron and the corresponding iron-dependent Fenton reaction, whereas Fer-1, an antioxidant, scavenges lipid peroxides, cannot completely mitigate iron overload as DFO does, an important initiation signal of ferroptosis[1,25–27]. We confirmed DFO, Fer-

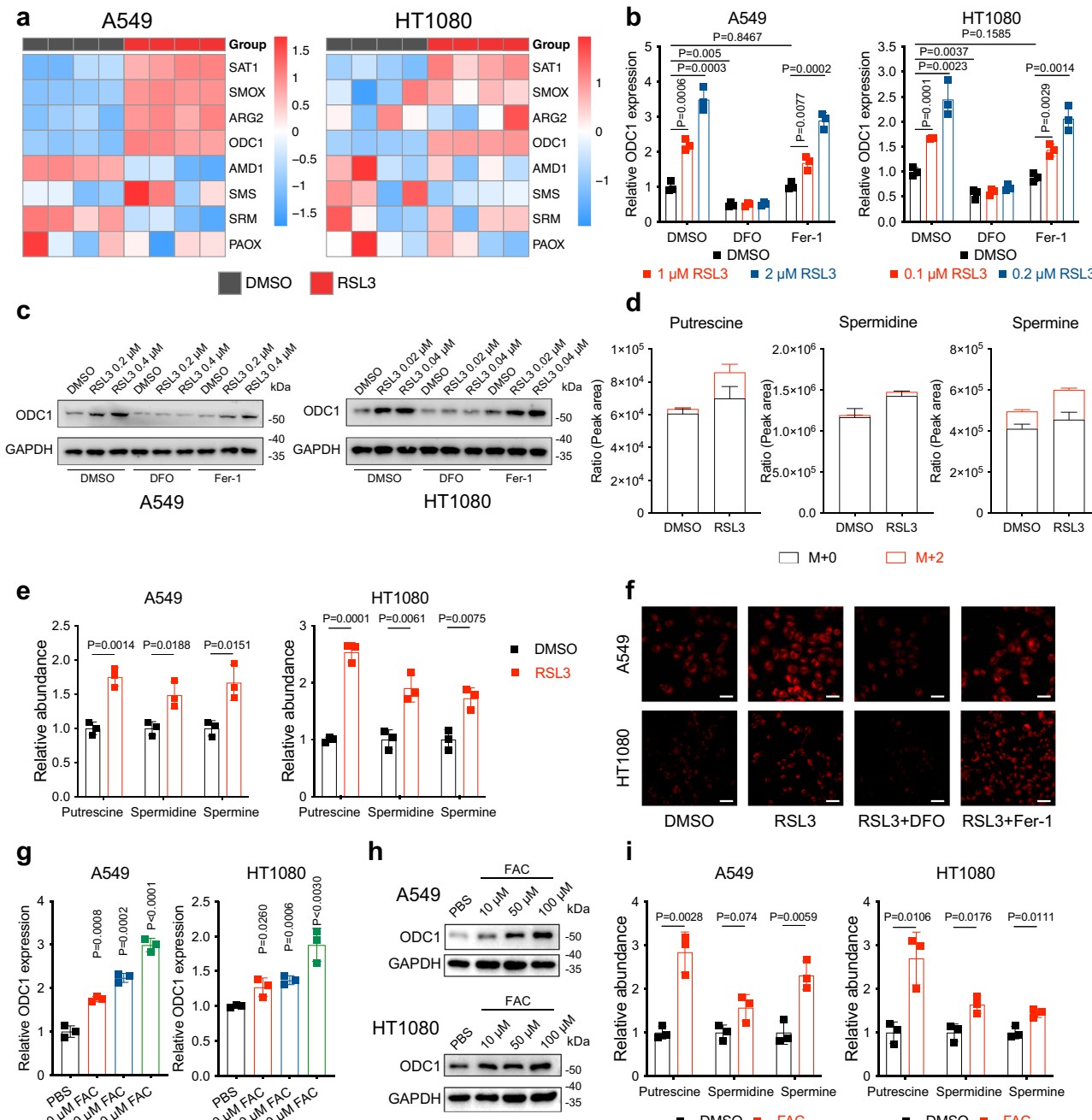

**Fig. 4 | Ferroptotic iron overload triggers ODC1 expression. a** Heatmaps exhibiting the expression levels of a series of polyamine metabolism-related genes in A549 or HT1080 cells treated with DMSO or RSL3 (A549: 2 µM; HT1080: 0.2 µM) for 4 h, determined by RNA-seq, $n = 4$. **b** mRNA levels of ODC1 in A549 or HT1080 cells treated with DMSO or RSL3 (A549: 0.5 µM; HT1080: 0.05 µM) combined with or without DFO or Fer-1 for 24 h, determined by qPCR. **c** Protein levels of ODC1 in A549 or HT1080 cells treated with DMSO or RSL3 (A549: 0.5 µM; HT1080: 0.05 µM) combined with or without DFO or Fer-1 for 24 h, determined by western blotting. **d** Quantification of $^{15}N_4$ abundance in the indicated polyamines in DMSO or RSL3 (0.5 µM for 24 h) treated A549 using LC-MS. **e** Relative abundance of polyamines in A549 and HT1080 cells treated with DMSO or RSL3 (A549: 0.5 µM; HT1080: 0.05 µM) for 24 h, determined by LC-MS. **f** Confocal microscope images of FerroOrange-stained A549 or HT1080 cells treated with RSL3 (A549: 2 µM; HT1080: 0.25 µM) combined with or without DFO (100 µM) or Fer-1 (10 µM) for 3 h (scale bars, 100 µm). **g, h** mRNA (**g**) and protein (**h**) levels of ODC1 in A549 or HT1080 cells treated with FAC as indicated for 24 h. **i** Relative abundance of polyamines in A549 and HT1080 cells treated with FAC for 24 h, determined by LC-MS. Data are presented as the mean ± SD, $n = 3$ independent experiments unless otherwise stated. Unpaired two-tailed Student's $t$ tests are used. Source data are provided as a Source Data file.

1, and $H_2O_2$'s impact on RSL3-induced iron overload by staining intracellular liable iron with FerroOrange (Fig. 4f and Supplementary Fig. S3m). Similar results obtained from another type of probe have also been reported in ref. 28. Furthermore, we then supplemented cells with ferric ammonium citrate (FAC) to mimic iron overload[26,29]. As expected, FAC treatment resulted in significant upregulated ODC1 mRNA and protein, as well as enhanced polyamine abundance (Fig. 4g–i). Together,

these evidences demonstrate that iron overload during ferroptosis process is responsible for RSL3-induced ODC1 activation.

**Iron overload upregulates ODC1 through the WNT/MYC pathway and amplifies ferroptosis**
Iron not only serves as an essential nutrient that facilitates cell proliferation but also participates in several canonical signalling

pathways[30]. Notably, the link between iron and WNT signalling has been implicated in several cancer-related research[31–33]. Generally, in cells with activated WNT signalling, β-catenin enters the nucleus and promotes T-cell factor (TCF)-lymphoid enhancer factor (LEF)-dependent transcription of a series of downstream target genes, such as MYC, whose direct transcriptive targets include ODC1[21,34,35]. Thus, it is reasonable to speculate that iron overload stimulates ODC1 expression through the WNT/MYC signalling pathway. To verify this hypothesis, we first treated A549 and HT1080 cells with FAC and RSL3 to induce iron overload and detected the activation level of WNT signalling using MYC as a specific indicator. As exhibited in Fig. 5a, b, both of them significantly increased the mRNA and protein levels of MYC, whereas DFO supplementation completely abolished this alteration. Furthermore, we noticed that treating cells with WNT-specific agonist SKL2001 substantially upregulated ODC1 and MYC levels, while WNT inhibitor LF3 generated the opposite result (Fig. 5c, d). Consistently, we further observed that SKL2001 and LF3 treatment respectively sensitised and desensitised cells to FINs in a concentration-dependent manner, suggesting WNT signalling pathway impairs cells' ferroptosis resistance (Fig. 5e, f and Supplementary Fig. S4a, b). SKL2001-caused ferroptosis sensitisation could be fully rescued by Fer-1 and DFO (Supplementary Fig. 4c). Besides, to ascertain whether MYC directly activates *ODC1*'s transcription, we took advantage of the ENCODE database and confirmed the enrichment of MYC at the promoter region of *ODC1* gene (Supplementary Fig. 4b)[36]. Multiple potential MYC binding sites (BS) were also identified through JASPAR (Supplementary Fig. 4e). In agreement with these predictions and previous reports, our CHIP assay revealed significant enrichment of MYC to BS2 and BS3 (Fig. 5g). To further confirm the binding specificity, a luciferase reporter gene assay was performed using plasmids containing *ODC1* promoter region (WT) and introduced mutations in the MYC binding sites (MUT). As shown in Fig. 5h, both FAC and SKL2001 treatment resulted in extensive transcriptional activation, while the iron chelator DFO led to inhibition, as reflected by the luciferase intensity in cells transfected with WT but not the all-MUT plasmid. Single mutation at the two binding sites also mitigated these changes to varying degrees. siRNA-mediated MYC knockdown also decreased *ODC1* mRNA level and blocked RSL3/FAC-induced ODC1 upregulation (Supplementary Fig. 4f, g). These data demonstrate that MYC directly activates *ODC1* transcription. Moreover, pre-treatment with WNT inhibitor LF3 largely, but not completely abrogated FAC or RSL3-induced MYC/ODC1 upregulation, suggesting the existence of other unknown mechanisms accounting for this phenomenon (Fig. 5i, j), although they are not the dominant factor.

Taken together, our data imply that ferroptotic iron overload activates the transcription of *ODC1* through the WNT/MYC signalling pathway. Combined this conclusion with the pro-ferroptotic role of ODC1 and corresponding polyamine synthesis mentioned above, these findings demonstrate that ODC1 sensed the important ferroptosis signal—iron overload in a WNT/MYC-dependent manner, and then catalysed the synthesis of polyamine, a potent ferroptosis mediator, thus forming a positive-feedback loop consisting of ferroptosis-iron-WNT/MYC-ODC1-polyamine-ferroptosis, which may function as a core ferroptosis amplifier.

## EV-dependent polyamine efflux in ferroptotic cancer cells expands the range of ferroptosis

It is known that dying cells, especially in the context of ferroptosis, communicate with the tumour microenvironment by a set of signals produced during cell death, such as the pro-inflammatory 5-HETE and HMGB1[37,38]. As such, considering that polyamines are abundant components of the extracellular environment, we determined their levels in cell culture medium on triggering ferroptosis by targeted LC-MS. The abundance of polyamines was increased in the medium of RSL3-treated A549 and HT1080 cells, demonstrating increased polyamine efflux during the ferroptosis process (Fig. 6a). Notably, the enhanced polyamine efflux lasted for several hours after the initial onset of ferroptosis, as we still detected increased polyamines after the RSL3-containing medium was replaced by normal fresh medium after RSL3 treatment and thorough flush with PBS (Fig. 6b). It is thus tempting to speculate that the released polyamines from ferroptotic cells might enter adjuvant non-ferroptotic cells and further render them sensitive to ferroptosis. To verify this hypothesis, we treated healthy A549 and HT1080 cells for 4 h with supernatants from either DMSO- or RSL3-treated ferroptotic cells and observed increased sensitivity to ferroptosis in A549 and HT1080 exposed to supernatants from the latter cells, whereas this effect was substantially abrogated by polyamine uptake inhibitor AMXT-1501(Fig. 6c, d). Similarly, in PAOX/SMOX-KO cells which cannot catalyse polyamine-dependent $H_2O_2$ production, A549 and HT1080 cultured in supernatants from ferroptotic or non-ferroptotic cells exhibited an identical response to RSL3, confirming the critical role of polyamine in this process (Fig. 6d). Collectively, these findings indicate that ferroptotic cells release polyamine to extracellular space, thus expanding the region of ferroptotic sensitisation.

Extracellular vesicles (EVs) transport various cargos among different cells[39,40]. Meanwhile, polyamine-sequestering vesicles (PSVs) are known to be the major approach to polyamine transportation[21,41]. Therefore, to examine whether polyamine is released from ferroptotic cells in an EV-dependent manner, we purified EVs from the medium of DMSO- or RSL3-treated A549 and HT1080 using the ultracentrifugation-based protocol[39]. As shown in Fig. 6e, polyamines were detectable within cell-derived EVs. Although the polyamine contents of EVs from DMSO- and RSL3-treated cells were nearly identical, ferroptotic cells produced more EVs than healthy cells, as evidenced by the larger amount of EV marker proteins presented in RSL3-treated cells (Fig. 6f). However, FAC treatment failed to generate similar effects, suggesting that the increased EV production was not a result of iron overload (Supplementary Fig. 4h).

Concomitant with our above-mentioned results, pre-treatment with ferroptotic cells-derived EVs rendered healthy cells sensitive to RSL3, whereas AMXT-1501 treatment or PAOX/SMOX depletion substantially abolished this effect (Fig. 6g). Taken together, these results support the notion that ferroptotic cells not only enhance intracellular polyamine synthesis but also release polyamine into the tumour microenvironment in an EV-dependent manner. This phenomenon might be regarded as a spontaneous self-protective mechanism to export excessive polyamine produced during the ferroptosis process. However, the released polyamine further sensitised neighbouring cells to FINs, thereby accelerating the "spread" of ferroptosis in the tumour region.

## Polyamine supplementation sensitises lung cancer cells to radiotherapy and chemotherapy by inducing ferroptosis

Radiotherapy (RT) and chemotherapy are effective routes for lung cancer therapy. Lei et al. have previously shown that ionising radiation induces ferroptosis[5], and a published study of our team has revealed an important role of ferroptosis in cisplatin (CDDP) induced cell death[6]. These evidences prompt us to examine the potential role of polyamine's pro-ferroptotic effect in the modulation of radio-and chemo-resistance in cancer cells. As expected, exogenous supplementation of polyamines significantly sensitised cells to RT or CDDP, as well as amplified the lipid peroxidation induced by these treatments (Fig. 7a–c). Importantly, the radio-/chemo-sensitising effect of polyamine could be largely abolished by the ferroptosis inhibitor Fer-1 (Fig. 7d). Collectively, these data suggest that polyamine treatment promotes radio-/chemo-sensitisation in lung cancer cells mainly through inducing ferroptosis.

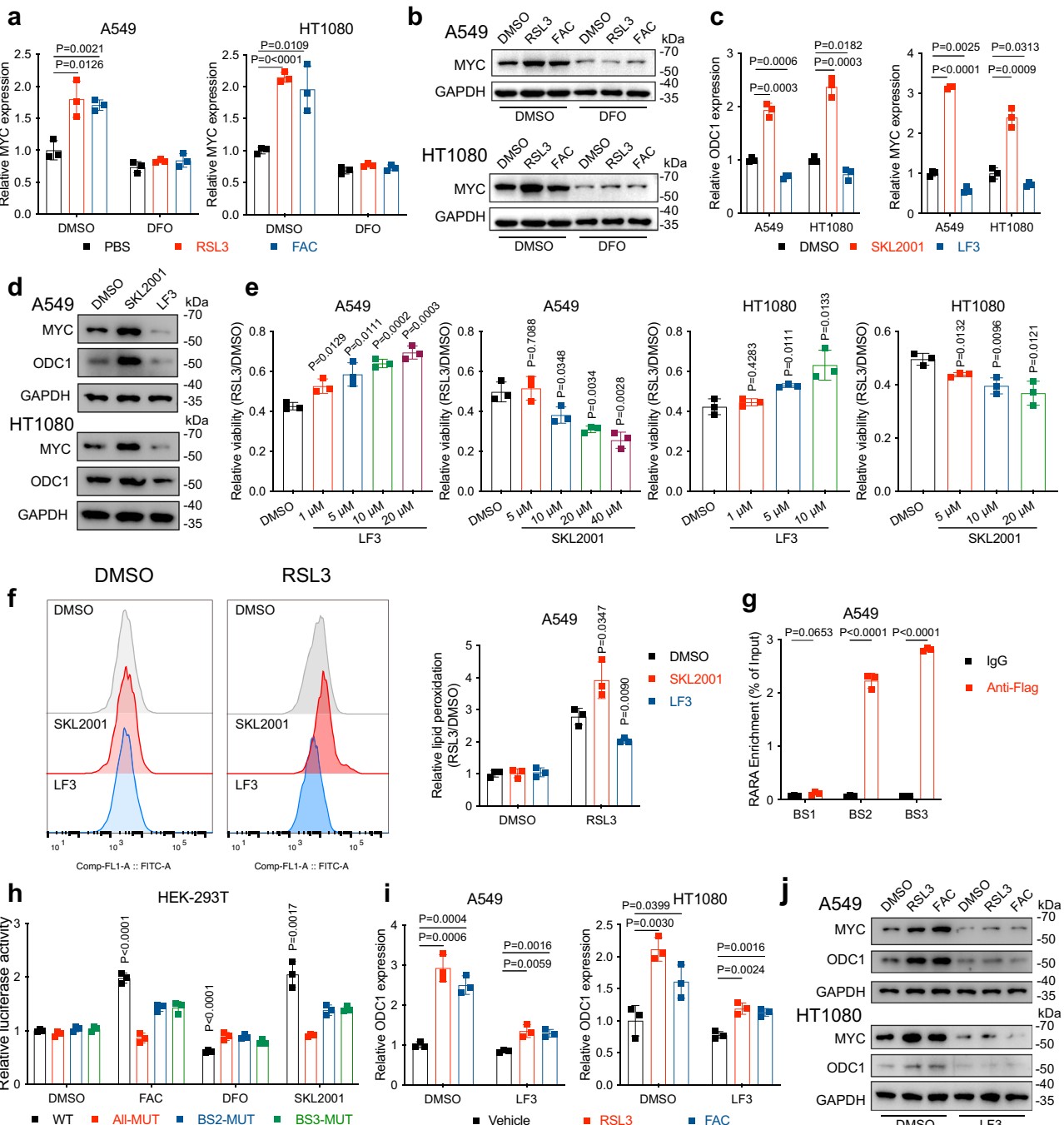

**Fig. 5 | Iron overload promotes ODC1 expression in an WNT/MYC-dependent manner. a**, **b** mRNA (**a**) and protein (**b**) levels of MYC in A549 or HT1080 cells treated with DMSO, 50 μM FAC, or RSL3 combined with or without DFO or Fer-1. **c**, **d** mRNA (**c**) or protein (**d**) levels of ODC1/MYC in A549 or HT1080 cells treated with 10 μM LF3 or SKL2001 for 24 h. **e** Cell viability in A549 or HT1080 cells treated with RSL3 following pre-treatment with LF3 or SKL2001 as indicated. **f** Lipid peroxidation in A549 treated with RSL3 following pre-treatment with 10 μM LF3 or SKL2001 for 24 h. **g** The binding between MYC and the promoter region of ODC1 was quantified through a CHIP assay followed by qPCR. **h** Luciferase activity in HEK-293T cells treated with for FAC, DFO or SKL2001 for 24 h, following transfection of indicated plasmids. **i** mRNA level of ODC1 in A549 or HT1080 cells treated with DMSO, FAC, or RSL3 following pre-treatment of LF3. **j** Protein level of ODC1 and MYC in A549 or HT1080 cells treated with DMSO, FAC or RSL3 following pre-treatment of LF3. Data are presented as the mean ± SD, *n* = 3 independent experiments. Unpaired two-tailed Student's *t* tests are used. Source data are provided as a Source Data file.

## Polyamines sensitise tumour to ferroptosis as well as RT/CDDP treatment in vivo

We then investigated the role of polyamine supplementation in lung tumour growth and ferroptosis in vivo. Tumour xenografts were generated with Cas9-NC and PAOX/SMOX-KO A549 cell lines, and IKE was used as the FIN for its excellent stability in animals[42]. Consistent with previous reports, singly administered spermine slightly promoted tumour growth (Fig. 8a). This phenomenon might be due to polyamine's immunosuppressive property that impaired anti-tumour immune response[21]. IKE treatment led to a marked reduction in the growth of tumours, while spermine further amplified this effect in Cas9-NC tumours, but not in PAOX/SMOX-KO tumours (Fig. 8a). Consistently, staining of 4-hydroxy-2-nonenal (4-HNE) in these tumour samples revealed that spermine treatment enhanced the level of lipid

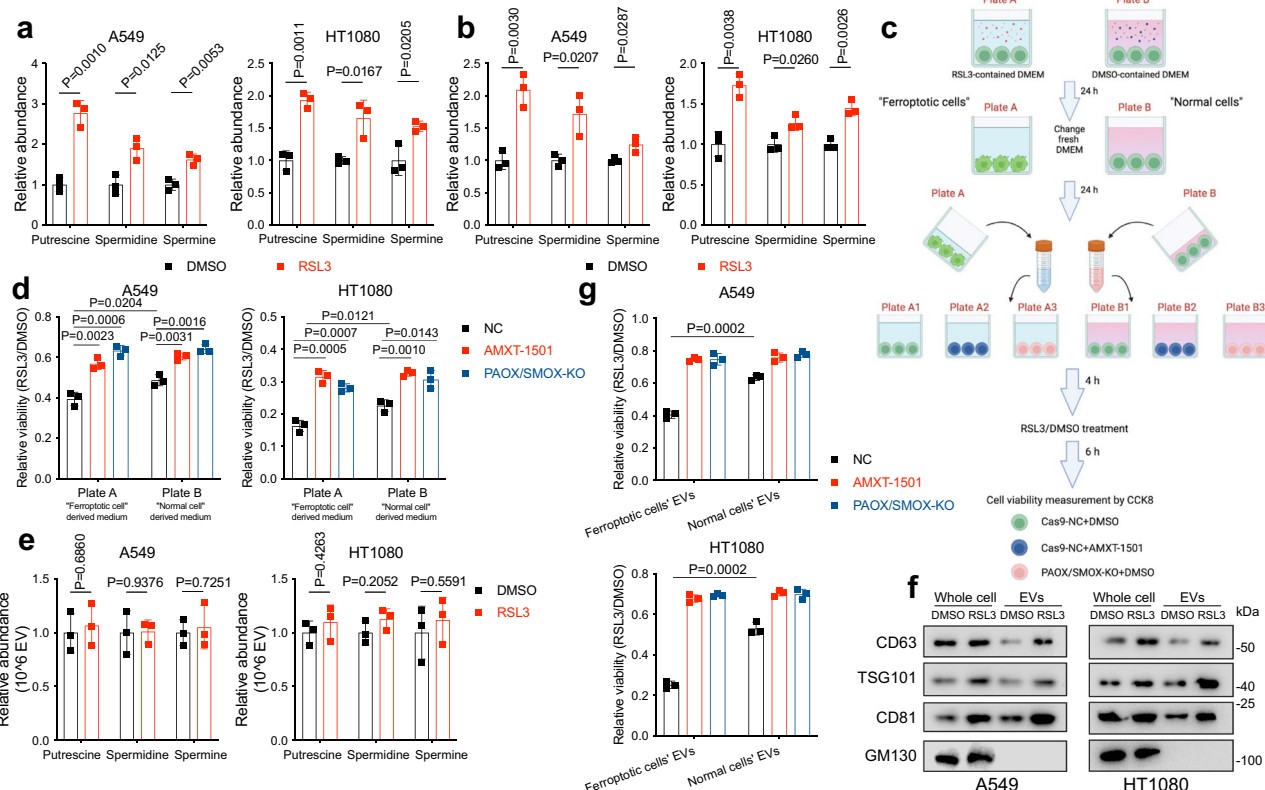

**Fig. 6 | Ferroptotic cells release polyamine-containing extracellular vesicles (EVs) and accelerate the spread of ferroptosis to surrounding healthy cells.**
**a**, **b** The cells were treated by DMSO- or RSL3 (A549: 0.5 µM; HT1080: 0.05 µM) containing medium for 24 h, then the medium was collected (**a**). The treated cells were then cultured in the same volume of fresh medium, and the new medium was also collected after 24 h incubation (**b**). Relative abundance of polyamines in the collected medium was determined by LC-MS. **c**, **d** Schematic depiction of the medium-exchange assay (**c**). Briefly, the cells seeded in 9-cm dishes were treated by DMSO- (plate B) or RSL3 (A549: 0.5 µM; HT1080: 0.05 µM, plate A) containing medium for 24 h, then the medium was discarded and the cells were washed with PBS. The cells were then incubated with the same volume of fresh medium for 24 h, and the new medium was collected as "medium B" or "medium A" and used for

subsequent experiments. Healthy A549 or HT1080 with indicated genotypes were seeded in 96-well plates. After pre-treatment with AMXT-1501, the cells were then cultured in "medium B" or "medium A" as indicated for 4 h, followed by RSL3 treatment and cell viability assessment using CCK8 (**d**). **e** Relative polyamine levels within EVs derived from DMSO- or RSL3-treated A549 and HT1080 cells. **f** EVs were isolated from cell culture medium of A549 and HT1080 cells treated with DMSO or RSL3 (A549: 0.5 µM; HT1080: 0.05 µM) for 24 h. The cells and EVs were lysed and the protein levels of CD63, CD81, TSG101 and GM130 were determined by western blotting. **g** Cell viability in A549 or HT1080 cells treated with RSL3 following pre-treatment with EVs derived from (**e**) and (**f**). Panel **c** is generated using Biorender. Data are presented as the mean ± SD, n = 3 independent experiments. Unpaired two-tailed Student's t tests are used. Source data are provided as a Source Data file.

peroxidation caused by IKE. In contrast, this effect was normalised by PAOX/SMOX depletion (Fig. 8b).

Moreover, we also tested the therapeutic potential of combining polyamine with RT or CDDP for the treatment of lung cancer in vivo. As expected, orally administered spermine significantly enhanced RT and CDDP's tumour-suppressive effect, which correlated with increased 4-HNE staining (Fig. 8c, d). Importantly, in these experiments, the treatments of spermine, IKE, RT, or CDDP did not significantly affect animal weight, suggesting the treatment was well-tolerated in our animal model (Supplementary Fig. 4i). Taken together, our data substantiate that spermine supplementation renders tumours sensitive to ferroptosis in vivo, and raised the possibility of combining RT or CDDP with polyamine in future cancer treatment for better effect.

### Upregulated polyamine synthesis renders cancer cells intrinsically susceptible to ferroptosis

Through the gene expression analysis of the TCGA database, we found that *ODC1* expression levels were markedly upregulated in multiple cancers compared to their corresponding normal tissues, including lung adenocarcinoma (LUAD), lung squamous carcinoma (LUSC), oesophageal cancer (ESCA), glioblastoma (GBM), head and neck cancer (HNSC), et al (Supplementary Fig. 4j)[43]. Meanwhile, a published single-cell RNA-seq study of our group focusing on 17 LUAD tissue and

12 normal lung tissue also indicated upregulated ODC1 levels in cancer cells compared with other cell types (Supplementary Fig. 4k, l)[44]. We validated this bioinformatic result through western blotting using the surgical resected LUAD samples obtained from our institution, thus implicating enhanced polyamine synthesis in the tumoral region (Supplementary Fig. 4m). Supportively, we performed spatially resolved metabolomics on a LUAD tissue section and distinguished tumoral and normal tissue using hematoxylin-eosin (H&E) staining. As exhibited in Fig. 8e, we noted significant enrichment of putrescine, spermidine, and spermine in the tumour region, which was in agreement with previous researchers' findings[21]. Generally, upregulated polyamines are considered to contribute to tumour transformation, proliferation, and progression[15]. However, taking the potent proferroptotic effect of polyamine into account, the elevated polyamine synthesis is no longer only a "bad boy" that leads to tumour development, instead, it renders cancer cells intrinsically susceptible to ferroptosis, thereby exposing vulnerabilities that could be therapeutically targetable in certain context.

## Discussion

Ferroptosis, as a unique iron-dependent cell death mechanism, has sparked great interest in the cancer research community as targeting ferroptosis might provide therapeutic opportunities in treating

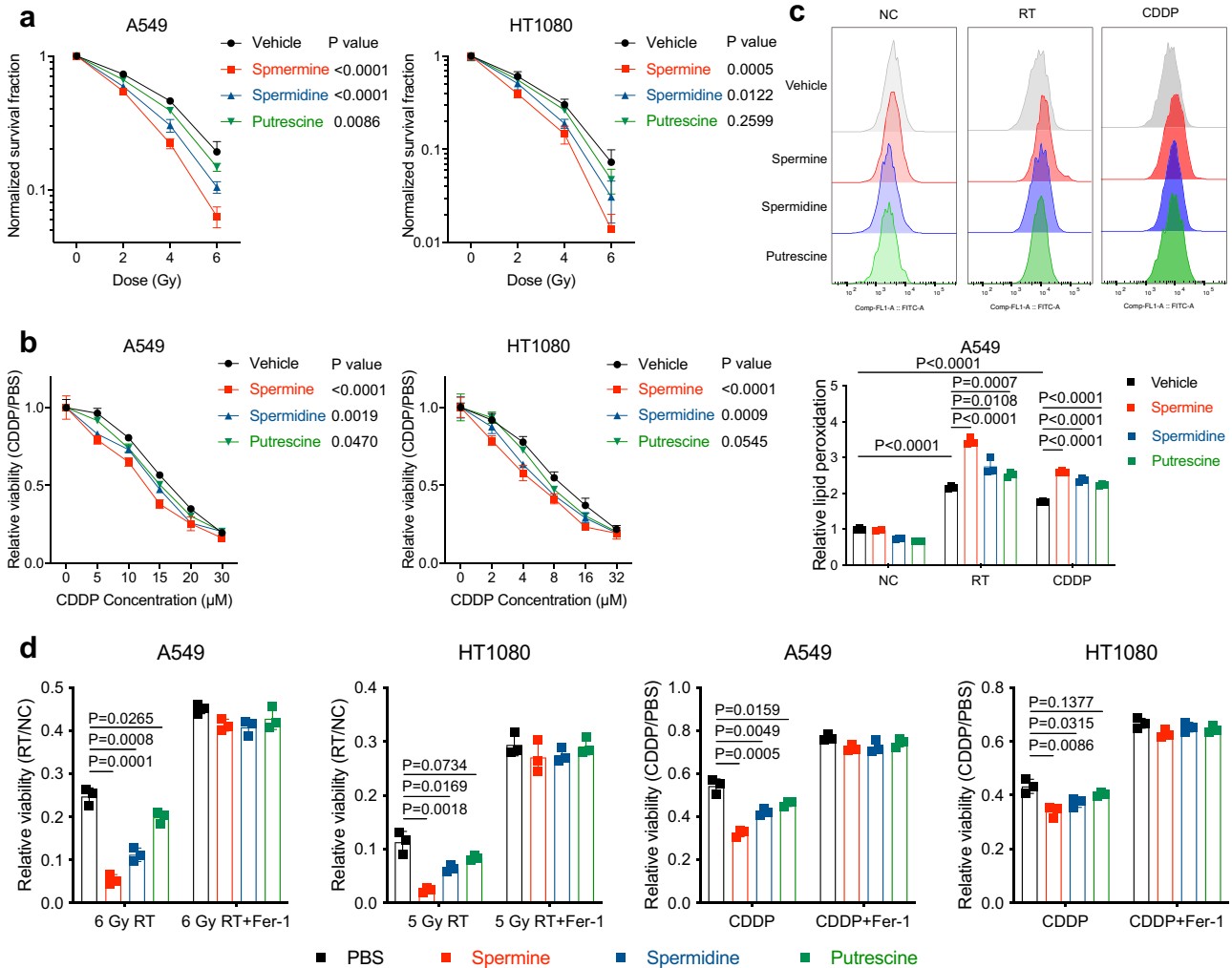

**Fig. 7 | Polyamine promotes radiotherapy- and chemotherapy-induced ferroptosis. a** Clonogenic survival curves of A549 or HT1080 cells exposed to radiotherapy (RT) at indicated doses following pre-treatment with polyamines. **b** Cell viability in A549 or HT1080 cells treated with CDDP as indicated for 48 h following pre-treatment with polyamines. **c** Lipid peroxidation assessment in A549 or HT1080 cells at 24 h after exposure to 6 Gy of RT or 10 μM CDDP following pre-treatment with polyamines. **d** (left) Clonogenic survival fraction of A549 or HT1080 cells exposed to radiotherapy (RT) at indicated doses combined with or without Fer-1 following pre-treatment with polyamines. (right) Cell viability in A549 or HT1080 cells treated with CDDP (A549: 20 μM; HT1080: 10 μM) combined with or without Fer-1 for 48 h following pre-treatment with polyamines. Data are presented as the mean ± SD, *n* = 3 independent experiments. Unpaired two-tailed Student's *t* tests are used. Source data are provided as a Source Data file.

cancers that are resistant to conventional therapies[3,45]. It is known to all that cancer cells are characterised by distinctive metabolic features to fulfil their increased demand for nutrients and energy for continuous proliferation. This kind of reprogramming, including the high load of reactive oxygen species (ROS) and enhanced PUFA-PL, renders some of the cancer cells intrinsically susceptible to ferroptosis, thereby exposing vulnerabilities that could be therapeutically targetable. Meanwhile, cancer cells also develop some ferroptosis defence systems by shifting certain metabolic pathways towards a protective phenotype, such as GCLC-mediated synthesis of ferroptosis inhibitor γ-glutamyl-peptide[46]. Therefore, understanding the crosstalk between tumour metabolism and ferroptosis is increasingly recognised as a potential avenue for developing anticancer strategies[47,48].

In this study, after the identification of the ferroptosis-promoting role of arginine using the metabolites library, we confirmed that the arginine-ornithine-polyamine axis emerges as the dominant metabolic pathway of arginine. Importantly, polyamines, including spermine, spermidine, and putrescine, significantly sensitised cancer cells to ferroptosis in an $H_2O_2$-dependent manner. Furthermore, we observed elevated polyamine synthesis in ferroptotic cells. Mechanically, the

important ferroptosis signal—iron overload—triggers the transcription of ODC1, the critical enzyme catalysing polyamine synthesis, through WNT/MYC signalling, thus forming a ferroptosis-iron overload-WNT/MYC/ODC1-polyamine-$H_2O_2$ positive-feedback axis that amplifies ferroptosis. Moreover, we noticed that ferroptotic cells release more polyamine-containing extracellular vesicles into the microenvironment, thereby further sensitising neighbouring cells to ferroptosis and accelerating the "diffusion" of ferroptosis in the tumour region (Fig. 8f). Besides, polyamine supplementation also sensitises cancer cells or xenograft tumours to radiation or CDDP by inducing ferroptosis, raising the possibility of combining canonical radiotherapy or chemotherapy with polyamines in cancer treatment for better response. In addition, we validated the upregulated ODC1 expression and polyamine synthesis in lung adenocarcinoma tissue compared with adjacent normal tissue. Considering the potent pro-ferroptotic effect of polyamines, the elevated polyamines in cancer cells might be a vulnerability that could be targetable in ferroptosis-based cancer treatment.

Emerging evidence has revealed that amino acid metabolism plays an important role in the initiation and regulation of

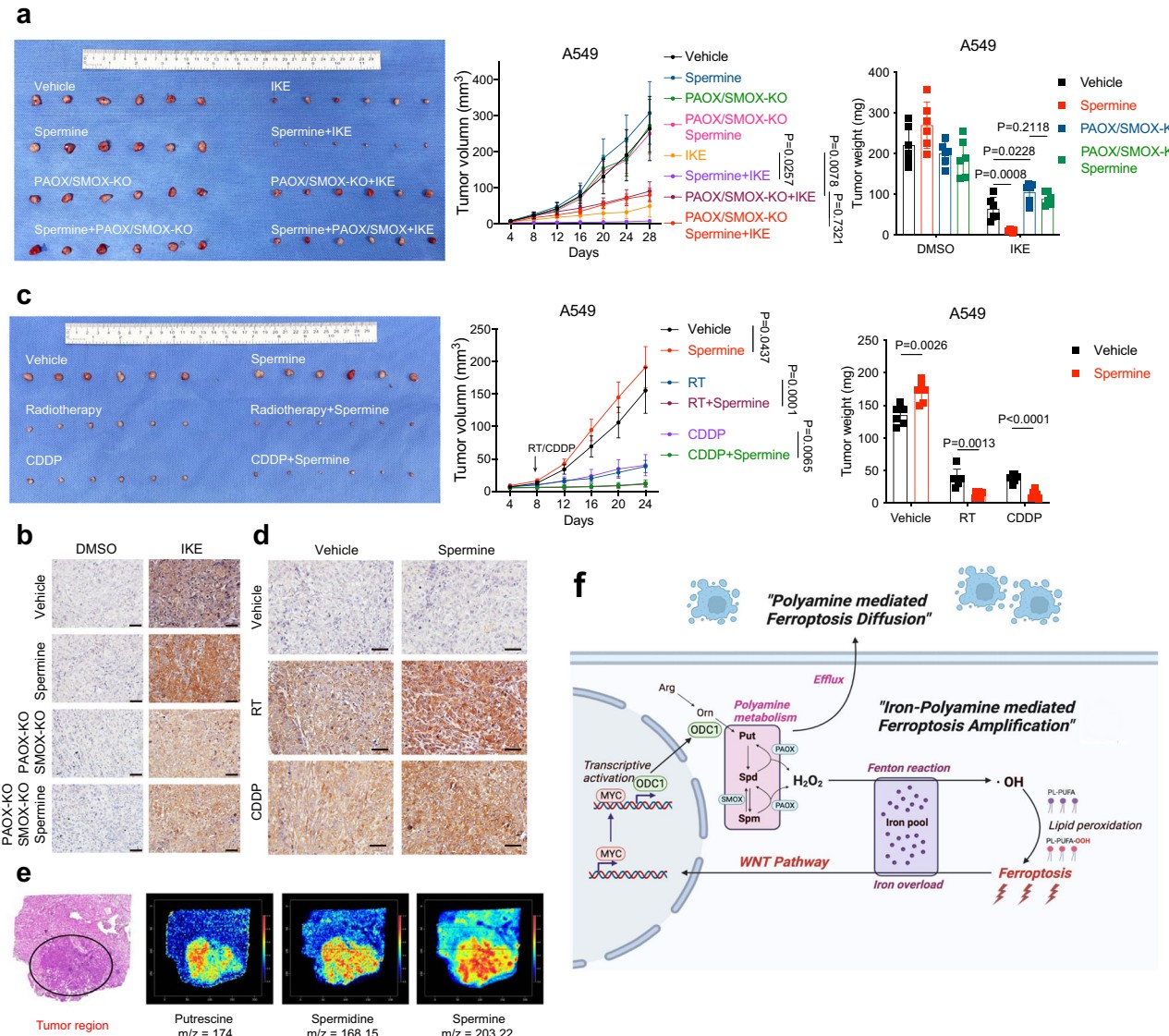

**Fig. 8 | Polyamine metabolism is a potential target of ferroptosis-based cancer treatment. a–d** Image of resected tumours from A549 mice xenografts. Groups of mice were treated as indicated (*n* = 6 per group). The growth of tumour volumes and the final weight of resected tumours were also shown (**a**, **c**). Representative immunohistochemical images of the resected tumours in each group (scale bars, 40 μm) (**b**, **d**). **e** The distribution of polyamines in the section of the surgically resected LUAD and adjacent normal tissue, determined by spatially resolved metabolomics. **f** A schematic model depicting the polyamine-mediated ferroptosis amplification and diffusion. Panel **f** is generated using Biorender. Data are presented as the mean ± SD. Unpaired two-tailed Student's *t* tests are used. Source data are provided as a Source Data file.

ferroptosis[49,50]. Aside from the well-known role of cysteine as a precursor of GSH, methionine also participates in the synthesis of GSH through the transsulfuration pathway upon extracellular cysteine limitation[51], while the tryptophan metabolites serotonin (5-HT) and 3-hydroxyanthranilic acid (3-HA) act as a potent radical trapping antioxidant to eliminate ferroptotic lipid peroxidation[52]. As one of the most versatile non-essential amino acids, arginine not only serves as a key metabolic intermediate of the urea cycle responsible for eliminating of excessive endogenous ammonia but also helps in cell division, wound healing, and immune response through its downstream metabolic products[11]. In recent years, arginine starvation has been well implicated in cancer therapy, whereas its involvement in ferroptosis and the corresponding mechanism remains enigmatic[53]. Although Conlon et al. have mentioned the protective effect of arginine deprivation in ferroptosis, the mechanism of this property remains unknown. Expected mTOR signalling, GCN2/ATF4-transsulfuration pathway, arginine's physical property, or shared metabolic roles with other ferroptosis-associated amino acids failed to explain this phenomenon[49]. In contrast, some researchers reported the antioxidative role of arginine derived from its participation in the production of nitric oxide (NO) and NRF2 pathway[54], but we obtained opposite results in our study. Through metabolic tracing analysis, we investigated the fate of $^{15}N_4$-arginine and confirmed that the arginine-ornithine-polyamine axis emerges as the dominant metabolic pathway of arginine. Since the polyamines exhibited potent pro-ferroptotic effect, and inhibiting the conversion of arginine to ornithine or polyamines diminished its ferroptotic mediating role, it is conceivable that the pro-ferroptotic property of arginine is primarily achieved through its conversion to polyamines.

Generally, the physical level of arginine in serum fluctuates between 50 and 150 μM, while the intracellular levels are in the range of 100–800 μM[55–57]. Notably, arginine deficiency is a hallmark of the tumour microenvironment[58]. Under such starvation, the anti-tumour responses of T cells are significantly blunted, including reduced CD3ζ

chain and cytokines, inhibited proliferation, and enhanced apoptosis[58–60]. This phenomenon is largely attributed to the arginine-consuming effect of arginase-expressing immunosuppressive myeloid cells such as M2 macrophage and myeloid-derived suppressor cells[55]. Moreover, the arginase level in these cells could be further enhanced by radiotherapy[61]. Therefore, is it possible that exogenously supplemented arginine could simultaneously promote tumour cell ferroptosis through the arginine-polyamine axis and enhance anti-tumour immunity? Does arginine's ferroptosis-promoting effect also apply to immune cells? The dynamic metabolic pattern of arginine in the tumour environment and its correlation with patients' response to ferroptosis-based therapy still warrants further investigation.

Consistent with our observation, Zhang et al. also noticed a strong correlation between polyamine metabolic activity and cells' sensitivity to cysteine starvation, a common cause of ferroptosis[62]. The inspiration of this study came from methionine, another precursor of polyamine. Generally, methionine is considered an anti-ferroptotic amino acid for its involvement in the transsulfuration pathway. However, in certain cells, methionine instead enters polyamine synthesis (Fig. 1f), thereby "shunting" the ferroptosis-protective phenotype to a promoting one. Our study focuses on the arginine-ornithine-polyamine axis and systematically elaborates the role of polyamine metabolism in ferroptosis. Mechanically, the oxidation of polyamine catalysed by PAOX and SMOX led to the production of $H_2O_2$, a key substrate of the Fenton reaction and subsequent lipid peroxidation. Genetically depleting these two oxidases or eliminating $H_2O_2$ through ectopically expressed catalase abolished polyamine's pro-ferroptotic property. Supportively, Ou et al. have identified *SAT1* (spermidine/spermine N'-acetyltransferase 1) gene as a transcription target of p53 and an inducer of ferroptotic lipid peroxidation, since SAT1 serves as the upstream enzyme of PAOX. Thus, the tumour-suppressive function of p53 seems to be in part mediated by polyamine-mediated ferroptotic sensitisation[63].

Using RNA-Seq and LC-MS, we noticed upregulated ODC1 expression and polyamine synthesis in RSL3-treated cells, suggesting a ferroptosis-ODC1-polyamine-ferroptosis positive-feedback loop. Zhang et al. have proposed a similar mechanism that PKCβII phosphorylates ACSL4 to amplify lipid peroxidation to induce ferroptosis. The activation of PKCβII in ferroptosis process could be completely inhibited by lipid peroxidation scavengers Fer-1 and liproxstatin-1, implying that PKCβII might function as a sensor of lipid peroxidation[64]. Supportively, it has been proposed that the metabolites resulting from the breakdown of peroxidized lipids have significant signalling capacity[65]. For instance, 4-hydroxynonenal (4-HNE), a specific final product of lipid peroxidation, stimulates PROM2 transcription through the p38-HSP pathway[66]. In contrast, 4-HNE accumulation also led to selective activation of NOX1, a membrane enzyme that produces ROS to initiate lipid peroxidation, thereby forming another positive-feedback loop that accelerates ferroptosis[27,67]. However, the signalling capacity of another key event in the ferroptosis process—iron overload—has not received enough attention yet.

Ferrous, also known as $Fe^{2+}$, is maintained in cells in the form of the liable iron pool, which bounds to low molecular weight compounds such as GSH[68]. Increased cellular liable iron is usually observed during the induction of ferroptosis initiated by either SLC7A11 inhibitor erastin or GPX4 inhibitor RSL3[25–27,69,70]. There are several explanations for this phenomenon. On the one hand, erastin-induced GSH depletion releases the bound $Fe^{2+}$ and mobiles them for Fenton reaction and subsequent ferroptosis[68]. On the other hand, accumulating data suggested that autophagic signals are generated in response to ferroptotic stress and promote the degradation of cellular iron storage protein ferritin, thus leading to iron accumulation[69,70]. This process is termed as "ferritinophagy". However, the exact mechanism of how ferroptosis triggers ferritinophagy still needs further exploration.

Our findings demonstrated the iron chelator DFO, but not Fer-1, could completely reverse ferroptosis-induced ODC1 upregulation, whereas FAC supplementation stimulated ODC1 expression just as RSL3 did, indicating the involvement of iron overload in the regulation of ODC1. Although Lane et al. have revealed the coupling of the polyamine and iron metabolism, the underlying mechanism of this crosstalk remains unclear[71]. Recent work has suggested that iron not only supports the function of several vital iron- and haem-containing enzymes as co-factor but also regulates a series of crucial pathways in tumours[30,72]. For instance, Chen et al. have proposed an interesting phenomenon that ferroptosis-induced iron accumulation promotes the formation of ALDH1B1- EIF4E complex, leading to ALDH1B1-degradation and ferroptosis sensitisation[27]. Among them, WNT signalling emerges as one of the best-known pathways impacted by iron. It has been implicated in multiple tumour types that elevated intracellular iron augments WNT signalling. Meanwhile, two independent groups identified iron chelators as top hits in high-throughput screens for WNT inhibitors[31,32]. Considering that ODC1 is a transcription target of MYC, a downstream factor of WNT signalling, and WNT activation sensitised cells to ferroptosis, we comprehensively elucidated the critical role of iron-WNT/MYC-ODC1 axis in the ferroptotic positive-feedback loop mentioned above. However, in our results, blocking WNT signalling only partially mitigated RSL3 or FAC-induced ODC1 upregulating, suggesting the existence of other alternative pathways mediating this effect. Besides, previous researchers also implied ferroptosis-inhibiting property of WNT signalling in gastric cancer and osteoblast differentiation[73,74], indicating a context-dependent role of this pathway in the regulation of ferroptosis. Besides, Fer-1 also slightly abrogates RSL3-induced ODC1 upregulation. On the one hand, although Fer-1 does not directly chelate iron as DFO does, the level of RSL3-induced iron accumulation might also be slightly suppressed by the absence of lipid peroxidation and ferroptotic cell death in Fer-1 group. On the other hand, we cannot exclude the existence of other unknown mechanisms accounting for this phenomenon.

It has been revealed that dying cells release a set of signals to impact surrounding cells, called "Ghost Message", and different cell death types generate profoundly different effects[75,76]. For instance, apoptotic cells synthesise and release several metabolites, including spermidine, through a particular protein channel, PANX1, thereby recruiting macrophages to express genes involved in tissue repair and inhibition of inflammation[77]. As for necrotic cells, they release damage-associated molecular patterns (DAMPs) like urea and drive inflammation[78]. Ferroptotic cells also release a set of signals to attract immune cells to the site of ferroptotically dying cells[45]. Brown's research has proposed an interesting model in which pro-ferroptotic stimuli induce the expression of prominin 2 that promotes the formation of ferritin-containing multivesicular bodies and exosomes, thereby facilitating exosomal transport of ferritin out of the cell and leading to ferroptotic resistance[66]. This effect could be considered a cell's self-productive mechanism against ferroptotic cell death. Besides, current modes of mammalian polyamine transport agree that polyamines exist in PSVs, from which they can be released into cytosol or secreted[21]. Inspired by this evidence, we investigated the abundance of extracellular polyamines after RSL3 treatment and found that ferroptotic cells tended to upregulate the secretion of excessive polyamine into the microenvironment through vesicles. We speculated that this process might also to some extent retard the polyamine-centred ferroptotic positive-feedback loop by expelling overproduced polyamine out of the cell. On the contrary, the exported polyamine further accelerated the diffusion of ferroptosis inside the tumour region. Previous studies have suggested that ferroptosis can spread through cell populations as a rapidly propagating wave, resulting in a distinct spatiotemporal pattern of cell death. This phenomenon involves the spreading of a cell swelling effect through cell populations in a lipid peroxide- and iron-dependent manner[79–81].

Together with these two ferroptotic executors, the released poly-amines from ferroptotic cells further "ignite" the spread of ferroptotic cell death. Therefore, polyamine metabolism contributes to a "dual circulation system" consisting of both intracellular ferroptosis-iron overload-WNT/MYC/ODC1-polyamine-$H_2O_2$ loop and extracellular released polyamine induced ferroptosis diffusion, suggesting the key modulating role of polyamine in the regulation and execution of fer-roptosis. Notably, considering that radio- or chemotherapy induces both apoptosis and ferroptosis, theoretically, both of which lead to spermidine synthesis and release, although through distinct pathways, it would be an interesting topic for future investigation whether these two pathways would interfere with each other and which one would have the dominating effect in different disease contexts. Besides, polyamines have been implicated to contribute to the formation of an immunosuppressive phenotype in the tumour microenvironment. Given that immune cells possess distinct responses to ferroptosis, it is worth further exploration of whether ferroptosis-induced polyamine might alter the composition of the tumour microenvironment in the presence of external ferroptotic stimuli.

Polyamines are essential for many fundamental processes of cell growth and survival[15]. In cancer, polyamine metabolism is frequently dysregulated, and multiple oncogenic pathways could result in upre-gulated polyamine synthesis, such as MYC, RAS-RAF-MEK-ERK, and PTEN-PI3K-mTORC1 signalling. Currently, most researchers believe that elevated polyamine levels are necessary for transformation and tumour progression, especially through the polyamine-eIF5A-hypusination axis[20,82]. Moreover, as illustrated above, polyamines are also associated with impaired anti-tumour immune response. There-fore, inhibiting the polyamine metabolic pathway is regarded as a target for therapeutic intervention[83]. Nevertheless, none of these strategies has been implemented in clinical use now. Using scRNA-seq and spatially resolved metabolomics, we confirmed the elevation of ODC1 and polyamine abundance in LUAD tissue. As reviewed by Lei et al., the oncogenic mutants-induced metabolic reprogramming often exposes new metabolic liabilities in cancer cells, especially, rendering some of them vulnerable to ferroptosis[3]. For example, cancer cells in a mesenchymal state, which are usually resistant to apoptosis induced by conventional therapies, are characterised by enhanced PUFA-PL synthesis and strong dependence on GPX4, consequently, these cells are ideal targets for ferroptosis inducers[84]. Therefore, as a potent fer-roptosis promoter, the key element of the positive-feedback loop, elevated polyamine might serve as cancer cells' Achilles heel that makes them susceptible to ferroptosis-based treatment. Specifically, spermine, the strongest ferroptosis promotor in the three polyamine types, which could be obtained from animal-derived foods[85], could be a promising candidate for tumour radio- and chemo-sensitisation.

In summary, through the identification of arginine as a ferroptosis mediator, our study elucidates the potent pro-ferroptotic property of polyamines and the corresponding mechanism, as well as the iron overload-WNT/MYC/ODC1-polyamine-$H_2O_2$ positive-feedback axis that amplifies ferroptosis. Our findings also present a previously unappreciated metabolic susceptibility for polyamine-overloaded tumours that may offer potential targets and strategies for ferroptosis-associated cancer therapy.

## Methods

The use of patient samples and animal models in this research com-plies with all relevant ethical regulations of Zhongshan Hospital Fudan University, including the Institutional Review Board and Institutional Animal Care and Use Committee.

### Cell lines and culture condition

All cells were cultured in a 37 °C incubator with humidified 5% $CO_2$ atmosphere. The human non-small cell lung cancer cell lines A549 (TCHu150), H1299 (TCHu160), PC9 (SCSP-5085), H23 (SCSP-581),

fibrosarcoma cell line HT1080 (TCHu170), and human embryonic kidney cell line HEK-293T (GNHu17) were purchased from the Chinese Academy of Science Cell Bank and cultured in high-glucose Dulbecco's modified Eagle's medium (DMEM, Hyclone, USA) supplemented with 10% foetal bovine serum (ScienCell, USA) and 100 U/mL penicillin/streptomycin/amphotericin B (Sangon Biotech, China). Low passage cells (<30 passages) were used for all experiments. The cell lines were authenticated by short tandem repeat profiling in 2023 and were passaged every 3–5 days according to cells' proliferating rates. To prevent Mycoplasma contamination, cells were tested every 2 months using a PCR-based method. Briefly, supernatant medium without any antibiotics was collected after 7 days cell culturing, from which the DNA was extracted and purified using silica-gel columns (TIANGEN, China). Then, PCR assays were performed with hot-start Taq DNA polymerase according to the protocol provided by the manufacturer. PCR products were separated on a 1.3% agarose-TAE gel (Sangon Bio-tech) containing 0.3 mg/mL ethidium bromide and the results were visualised with a UV transilluminator. If the tested medium contained Mycoplasma, an additional band at 515–525 bp could be observed. Only the Mycoplasma-negative cells are kept for subsequent experiments.

### Human LUAD samples

The tumour and corresponding adjacent normal tissues were obtained from the patients diagnosed with LUAD who received surgery in the Department of Thoracic Surgery, Zhongshan Hospital, Fudan Uni-versity, between May 2021 and Mar 2023 (female 6, male 4). The use of patient samples in this study was approved by the Institutional Review Board of Zhongshan Hospital, Fudan University, China (Approval No: B2022-180R). All of the patients have provided written informed con-sent, and the study was conducted in accordance with the Declaration of Helsinki. After quick-freezing with liquid nitrogen, the samples were stored at −80 °C before processing. Frozen human tissues (6 pairs, female 4, male 2) were used for protein extraction-western blotting and IHC staining (4 pairs, female 2, male 2). The detailed demographic data of LUAD patients (female 13, male 13) included in scRNA-Seq could be found in our previous publication[44].

### Compounds

The following compounds were obtained from TargetMol (USA): RSL3 (T3646), IKE (T5523), ferrostatin-1 (T6500), deferoxamine mesylate (T1637), Z-VAD (OMe)-FMK (T6013), necrosulfonamide (T7129), FIN56 (T4066), FINO$_2$ (T60084), LF3 (T7399), SKL2001 (T6989), cisplatin (T1564), aminoguanidine hydrochloride (T0358). The following com-pounds were obtained from MedChemExpress (MCE, USA): AMXT-1501 (HY-124617A), DFMO (HY-B0744), GC7 (HY-108314A). The fol-lowing compounds were obtained from Sangon Biotech (China): L-arginine (A600205), L-citrulline (A604057L), L-proline (A600923), putrescine dihydrochloride (A600793). The following compounds were obtained from Aladdin (China): L-ornithine monohydrochloride (O108803), urea (A610148), spermidine (S107071), spermine (S109704), ammonium ferric citrate (A100170).

### Metabolite library screening

The human endogenous metabolite library containing 889 com-pounds was purchased from MCE (HY-L030). A549 cells were plated in 96-well plates at 5000 cells per well. On the second day, the cells were pre-incubated with indicated metabolites or DMSO (50 μM). Twelve hours later, the cells were treated with 4 μM RSL3 for 8 h. The cell viability was measured using CCK8 and data were collected.

### Cell viability assay

Cell viability was determined using Cell Counting Kit-8 (CCK8, Topscience), as previously described[6]. Briefly, 5000 cells (for cyto-toxicity assay) per well were seeded in 96-well plates and incubated for

24 h. Subsequently, cells were treated as indicated and exposed to 10 μl of CCK8 reagent (100 μl medium per well) for 1 h at 37 °C, 5% CO₂. The absorbance at a wavelength of 450 nm was measured using a Molecular Devices microplate reader (Biotek, USA). Since the cells were cultured in DMEM medium containing 400 μM arginine, we adopted a concentration gradient of 0-100-200-400-800 μM exogenous arginine in ferroptosis-related cell viability assay and finally chose 500 μM. At such concentration, the supplemented arginine did not impact cells' viability or proliferation.

### Measurement of lipid peroxidation by BODIPY-C11
The determination of lipid peroxidation was performed as previously described[6]. Briefly, cells were seeded on 12-well plates and incubated overnight. The next day, cells were treated as indicated, harvested by trypsinization and washed with phosphate-buffered saline (PBS, Beyotime, China). Next, the cells were suspended in a fresh medium containing 4 μM BODIPY 581/591 C11 dye (Thermo Fisher, USA) at 37 °C in a humidified 5% CO₂ atmosphere. After 30 min of incubation, the cells were washed with PBS, and the lipid peroxidation levels were assessed using the flow cytometer Accuri C6 (BD Bioscience, USA) with a 488-nm laser. The results were analysed in FlowJo software (TreeStar, Woodburn, OR, USA). A figure exemplifying the gating strategy is provided in Supplementary Fig. 4n.

### Detection of amino acid and polyamine abundance by metabolomic analysis
**Sample preparation.** For cell samples ($n = 3$ for both experimental and control groups), after being washed with ice-cold PBS, the cells were collected using a cell scraper and centrifuged for 5 min at $1000 \times g$. The medium samples were directly collected from the plates. The samples were homogenised in liquid N₂ for 1 min and defrosted in 4 °C. For amino acid level determination, 600 μL extraction solution (60:40, v/v, acetonitrile/water) was added, and the samples were lysed by 5 min vortex and 5 min grind, followed by sonication in ultrasonic water bath for 30 min at 4 °C and stand for 30 min at 4 °C. Then, the samples were centrifuged at $17,000 \times g$ for 10 min at 4 °C, and the supernatant was collected for subsequent LC-MS analysis. For polyamine level determination, extraction solution containing 300 μL acetonitrile and 0.1 g NaCl was added, and the samples were vortexed for 5 min, followed by centrifugation at $17,000 \times g$ for 15 min at 4 °C. Supernatant was collected, purified using C18/PSA sorbent, filtered through 0.22-μm membrane, and then subjected to subsequent LC-MS analysis.

**LC-MS conditions.** The liquid chromatogram (LC) separation of the compounds was carried out on a Acquity UPLC BEN Amide (amino acid detection: 1.7 μm, 2.1 mm × 100 mm; polyamine detection: 1.8 μm, 2.1 mm × 100 mm) (Waters Corporation, USA). For amino acid detection, the mobile phase, which consisted of 0.1% formic acid in water (A) and acetonitrile (B), was delivered at a flow rate of 0.3 mL/min under a gradient programme. The column temperature was 40 °C, and the injection volume was 5 μL. The gradient system was 0–1 min, 90% B; 1–2.5 min, 90–35% B; 2.5–4.0 min, 35% B; 4.0–4.1 min, 35–90% B, 4.1–5 min, 90% B. The mass spectra (MS) were acquired using an AB SCIEX 5500 Qtrap-MS system. MS parameters were set as follows: curtain gas, 35 arb (arbitrary units); collision gas, 9 arb; IonSpray voltage, 4000 V; IonSource temperature, 400 °C; IonSource gas1, 55 arb; and IonSource gas2, 55 arb. For polyamine detection, the mobile phase, which consisted of 0.2% formic acid in water (A) and acetonitrile (B), was delivered at a flow rate of 0.3 mL/min under a gradient programme. The column temperature was 30 °C, and the injection volume was 4 μL. The gradient system was 0–1 min, 10% B; 1–3.0 min, 10–90% B; 3.0–4.0 min, 90% B; 4.0–4.1 min, 90–10% B, 4.1–5 min, 10% B. MS parameters were set as follows: curtain gas, 35 arb (arbitrary units); collision gas, 9 arb; IonSpray voltage, 4500 V; IonSource temperature,

450 °C; IonSource gas1, 55 arb; and IonSource gas2, 55 arb. According to the conditions described above, the prepared standard solution was added to the sample vial to quantify metabolite levels in the samples. The data were analysed by MultiQuant Software (SCIEX, Foster City, CA, USA). All detections were performed by Sensichip Biotech Company (Shanghai, China).

### Metabolic profiling and tracing studies
**Sample preparation.** A549 cells were seeded into 9-cm plates in complete medium and were allowed to adhere overnight. Cells were washed with PBS, and the arginine-free DMEM medium supplemented with 84 mg/L ¹⁵N₄-labeld arginine hydrochloride (Shanghai ZZBio Co. Ltd., China) containing 0.5 μM RSL3 ($n = 3$) or vehicle (DMSO, $n = 3$) were added. After 24-h of incubation, the medium was discarded and the cells were washed with ice-cold PBS, and collected using cell scrapers. After centrifugation, the cell samples were homogenised in liquid N₂ for 1 min and defrosted in 4 °C. In total, 200 μL 80% methyl alcohol was added, followed by vortex for 1 min, sonication for 30 min at 4 °C, and stand for 1 h at −20 °C. Then, the samples were centrifuged at $17,000 \times g$ for 15 min at 4 °C, and the supernatant was collected for subsequent LC-MS analysis. 2.5 μL 1 mg/mL 2-amino-3-(2-chloro-phenyl)-propionic acid was added as internal standard.

**LC-MS conditions.** The LC separation of the compounds was carried out on an Acquity UPLC BEN Amide (1.7 μm, 2.1 mm × 100 mm). The mobile phase, which consisted of 10 mM ammonium formate and 0.05% ammonia in 90:10 v/v acetonitrile/water (A) and 10:90 v/v acetonitrile/water (B), was delivered at a flow rate of 0.3 mL/min under a gradient programme. The column temperature was 40 °C, and the injection volume was 5 μL. The gradient system was 0–1 min, 100% B; 1–12.5 min, 100–55% B; 12.5–14.5 min, 55% B; 14.5–14.6 min, 55–100% B, 14.6–16 min, 100% B. The MS was acquired using an AB SCIEX 5500 Qtrap-MS system. MS parameters were set as follows: ESI +: heater temp 300 °C; sheath gas flow rate, 45 arb; aux gas flow rate, 15 arb; sweep gas flow rate, 1arb; spray voltage, 3.0 KV; capillary temp, 350 °C; S-Lens RF level, 30%; ESI-: heater temp 300 °C, sheath gas flow rate,45 arb; aux gas flow rate, 15 arb; sweep gas flow rate, 1arb; spray voltage, 3.2 KV; capillary temp, 350 °C; S-Lens RF level, 60%. Peak identities were confirmed matching masses and retention times to authentic standards. The data were analysed using Compound Discover 3.1 software (Thermo Fisher Scientific, USA). All detections were performed by Sensichip Biotech Company (Shanghai, China).

### Transmission electron microscopy
The samples for transmission electron microscopy were prepared as previously described[6]. Briefly, treated cells cultured in 6 cm dishes were fixed with a solution containing 2.5% glutaraldehyde. After being washed in 0.1 M phosphate buffer (pH 7.4), cells were postfixed with phosphate buffer containing 1% osmic acid, then washed in 0.1 M phosphate buffer (pH 7.4) for three more times. After dehydration and embedding, samples were incubated in a 60 °C drying oven for 48 h. Ultrathin sections were prepared and stained with lead citrate and uranyl acetate. After drying overnight, the sections were examined with a Hitachi transmission electron microscope (Japan).

### CRISPR/Cas9-mediated gene knockout
CRISPR/Cas9 technology was employed to knock out target genes in the cell lines. sgRNAs were cloned into the GV392 plasmids containing puromycin or blasticidin resistance gene and hSpCas9 gene. The following sgRNA sequences were used: sgARG2-#1, 5′-GCCAGCTTCTCTTATGGCAG-3′; sgARG2-#2, 5′-GGCTCTCCAGTTTGGGTAAG-3′; sgODC1-#1, 5′-GATTGTCACTGCTGTTCCAA-3′; sgODC1-#2, 5′- GTTCCAAGGGCACACGCAGA-3′; sgPAOX: 5′-GACCCCGGTGCCCAGCGTCG-3′; sgSMOX: 5′-GTCATCCGCACTGTCACCAC-3′; sgRNA-Control: 5′-CGCTTCCGCGGCCCGTTCAA-3′. Then, the edited vectors were packaged into

lentiviruses in HEK-293T cells. The design and construction of the vectors and lentiviruses were performed by Genechem Technology (Shanghai, China).

## Immunoblotting

Western blotting was performed as previously described[6]. Briefly, cells were lysed using RIPA buffer (Beyotime) with protease and phosphatase inhibitor cocktail (Beyotime), and the proteins were boiled in 5× SDS-PAGE loading buffer (EpiZyme Biotech, China) for 10 min at 100 °C. About 15–20 μg proteins were separated using SDS-PAGE (EpiZyme Biotech) and transferred onto polyvinylidene fluoride membranes (Merck-Millipore, USA). The membranes were blocked with 5% nonfat milk and incubated overnight at 4 °C with antibodies against ARG2 (1:1000, A19233, Abclonal, USA), ODC1 (1:1000, A3898, Abclonal), GPX4 (1:1000, DF6701, Affinity, USA), ACSL4 (1:1000, abs106075, Absin, China), SLC7A11 (1:1000, DF12509, Affinity), GAPDH (1:2000, AF0006, Beyotime), PAOX (1:1000, abs139256, Absin), SMOX (1:1000, abs151305, Absin), catalase (1:1000, A18018, abclonal), c-Myc (1:1000, A1309, Abclonal), TSG101 (1:1000, A1692, Abclonal), CD63 (1:500, abs149061, Absin), GM130 (1:2000, A5344, Abclonal), CD81 (1:1000, A4863, Abclonal). After the membranes were washed with Tris-buffered saline-Tween solution, the secondary antibodies were added to the membranes at room temperature. Finally, the protein bands were visualised with a BeyoECL Plus kit (Beyotime).

## Reduced GSH measurement

The reduced GSH level was determined as previously described[6]. The treated cells were harvested by trypsinization, transferred to new tubes, washed in PBS, and centrifuged at $1500 \times g$ at 4 °C for 5 min. The cell pellets were resuspended in 60 μL protein removal solution, thoroughly mixed, and incubated at −196 °C (liquid nitrogen) and 37 °C sequentially twice for fast freezing and thawing, then incubated at 4 °C for 5 min and centrifuged at $10,000 \times g$ for 10 min. The supernatant was pipetted to determine the amount of GSH in the sample. This assay was conducted using the GSH and GSSG Assay Kit (Beyotime) according to the manufacturer's protocol.

## siRNA-mediated gene knockdown

siRNAs and corresponding negative controls, used in the presented research, were purchased from RiboBio (Guangzhou, China). The target sequences are listed as: siSAT1#1: CGACAAGGAGTACTTGCTA, siSAT1#2: GGATCAGAAATTCTGAAGA; siPAOX#1: GGGAGTACCTCA GAAGGA, siPAOX#2: GGTAGAGTGTGAGGATGGA; siSMOX#1: GCATCAGCCTCTATTCCAA, siSMOX#2: GGAACCCTATCTATCATCT; siAMD1#1: GGATTACAGTGGGTTTGACTCAATT, siAMD1#2: TACAGTG GGTTTGACTCAATTCAAA; siSRM#1: AGGAGTCCTATTACCAGCTCA TGAA, siSRM#2: CAGGATGCCTTCGACGTGATCATCA; siSMS#1: CACCTGGCAGGACCATGGCTATTTA, siSMS#2: CATGGCTATTTAGCA ACCTACACAA; siMYC#1: GTGCAGCCGTATTTCTACT; siMYC#2: GGAACTATGACCTCGACTA. The cells were seeded into six-well plates at 60%–80% confluence. The next day, the medium was replaced with fresh medium containing siRNAs and Lipo8000 (Beyotime) as the transfection reagent. The cells were harvested for subsequent analyses 48 h after transfection.

## Quantitative real-time PCR (qRT-PCR)

RNA extraction and qRT-PCR were performed as previously described[6]. Briefly, total RNA was extracted from cells using TRIzol reagent (TIANGEN), and cDNA was synthesised with a Hifair® II 1st Strand cDNA Synthesis Kit (gDNA digester plus, Yensen Biotechnology, China). qPCR was performed with a Hifair® III One Step RT-qPCR SYBR Green Kit (Yensen Biotechnology), and triplicate samples were run on an ABI QuantStudio 5 Real-Time PCR System (Thermo Fisher, USA). The threshold cycle (Ct) values for each gene were normalised to GAPDH as the endogenous control, and the 2-ΔΔCt method was used

for quantitative analysis. The primers used were synthesised by Sangon Biotech. GAPDH: F: AGAAGGCTGGGGCTCATTTG, R: AGGGGCC ATCCACAGTCTTC; ODC1: F: GGCTGTACCGATCCTGAGACCTT, R: GCCACCGCCAATATCAAGCAGAT; AMD1: F: AGAAGCAGCAACAACA GCAGAGT, R: ACAGCAAGAGTGGCAGAGAATACC; SRM: F: GATGAT CGCCAACCTGCCTCTC, R: TCTCACACTGGACCACGGACTC; SMS: F: GGATTGGTGTTGCTGGACCTTCA, R: ATGGCTCCTCCTCGCACTATGG; SAT: F: CGGAAGGACACAGCATTGTTGGT, R: ACTGGACAGATCAGAA GCACCTCT; SMOX: F: CGCTCGCCGCAGACTTACTT, R: CCGCACTGTC ACCACTGGATTC; PAOX: F: TTCCAGTGTCGGTAGAGTGTGAG, R: TC TTCCTGATTGCTTCTGCCTTCT; MYC: F: GGCTCCTGGCAAAAGGTCA, R: CTGCGTAGTTGTGCTGATGT.

## Measurement of intracellular $H_2O_2$ levels by H2DCFDA

Intracellular $H_2O_2$ levels were assessed using the H2DCFDA ROS Assay Kit (Beyotime) according to the manufacturer's protocol. Briefly, cells seeded in six-well plates were treated as indicated, then cells were incubated with serum-free medium containing 10 μM H2DCFDA probes for 20 min at 37 °C incubator with 5% $CO_2$. Cells were washed with serum-free medium for three times to deplete uncombined probes before data were acquired by excitation of the probes at 488 nm, and emission was measured at 525 nm using a microplate reader.

## Measurement of intracellular iron levels by FerroOrange

Intracellular ferrous ion was assessed using FerroOrange (Dojindo, Japan) according to the manufacturer's instruction. Briefly, cells seeded in confocal dishes were stained with HBSS buffer containing 1 μM FerroOrange regent for 30 min at 37 °C incubator with 5% $CO_2$. Fluorescence images were captured by the Laser Scanning Confocal Microscope FV3000 (Olympus, Japan).

## Chromatin immunoprecipitation (CHIP) assay

The assay was conducted using the SimpleChIP Plus Enzymatic Chromatin IP Kit (Cell Signalling Technology, USA) according to the manufacturer's protocol. Briefly, cells were fixed with formaldehyde to cross-link histone and non-histone proteins to DNA. Then, chromatin was digested with Micrococcal Nuclease into 150–900 bp DNA/protein fragments. IgG or antibody specific to c-Myc (1:50, #9402, Cell Signalling Technology) were added, the complex co-precipitated and was captured by Protein G magnetic beads. Next, the chromatin was eluted, and the cross-links were reversed. Finally, DNA was purified using spin columns and quantified by qRT-PCR. Primer 1: F: AAGTCCGCATCAC-CACAGAATCA, R: AGCCGTAAGCAGAAGTGAGTCTT; primer 2: F: GACTGTGCCTCAGACCTGGTT, R: ACAGCCGATGAAGAGTTAAGGA GA; primer 3: F: CCAATTCAAGTGCAGTGCCTCC, R: CGCTCTGAGAG TTACGGAAGTCC.

## Dual-luciferase reporter assay

ODC1 promoter region spanning from −2000 to +200 of the transcription start site, and corresponding mutant sequences were cloned into PHY-811 vectors. The assay was conducted in the HEK-293T cell line using a Luciferase Reporter Gene Assay Kit (Beyotime) as previously described[6].

## Extracellular vesicle (EV) isolation

Cells were cultured in medium containing EV-free serum and treated with DMSO or RSL3 for 24 h. Supernatants were collected, and debris and dead cells were discarded by centrifugation at $1000 \times g$ for 10 min at 4 °C followed by filtration through a 0.2 um filter. Then, the medium was further centrifuged at 4 °C at $4500 \times g$ for 10 min, $5500 \times g$ for 30 min prior to ultracentrifugation at $31,000 \times g$ for 140 min. Pellets were resuspended in cold PBS and centrifuged again with the same protocol. Finally, the EVs-containing pellets were lysed in RIPA buffer and the proteins were extracted as described above.

## Irradiation and clone formation assay

The assay was conducted as previously described[86]. Briefly, 800 cells per well were seeded in triplicates into 6-well plates and allowed to grow for 24 h. Then, cells were pre-treated with polyamines or PBS for 5 h, followed by irradiation with a gradient dose using an ONCOR™ linear accelerator. After 10 days of incubation in fresh medium, the cells were fixed with 4% methanol and stained with 1% purple crystal. The colonies in each well were counted visually.

## Tumour xenograft experiment

All animal studies were conducted in compliance with the policies of the Institutional Animal Care and Use Committee of Zhongshan Hospital, Fudan University. Four-week-old male homozygous (Foxn1$^{nu}$) mut/mut BALB/c nude mice were purchased from the GemPharmatech (Nanjing, China) and housed under specific-pathogen-free conditions with a 12 h light–12 h dark cycle. The ambient temperature was 21–23 °C, with 45% humidity and the mice had ad libitum access to water and standard chow (LabDiet, #5053). In total, $2 \times 10^6$ Cas-NC or PAOX/SMOX-KO A549 cells were resuspended in 100 μL cold PBS and subcutaneously injected into the right flank of each nude mouse. When the tumours reached palpable, the mice were assigned randomly into eight treatment groups ($n = 6$ for each group at the beginning). Spermine was administered orally via drinking water (3 mM), while control animals received regular drinking water (changed twice a week)[87,88]. Prior to each exchange of the drinking bottles, the weight of the bottles was measured to calculate the average volume consumed per animal per day. For IKE or CDDP treatment, IKE (30 mg/kg) dissolved in DMSO/corn oil (Beyotime) and CDDP (3 mg/kg) dissolved in PBS were intraperitoneally injected every two days. The drug administration was continued until the endpoint, as indicated in the corresponding figures. For radiotherapy, ionising radiation (X-ray) was applied locally to the tumour in the flank of mice at 8 Gy. The tumour volume was measured using a caliper every four days until the endpoint and calculated according to the equation: v = length * width$^2$ * 1/2. Mice that were killed before the end of the experiment were not included in the final analysis. The xenograft tumours were harvested for subsequent immunohistochemistry at the end of the experiment. The maximal tumour burden permitted by the ethics committee is a length of 1.5 cm, and the maximal tumour burden did not exceed the limit.

## Immunohistochemistry

Tissue specimens were obtained from the tumour xenograft experiment mentioned above. The paraffin-embedded tissues were dewaxed, rehydrated, and stained using the Immunohistochemistry kit according to the manufacturer's protocol (SP-9002, ZSGB-BIO, China). The antibody used for IHC was anti-4-HNE (1:200, Abcam, ab46545) and ODC1 (1:100, A3898, Abclonal).

## Spatially resolved metabolomics

**Sample preparation.** The embedded LUAD samples were stored at −80 °C before being sectioned. The samples were cut into consecutive sagittal slices 10 μm by a cryostat microtome (Leica CM 1950, Leica Microsystem, Germany) and were thaw-mounted on positive charge desorption plate (Thermo Scientific, USA). Sections were stored at −80 °C before further analysis. They were desiccated at −20 °C for 1 h and then at room temperature for 2 h before mass spectrometry imaging (MSI) analysis. Meanwhile, an adjacent slice was left for H&E staining.

**Data acquisition and MSI analysis.** The analyses were performed as previously reported[89]. In brief, this experiment was carried out with an AFADESI-MSI platform (Beijing Victor Technology Co., LTD, Beijing, China) in tandem with a Q-Orbitrap mass spectrometer (Q Exactive, Thermo Scientific, USA). Here, the solvent formula was acetonitrile/ H$_2$O (8:2) at negative mode and acetonitrile/H$_2$O (8:2, 0.1% FA) at

positive mode and the solvent flow rate was 5 μL/min, the transporting gas flow rate was 45 L/min, the spray voltage was set at 7 kV, and the distance between the sample surface and the sprayer was 3 mm as was the distance from the sprayer to the ion transporting tube. The MS resolution was set at 70,000, the mass range was 70–1000 Da, the automated gain control (AGC) target was 2E6, the maximum injection time was set to 200 ms, the S-lens voltage was 55 V, and the capillary temperature was 350 °C. The MSI experiment was carried out with a constant rate of 0.2 mm/s continuously scanning the surface of the sample section in the x direction and a 0.1-mm vertical step in the y direction.

**Data analysis.** The collected.raw files were converted into imML format using imzMLConverter and then imported into MSiReader for ion image reconstructions after background subtraction using the Cardinal software package. All MS images were normalised using total ion count normalisation (TIC) in each pixel. Region-specific MS profiles were precisely extracted by matching high-spatial resolution H&E images. The discriminating endogenous molecules of different tissue microregions were screened by a supervised statistical analytical method: orthogonal partial least squares discrimination analysis (OPLS-DA). Variable Importance of Projection (VIP) values obtained from the OPLS-DA model were used to rank the overall contribution of each variable to group discrimination. Differential metabolites were selected with VIP > 1.0 and $P$ values < 0.05. The ions detected by AFA-DESI were annotated by the pySM pipeline and an in-house SmetDB database (Lumingbio, Shanghai, China).

## Statistical analysis and reproducibility

All experiments were independently performed in at least three times. Unpaired two-tailed Student $t$ tests was utilised to analyse the statistical significance of differences between two groups using GraphPad Prism software (7.0) and Excel. Two-way analysis of variance (ANOVA) was used for grouped analysis. The results are presented as means, and the error bars represent the standard deviation. Bioinformatical analyses, including RNA-Seq and scRNA-Seq, were conducted using R software as previously described[6]. $P$ values < 0.05 were considered statistically significant. All cells and the animals were randomly allocated to experimental groups. The investigators were blinded to allocation during experiments and outcome assessment.

## Reporting summary

Further information on research design is available in the Nature Portfolio Reporting Summary linked to this article.

## Data availability

The TCGA and CTRP publicly available data used in this study are available at http://gepia.cancer-pku.cn/detail.php?gene=ODC1[43], and https://portals.broadinstitute.org/ctrp.v2.1/[19]. The ENCODE data used in this study is available at https://www.encodeproject.org/ experiments/ENCSR000DLR/[36]. The scRNA-Seq publicly available data used in this study are available in the ArrayExpress database [accession numbers E-MTAB-6149 and E-MTAB-6653] and Human Cell Atlas Data Coordination Platform database [accession number PRJEB31843][44]. The RNA-Seq data generated in this study have been deposited in the SRA database under accession code PRJNA979805. The metabolomics data are deposited in the Figshare database under accession code 24037836. The processed data are available in Source data file. The remaining data are available within the Article, Supplementary Information or Source Data file. Source data are provided with this paper.

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

## Acknowledgements

This research was supported by the Natural Science Foundation of Shanghai (No. 22ZR1411900, to C.Z.), the Special Foundation for Supporting Biomedical Technology of Shanghai (No. 22S11900300, to C.Z.), the National Natural Science Foundation of China (No. 82103311, to T.L.), the China Postdoctoral Science Foundation (No. 2022M710763, to T.L.), and the Outstanding Resident Clinical Postdoctoral Program of Zhongshan Hospital Affiliated to Fudan University (to G.B.).

## Author contributions

G.B. and C.Z., designed the research. G.B., J.L. and Y.B. carried out the experiments and analysed the data. G.S., Y.H., T.L., H.Z., X.J., Z.C. and M.Z. helped with the discussion and interpretation of results. G.B., J.L. and Y.B. wrote the manuscript under the supervision of H.F., Q.W., B.G. and C.Z. All authors have read and approved the manuscript.

## Competing interests

The authors declare no competing interests.
