## [Peer Review File · Nature Communications]

Reviewers' Comments:

Reviewer #1:

Remarks to the Author:

Cellular ferroptosis is an iron-dependent form of regulated cell death due to excessive iron-induced lipid oxidation and redox imbalance both of which play a fundamental role in tumor radiotherapy. This study by Bi et al reveals that arginine is one of the amino acids detected by screening the metabolites in RSL-3-induced ferroptosis in lung cancer A549 cells and adding arginine can enhance the RSL-3 mediated cytotoxicity. By downstream investigation, the polyamines that can be converted from Arginine are found to be the executor for RSL-3-mediated cytotoxicity which is identified by the activation of ODC1 that governs the polyamine synthesis. Based on these findings, a pathway of iron-overload induced ODC1-polyamine-H₂O₂ is suggested which is innovative to the field. Additional findings include the polyamine-containing extracellular vesicles in the tumor microenvironment leading to neighboring cell ferroptosis and polyamine-mediated ferroptosis in tumor sensitization to radiotherapy and chemotherapy.

The overall experimental design and data explanation are reasonable, and the results can fill up some gaps in understanding the mechanisms underlying ferroptosis-contributed tumor radiosensitivity. The evidence supporting the iron-overloading mediated metabolic feedforward loop is informative but could be affected by other cell death pathways which may be discussed. Although some details are in need which include the relevance of physiologic arginine metabolism and ferroptosis-inducing concentrations in tumor cells and potential risk in normal cells. However, the major experimental data are supportive to the conclusion. This work contributes to a new version of metabolism-associated ferroptosis and holds a potential clinical benefit.

Scientific Comments:

1. In the Abstract, a ferroptosis-iron overload-WNT/MYC-ODC1-polyamine-H₂O₂ positive feedback loop that amplifies ferroptosis level is raised but it is unclear how it is termed as a "positive feedback loop" since it does not show H₂O₂ can enhance iron loading.
2. In Figure 1 cartoon, a part showing how the metabolites were generated should be added.
3. Ferroptosis cell death marker may be added in the cell viability test since the arginine and other amino acids also induce other types of cell death
4. Line 101-103, "Screening of this library identified arginine, one of the most versatile non-essential amino acids involved in multiple biological processes¹¹, significantly promoted RSL3 induced ferroptosis (Figure 1b)". This description is not precise for the data shown in Figure 1b since the screening was conducted by RSL3 not Arginine-induced ferroptosis and there seem other metabolites located in the RSL3-sensitive cells.
5. RSL3-induced ferroptosis was shown to be enhanced by adding Arginine which is being increasingly studied for an array of physiological and pathological functions including innate and adaptive immune regulation. Thus a discussion on physical Arginine concentration and dynamics especially in tumor and irradiated tumor microenvironment would be appreciated.
6. A calculation of physiological basal cellular arginine concentration (such as essential diet supplements) with the RSL-3 sensitive cells and the concentration added to enhance RSL-3 ferroptosis will be informative.
7. Line 114-128, it is unclear if the arginine downstream metabolites that have been further studied in the following experiments were also detected in the RSL-3 cytotoxicity shown in Figure 1b.
8. The polyamines released from the ferroptotic cells may belong to the categories of Ghost Messages from different cell death including radiation-induced apoptosis. Thus, it could be highly informative to differentiate or discuss the polyamines as the ghost message from other types of cell death induced by radiotherapy (may refer: Ghostly metabolic messages from dying cells, D.R. Green).
9. Based on the history and challenges in metabolites-targeting approaches, it is appreciated if a specific metabolite in the proposed iron-polyamine loop in ferroptosis could be the top candidate for tumor radiosensitization or chemosensitization.

Reviewer #2:

Remarks to the Author:

This study found that arginine metabolites especially polyamine could activate RSL3 or IKE induced ferroptosis, which was activated by the initial ferroptosis signal—iron overload mediated WNT/MYC pathway, and the subsequent increased levels of Odc1, thus providing new therapeutic strategies in cancer treatment. The results were interesting, and the findings is important. But I have some concerns:

Lack of ferroptosis inhabitation experiments in some of important ferroptotic killing experiments. Please add the data.

The effect differences in some cell viability experiments look not that much. please find the optimum conditions.

Reviewer #3:

Remarks to the Author:

Emerging evidence has shown that certain cancers are vulnerable to ferroptosis, making this recently uncovered non-apoptotic cell death pathway a new target for cancer therapy. In the current study, Bi et al. identified Arginine as a ferroptosis promoter. Mechanistically, Arginine is converted into polyamine. The metabolism of polyamines produces H₂O₂, which increases cellular ROS level and lipid peroxidation via the Fenton reaction. They also noticed that the initiation of ferroptosis by RSL3 caused iron overload, which, through WNT signaling, increased the expression of a critical enzyme catalyzing polyamine synthesis called ODC1. In addition, cells under ferroptosis also enhanced EV production as a protective mechanism to reduce intracellular polyamine levels. However, the release of polyamine-containing EVs also sensitizes the surrounding cells to ferroptosis. Overall, most of the data presented in the current study are convincing and supportive of the conclusion. However, one of the key conclusions that polyamine enhances ferroptosis has been shown by other studies, which compromises the novelty of the current study. The ODC1-mediated feedback loop model is interesting, but additional evidence and clarification are needed.

Major points:

1. Polyamines sensitize cells to RSL3 and FIN56. How about FINO2, which directly oxidizes iron?
2. Where does the overloaded iron come from? Does GPX4 KO cause iron overload?
3. WNT has been reported to inhibit ferroptosis (PMID: 35534546). In this aforementioned study, LF3 treatment sensitized cells to Erastin-induced ferroptosis but not H₂O₂-induced apoptosis, which contradicts what the current study showed.
4. Authors concluded that iron overload as the initial ferroptosis signal resulted in WNT-mediated ODC1 expression, which facilitated the synthesis of polyamine from Arginine. The metabolism of polyamine increased H₂O₂, consequently enhancing ferroptosis via the Fenton reaction. However, this reviewer is not aware of any studies about iron overload as the initial prerequisite ferroptosis signal. References are required.
5. Is the increase in EV production caused by iron overload or GPX4 inhibition by RSL3?
6. The model that polyamine senses ferroptosis signal and proactively enhances ferroptosis, but at the same time, the cell undergoing cell death also produces more EV to reduce polyamine level and save itself from ferroptosis seems strange. Are these two events happening at the same time? Or do they represent two different stages of the cell under ferroptosis?

Minor points:

1. Fig 6e is mislabelled
2. The ouroboros symbol in Fig 6j is misleading, as ferroptosis is a terminal cell death process.
3. Some of the English is awkward and hard to understand.

Reviewer #4:

Remarks to the Author:

Bi et al show the potential of exploiting polyamine-mediated ferroptosis as a therapeutic target. The authors present a logical study showing how polyamines can influence the sensitivity to ferroptosis inhibitors and they show some mechanistic understanding behind this.

However a number of areas need to be clarified prior to acceptance:

What concentration of the metabolites were used in the screen? Are they cytotoxic at that concentration? Are all the metabolites screened cell permeable? Why were cells only treated for 20h in the screen,

This concentration of polyamines are cytotoxic on the cells regardless of ferroptosis/RSL3 – is this not just an additive effect?

Need a greater explanation of how the RSL3-A549 cell model works

Fig 2e – the amplification of ferroptotic morphological features upon pre-treatment with polyamines is not obvious from the images – can it be quantified?

Why pre-treatment in Fig 2

Fig 4C – protein levels should be quantified as the increase in ODC1 expression is not obvious

In Fig 4C Fer-1 also reduced ODC1 expression in comparison to DMSO – therefore not convinced iron overload alone is the reason for ODC1 overexpression

Fig 4l-m – these panels jump about a bit in the figure, they do not follow a logical order.

Not clear how specific the effect is on Wnt/Myc – as this is the only pathway tested here. Myc is regulated by other pathways other than Wnt signalling.

The effects of addition of the media from ferroptotic cells vs normal cells in Fig 5c-d are significant but minimal – how biologically relevant are these small changes?

It is important to show the specificity of the samples for EVs in Fig 5f – need to blot with a specific EV marker

Again the effects in Fig 5g are minimal and difficult to understand whether this would have biological significance

Response to Reviewer Comments

Dear Editors and Reviewers,

We appreciate your valuable comments and suggestions for our manuscript titled “Polyamine-mediated ferroptosis amplification acts as a targetable vulnerability in cancer” (Manuscript number: NCOMMS-23-37087-T). We are grateful for the detailed comments and suggestions provided by each Reviewer, and we believe that their input has greatly improved our manuscript. In this letter, we include their comments verbatim in blue, followed by our comments and revisions. We use quotation marks and *Italic* to mark the sentences cited from the manuscript and use **Yellow Background** for the revised messages. Although this makes a rather lengthy letter, we believe that it will provide you and the Reviewers with the best explanation of the changes that we have made.

REVIEWER COMMENTS

Reviewer #1 - radiotherapy, metabolism, lung cancer (Remarks to the Author):

Cellular ferroptosis is an iron-dependent form of regulated cell death due to excessive iron-induced lipid oxidation and redox imbalance both of which play a fundamental role in tumor radiotherapy. This study by Bi et al reveals that arginine is one of the amino acids detected by screening the metabolites in RSL-3-induced ferroptosis in lung cancer A549 cells and adding arginine can enhance the RSL-3 mediated cytotoxicity. By downstream investigation, the polyamines that can be converted from Arginine are found to be the executor for RLS-3-mediated cytotoxicity which is identified by the activation of ODC1 that governs the polyamine synthesis. Based on these findings, a pathway of iron-overload induced ODC1-polyamine-H₂O₂ is suggested which is innovative to the field. Additional findings include the polyamine-containing extracellular vesicles in the tumor microenvironment leading to neighboring cell ferroptosis and polyamine-mediated ferroptosis in tumor sensitization to radiotherapy and chemotherapy.

The overall experimental design and data explanation are reasonable, and the results can fill up some gaps in understanding the mechanisms underlying ferroptosis-contributed tumor radiosensitivity. The evidence supporting the iron-overloading mediated metabolic feedforward loop is informative but could be affected by other cell death pathways which may be discussed. Although some details are in need which include the relevance of physiologic arginine metabolism and ferroptosis-inducing concentrations in tumor cells and potential risk in normal cells. However, the major experimental data are supportive to the conclusion. This work contributes to a new version of metabolism-associated ferroptosis and holds a potential clinical benefit.

Scientific Comments:

1. In the Abstract, a ferroptosis-iron overload-WNT/MYC-ODC1-polyamine-H₂O₂ positive feedback loop that amplifies ferroptosis level is raised but it is unclear how it is termed as a “positive feedback loop” since it does not show H₂O₂ can enhance iron loading.
2. In Figure 1 cartoon, a part showing how the metabolites were generated should be added.
3. Ferroptosis cell death marker may be added in the cell viability test since the arginine and other amino acids also induce other types of cell death
4. Line 101-103, “Screening of this library identified arginine, one of the most versatile non-essential amino acids involved in multiple biological processes¹¹, significantly promoted RSL3

induced ferroptosis (Figure 1b)". This description is not precise for the data shown in Figure 1b since the screening was conducted by RSL3 not Arginine-induced ferroptosis and there seem other metabolites located in the RSL3-sensitive cells.

5. RSL3-induced ferroptosis was shown to be enhanced by adding Arginine which is being increasingly studied for an array of physiological and pathological functions including innate and adaptive immune regulation. Thus a discussion on physical Arginine concentration and dynamics especially in tumor and irradiated tumor microenvironment would be appreciated.

6. A calculation of physiological basal cellular arginine concentration (such as essential diet supplements) with the RSL-3 sensitive cells and the concentration added to enhance RSL-3 ferroptosis will be informative.

7. Line 114-128, it is unclear if the arginine downstream metabolites that have been further studied in the following experiments were also detected in the RSL-3 cytotoxicity shown in Figure 1b.

8. The polyamines released from the ferroptotic cells may belong to the categories of Ghost Messages from different cell death including radiation-induced apoptosis. Thus, it could be highly informative to differentiate or discuss the polyamines as the ghost message from other types of cell death induced by radiotherapy (may refer: Ghostly metabolic messages from dying cells, D.R. Green).

9. Based on the history and challenges in metabolites-targeting approaches, it is appreciated if a specific metabolite in the proposed iron-polyamine loop in ferroptosis could be the top candidate for tumor radiosensitization or chemosensitization.

Reviewer #2 - Ferroptosis, mechanistic (Remarks to the Author):

This study found that arginine metabolites especially polyamine could activate RSL3 or IKE induced ferroptosis, which was activated by the initial ferroptosis signal—iron overload mediated WNT/MYC pathway, and the subsequent increased levels of Odc1, thus providing new therapeutic strategies in cancer treatment. The results were interesting, and the findings is important. But I have some concerns:

Lack of ferroptosis inhabitation experiments in some of important ferroptotic killing experiments. Please add the data.

The effect differences in some cell viability experiments look not that much. please find the optimum conditions.

Reviewer #3 - ferroptosis, metabolism (Remarks to the Author):

Emerging evidence has shown that certain cancers are vulnerable to ferroptosis, making this recently uncovered non-apoptotic cell death pathway a new target for cancer therapy. In the current study, Bi et al. identified Arginine as a ferroptosis promoter. Mechanistically, Arginine is converted into

polyamine. The metabolism of polyamines produces H₂O₂, which increases cellular ROS level and lipid peroxidation via the Fenton reaction. They also noticed that the initiation of ferroptosis by RSL3 caused iron overload, which, through WNT signaling, increased the expression of a critical enzyme catalyzing polyamine synthesis called ODC1. In addition, cells under ferroptosis also enhanced EV production as a protective mechanism to reduce intracellular polyamine levels. However, the release of polyamine-containing EVs also sensitizes the surrounding cells to ferroptosis. Overall, most of the data presented in the current study are convincing and supportive of the conclusion. However, one of the key conclusions that polyamine enhances ferroptosis has been shown by other studies, which compromises the novelty of the current study. The ODC1-mediated feedback loop model is interesting, but additional evidence and clarification are needed.

Major points:

1. Polyamines sensitize cells to RSL3 and FIN56. How about FINO2, which directly oxidizes iron?
2. Where does the overloaded iron come from? Does GPX4 KO cause iron overload?
3. WNT has been reported to inhibit ferroptosis (PMID: 35534546). In this aforementioned study, LF3 treatment sensitized cells to Erastin-induced ferroptosis but not H₂O₂-induced apoptosis, which contradicts what the current study showed.
4. Authors concluded that iron overload as the initial ferroptosis signal resulted in WNT-mediated ODC1 expression, which facilitated the synthesis of polyamine from Arginine. The metabolism of polyamine increased H₂O₂, consequently enhancing ferroptosis via the Fenton reaction. However, this reviewer is not aware of any studies about iron overload as the initial prerequisite ferroptosis signal. References are required.
5. Is the increase in EV production caused by iron overload or GPX4 inhibition by RSL3?
6. The model that polyamine senses ferroptosis signal and proactively enhances ferroptosis, but at the same time, the cell undergoing cell death also produces more EV to reduce polyamine level and save itself from ferroptosis seems strange. Are these two events happening at the same time? Or do they represent two different stages of the cell under ferroptosis?

Minor points:

1. Fig 6e is mislabelled
2. The ouroboros symbol in Fig 6j is misleading, as ferroptosis is a terminal cell death process.
3. Some of the English is awkward and hard to understand.

Reviewer #4 - polyamines (Remarks to the Author):

Bi et al show the potential of exploiting polyamine-mediated ferroptosis as a therapeutic target. The authors present a logical study showing how polyamines can influence the sensitivity to ferroptosis inhibitors and they show some mechanistic understanding behind this.

However a number of areas need to be clarified prior to acceptance:

What concentration of the metabolites were used in the screen? Are they cytotoxic at that concentration? Are all the metabolites screened cell permeable? Why were cells only treated for 20h in the screen,

This concentration of polyamines are cytotoxic on the cells regardless of ferroptosis/RSL3 – is this not just an additive effect?

Need a greater explanation of how the RSL3-A549 cell model works

Fig 2e – the amplification of ferroptotic morphological features upon pre-treatment with polyamines is not obvious from the images – can it be quantified?

Why pre-treatment in Fig 2

Fig 4C – protein levels should be quantified as the increase in ODC1 expression is not obvious

In Fig 4C Fer-1 also reduced ODC1 expression in comparison to DMSO – therefore not convinced iron overload alone is the reason for ODC1 overexpression

Fig 4l-m – these panels jump about a bit in the figure, they do not follow a logical order.

Not clear how specific the effect is on Wnt/Myc – as this is the only pathway tested here. Myc is regulated by other pathways other than Wnt signalling.

The effects of addition of the media from ferroptotic cells vs normal cells in Fig 5c-d are significant but minimal – how biologically relevant are these small changes?

It is important to show the specificity of the samples for EVs in Fig 5f – need to blot with a specific EV marker

Again the effects in Fig 5g are minimal and difficult to understand whether this would have biological significance

To Reviewer #1:

Comment (1).

• In the Abstract, a ferroptosis-iron overload-WNT/MYC-ODC1-polyamine-H₂O₂ positive feedback loop that amplifies ferroptosis level is raised but it is unclear how it is termed as a “positive feedback loop” since it does not show H₂O₂ can enhance iron loading.

Answer:

Thanks very much for your comment. Generally, H₂O₂ reacts with labile iron via the Fenton reaction to generate hydroxyl radicals, which subsequently promote PUFA-PL peroxidation and ferroptotic cell death, especially when the intracellular ferroptosis-protective system, such as GPX4/GSH system, was pharmacologically inhibited. Therefore, H₂O₂ could be regarded as a critical driver of ferroptosis⁽¹⁻³⁾. However, the direct role of H₂O₂ on iron overload has rarely been investigated. Here, as the Reviewer kindly suggested, we further validated that H₂O₂ enhances iron loading in both the presence and absence of RSL3, a GPX4 inhibitor. Mechanically, H₂O₂ promotes ferroptosis induction, thus indirectly enhancing ferroptosis-induced iron loading. We believe these data will provide an essential (patho) physiological link between H₂O₂ and iron overload in the ferroptosis process and make the “positive feedback loop” more convincing. Please find them below.

(Results: Page 10, line 271)

“...an important initiation signal of ferroptosis. We confirmed DFO, Fer-1, and H₂O₂ 's impact on RSL3-induced iron overload by staining intracellular liable iron with FerroOrange (Figure 4f, S3i). Similar results...”

(Revised Figure S3i)

Comment (2).

• In Figure 1 cartoon, a part showing how the metabolites were generated should be added.

Answer:

Thanks very much for your comment. We supplemented our schematic depiction of arginine/polyamine metabolism with information like the source of arginine, urea cycle, generation of putrescine from ornithine decarboxylation, polyamine acetylation and export, and regeneration of methionine (Methionine salvage) as the Reviewer kindly suggested. Please check the revised Figure 1f.

(Revised Figure 1f)

Comment (3).

• Ferroptosis cell death marker may be added in the cell viability test since the arginine and other amino acids also induce other types of cell death.

Answer:

Thanks very much for your comment. In terms of amino acids-induced cell death, Zhang et al. have revealed that excessive branched-chain amino acid sensitized adipose-derived mesenchymal stem cells to stress-induced premature senescence and death⁽⁴⁾. Therefore, it is of great importance

to confirm the presence of ferroptotic cell death in the cell viability test, as the Reviewer kindly suggested. Generally, lipid peroxidation detected by BODIPY-C11 and upregulated PTGS2 mRNA level are considered reliable ferroptosis cell death marker⁽²⁾. In addition, the ferroptotic cell death could only be suppressed by specific ferroptosis inhibitors like ferrostatin-1 (Fer-1, lipid peroxidation scavenger) and deferoxamine (DFO, iron chelator), but not by apoptosis inhibitor z-VAD(OMe)-FMK or necroptosis inhibitor (necrosulfonamide). We added these data and further validated that arginine sensitized cells to ferroptosis rather than other cell death types. Please find them below.

(Results: Page 4, line 108)

“...culture medium (Figure 1c-d, S1a). Since lipid peroxidation is a major hallmark of ferroptosis, we stained cells with BODIPY-C11 581/591 and found that exogenous arginine significantly amplified lipid peroxidation induced by RSL3 (Figure 1e). Upregulation of prostaglandin-endoperoxide synthase 2 (PTGS2, a biomarker of ferroptosis) mRNA was also detected in arginine-treated cells (Figure S1b). Moreover, arginine-caused RSL3 sensitization could be fully rescued by the ferroptosis inhibitor ferrostatin-1 (Fer-1, lipid peroxidation scavenger) and deferoxamine (DFO, iron chelator), but not by apoptosis inhibitor z-VAD(OMe)-FMK or necroptosis inhibitor (necrosulfonamide), confirming the specific promoting role of arginine in ferroptosis (Figure S1c). These data indicate that arginine specifically enhances cells' vulnerability to ferroptosis.”

(Revised Figure S1b)

(Figure S1c)

Comment (4).

• Line 101-103, “Screening of this library identified arginine, one of the most versatile non-essential amino acids involved in multiple biological processes¹¹, significantly promoted RSL3 induced ferroptosis (Figure 1b)”. This description is not precise for the data shown in Figure 1b since the screening was conducted by RSL3 not Arginine-induced ferroptosis and there seem other metabolites located in the RSL3-sensitive cells.

Answer:

Thanks very much for your comment. We apologize for failing to exhibit our findings clearly

and precisely. In the library-screening assay, after cell seeding, the A549 cells were pre-incubated with the metabolites in the library (MCE HY-L030) or vehicle (DMSO, 50 μ M) for 12 hours. Then, the cells were further treated with 4 μ M RSL3 or DMSO for 8 h and the cell viability was measured using CCK8. The algorithm for the “Relative viability” exhibited in Figure 1b was added to the Figure: in the formula, “X” represents the pre-treatment of a specific metabolite, and “V” represents the pre-treatment of Vehicle DMSO.

$$\text{Relative viability} = \frac{X_{RSL3}/X_{DMSO}}{V_{RSL3}/V_{DMSO}}$$

Therefore, the metabolites promoting or inhibiting RSL3-induced ferroptosis were identified. Here, DMSO was used as both the “control” groups for metabolite pre-treatment and RSL3 treatment. To avoid unnecessary misunderstanding, we added the formula of “Relative viability” to Figure 1b and the corresponding Figure legend and added “RSL3” to the “Resistant/Sensitizing” part. Please check it.

(Figure legend: Page 38, line 1186)

“Figure 1b. The metabolite library screening results were exhibited as the relative viability of metabolite-treated cells versus vehicle-treated cells. After cell seeding, the A549 cells were pre-incubated with the metabolites contained in the library (MCE HY-L030) or Vehicle (DMSO, 50 μ M) for 12 hours. Then, the cells were further treated with 4 μ M RSL3 or DMSO for 8 h and the cell viability was measured using CCK8. In the algorithm for the “Relative viability,” “X” represents the pre-treatment of a specific metabolite, and “V” represents the pre-treatment of Vehicle DMSO.”

(Revised Figure 1b)

The “Resistant to RSL3” and “Sensitizing to RSL3” indicate that the pre-treatment with a specific metabolite alters A549 cells’ sensitivity to RSL3-induced ferroptotic cell death. Aside from arginine and polyamine mentioned in our research, known ferroptosis promotor “arachidonic acid,” and several other metabolites sensitizing cells to ferroptosis were also identified in the screening, such as 2,4-Di-tert-butylphenol and deoxycorticosterone acetate. Our subsequent research will investigate the underlying mechanism of their ferroptosis-sensitizing effect. In the present study, we chose arginine because of the significance of amino acids in biological metabolism. A recent study published in *Nature Chemical Biology* mentioned the ferroptosis-sensitizing effect but did not elucidate the corresponding mechanism⁽⁵⁾. We mentioned this in the Discussion section. Please check.

(Discussion: Page 16, line 444)

“Emerging evidence has revealed that amino acid metabolism plays an important role in the

initiation and regulation of ferroptosis. Aside from the well-known role of cysteine as a precursor of GSH, methionine also participates in the synthesis of GSH through the transsulfuration pathway upon extracellular cysteine limitation, while the tryptophan metabolites serotonin (5-HT) and 3-hydroxyanthranilic acid (3-HA) act as a potent radical trapping antioxidant to eliminate ferroptotic lipid peroxidation. As one of the most versatile non-essential amino acids, arginine not only serves as a key metabolic intermediate of the urea cycle responsible for eliminating excessive endogenous ammonia but also helps in cell division, wound healing, and immune response through its downstream metabolic products. In recent years, arginine starvation has been well implicated in cancer therapy, whereas its involvement in ferroptosis and the corresponding mechanism remains enigmatic⁴⁷. Although Conlon et al. have mentioned the protective effect of arginine deprivation in ferroptosis, the mechanism of this property remains unknown. Expected mTOR signalling, GCN2/ATF4-transsulfuration pathway, arginine's physical property, or shared metabolic roles with other ferroptosis-associated amino acids failed to explain this phenomenon. In contrast, some researchers reported the antioxidative role of arginine derived from its participation in the production of nitric oxide (NO) and NRF2 pathway, but we obtained opposite results in our study. Through metabolic tracing analysis... ”

Comment (5).

- RSL3-induced ferroptosis was shown to be enhanced by adding Arginine which is being increasingly studied for an array of physiological and pathological functions including innate and adaptive immune regulation. Thus a discussion on physical Arginine concentration and dynamics especially in tumor and irradiated tumor microenvironment would be appreciated.

Answer:

Thanks very much for your comments. Considering the immune-modulating effect of arginine, we discussed the physical arginine level and dynamic metabolic pattern in tumour and the irradiated tumour microenvironment in detail as the Reviewer kindly suggested. Please find the information in the Discussion section.

(Discussion: Page 16, line 466)

“Generally, the physical level of arginine in serum fluctuates between 50 and 150 μ M, while the intracellular levels are in the range of 100-800 μ M. Notably, arginine deficiency is a hallmark of the tumour microenvironment. Under such starvation, the anti-tumour responses of T cells are significantly blunted, including reduced CD3 ζ chain and cytokines, inhibited proliferation, and enhanced apoptosis. This phenomenon is largely attributed to the arginine-consuming effect of arginase-expressing immunosuppressive myeloid cells such as M2 macrophage and myeloid-derived suppressor cells. Moreover, the arginase level in these cells could be further enhanced by radiotherapy, exacerbating the arginine deficiency and T cell dysfunction. Therefore, is it possible that exogenously supplemented arginine could simultaneously promote tumour cell ferroptosis through the arginine-polyamine axis and enhance anti-tumour immunity? Does arginine's ferroptosis-promoting effect also apply to immune cells? The dynamic metabolic pattern of arginine in the tumour environment and its correlation with patients' response to ferroptosis-based therapy still warrants further investigation.”

Comment (6).

- A calculation of physiological basal cellular arginine concentration (such as essential diet

supplements) with the RSL-3 sensitive cells and the concentration added to enhance RSL-3 ferroptosis will be informative.

Answer:

Thanks very much for your comments. As indicated by previous researchers, generally, the physical level of arginine in serum fluctuates between 50 and 150 μM , while the intracellular levels are in the range of 100-800 μM ^(6, 7). We also performed targeted metabolomics and found that the arginine concentration in the A549 cell line used in our study was 91 μM . The DMEM medium which we used to culture cells containing 400 μM arginine to support the rapid-proliferating rate of tumour cells. Therefore, to select an optimal arginine concentration for the ferroptosis-viability test in our cell model, we adopted a concentration gradient of 0-100-200-400-800 μM when investigating exogenous arginine's impact on cells' response to ferroptosis and finally chose 500 μM , thus making the actual arginine concentration (900 μM) in cultural medium about twice that of normal DMEM medium. The supplemented arginine did not impact cells' viability or proliferation at such concentration, but significantly sensitized cells to ferroptosis. As the Reviewer kindly suggested, we explained the reason for choosing this concentration gradient in the Method section. Please find them below.

(Methods: Page 23, line 657)

"...Molecular Devices microplate reader (Biotek, U.S.A). Since the cells were cultured in DMEM medium containing 400 μM arginine, we adopted a concentration gradient of 0-100-200-400-800 μM exogenous arginine in ferroptosis-related cell viability assay and finally chose 500 μM . At such concentration, the supplemented arginine did not impact cells' viability or proliferation."

And of course, considering the complexity of tumour microenvironment, the actual concentration of arginine in tumour region and its correlation with diet arginine level is determined by numerous factors such as arginine absorbance, transportation, accumulation, and the arginine-depleting effect of immunosuppressive myeloid cells. Further in vivo studies on diet arginine and tumour patients' response to ferroptosis-based therapy are still warranted.

(Discussion: Page 17, line 476)

"...immune cells? The dynamic metabolic pattern of arginine in the tumour environment and its correlation with patients' response to ferroptosis-based therapy still warrants further investigation."

Comment (7).

• Line 114-128, it is unclear if the arginine downstream metabolites that have been further studied in the following experiments were also detected in the RSL-3 cytotoxicity shown in Figure 1b.

Answer:

Thanks very much for your comments. In the present study we confirmed that pro-ferroptotic role of arginine is mediated by its downstream metabolite ornithine and polyamine. In addition to arginine, the metabolite library adopted in this study also includes ornithine, citrulline, proline, spermine, spermidine, except putrescine. In our initial metabolites screening results, the ferroptosis-promoting effects were detected in ornithine (0.661), spermine (0.298), spermidine (0.517), but not in citrulline (1.005, not shown in Figure 1b) or proline (0.998, not shown in Figure 1b), which was consistent with our subsequent experiments (Figure 1h, S1c, 2a-b, S2a-d). We further emphasized

this consistency as the Reviewer kindly suggested. Please find them below.

(Results: Page 6, line 149)

“...whereas polyamines displayed stronger ferroptosis sensitizing effect than arginine even in a low concentration without markedly impacting cell’s viability when solely administered (spermine > spermidine > putrescine), which was consistent with our initial metabolites screening results (Figure 1b, 2a, S2a-b). The broad-spectrum sensitizing activity...”

(Revised Figure 1b)

Comment (8).

• The polyamines released from the ferroptotic cells may belong to the categories of Ghost Messages from different cell death including radiation-induced apoptosis. Thus, it could be highly informative to differentiate or discuss the polyamines as the ghost message from other types of cell death induced by radiotherapy (may refer: Ghostly metabolic messages from dying cells, D.R. Green).

Answer:

Thanks very much for your comments. We consulted Dr. D.R. Green’s publication and the research conducted by Medina et al. focusing on apoptotic cells’ “good-bye signal”, and further discussed these exciting discoveries as the Reviewer kindly suggested. Please find them below.

(Discussion: Page 19, line 541)

“It has been revealed that dying cells release a set of signals to impact surrounding cells, called “Ghost Message”, and different cell death types generate profoundly different effects. For instance, apoptotic cells synthesize and release several metabolites, including spermidine, through a particular protein channel, PANX1, thereby recruiting macrophages to express genes involved in tissue repair and inhibition of inflammation. As for necrotic cells, they release damage-associated molecular patterns (DAMPs) like urea and drive inflammation. Ferroptotic cells also release a set of signals to attract immune cells to the site of ferroptotically dying cells. Brown’s research has proposed an interesting model in which pro-ferroptotic stimuli induce the expression of prominin 2 that promotes the formation of ferritin-containing multivesicular bodies and exosomes, thereby facilitating exosomal transport of ferritin out of the cell and leading to ferroptotic resistance. This effect could be considered a cell’s self-productive mechanism against ferroptotic cell death. Besides, current modes of mammalian polyamine transport agree that polyamines exist in PSVs, from which they can be released into cytosol or secreted. Inspired by this evidence, we investigated the abundance of extracellular polyamines after RSL3 treatment and found that ferroptotic cells tended to upregulate the secretion of excessive polyamine into the microenvironment through vesicles. We

speculated that this process might also to some extent retard the polyamine-centred ferroptotic positive feedback loop by expelling over-produced polyamine out of the cell. On the contrary, the exported polyamine further accelerated the diffusion of ferroptosis inside the tumour region. Previous studies have suggested that ferroptosis can spread through cell populations as a rapidly propagating wave, resulting in a distinct spatiotemporal pattern of cell death. This phenomenon involves the spreading of a cell swelling effect through cell populations in a lipid peroxide- and iron-dependent manner. Together with these two ferroptotic executors, the released polyamines from ferroptotic cells further “ignite” the spread of ferroptotic cell death. Therefore, polyamine metabolism contributes to a “dual circulation system” consisting of both intracellular ferroptosis-iron overload-WNT/MYC/ODC1-polyamine-H₂O₂ loop and extracellular released polyamine induced ferroptosis diffusion, suggesting the key modulating role of polyamine in the regulation and execution of ferroptosis. **Notably, considering that radio- or chemotherapy induces both apoptosis and ferroptosis, theoretically, both of which lead to spermidine synthesis and release, although through distinct pathways, it would be an interesting topic for future investigation whether these two pathways would interfere with each other and which one would have the dominating effect in different disease contexts. Besides, polyamines...**”

Comment (9).

• Based on the history and challenges in metabolites-targeting approaches, it is appreciated if a specific metabolite in the proposed iron-polyamine loop in ferroptosis could be the top candidate for tumor radiosensitization or chemosensitization.

Answer:

Thanks very much for your comments. Our study demonstrated that polyamines’ ferroptosis mediating effect depends on PAOX/SMOX-catalysed H₂O₂ generation. Among the three types of polyamines, spermine, whose conversion to putrescine leads to the production of two H₂O₂ molecules, exhibited the strongest ferroptosis-promoting property. Spermidine’s conversion to putrescine generates one H₂O₂ molecule, making it the second strongest ferroptosis promoter. As for putrescine, only after being transformed to spermine or spermidine again can it exhibit a pro-ferroptotic role. Therefore, spermine, which could be easily obtained from animal-derived foods, serves as the top candidate for tumour radio-sensitization or chemo-sensitization. We further mentioned this in our manuscript as the Reviewer kindly suggested. Please find them below.

(Results: Page 8, line 230)

“...Taken together, polyamines’ ferroptosis mediating effect is dependent on PAOX/SMOX catalysed H₂O₂ generation. Among the three types of polyamines, spermine, whose conversion to putrescine leads to the production of two H₂O₂ molecules, exhibited the strongest ferroptosis-promoting property. Spermidine’s conversion to putrescine generates one H₂O₂ molecule, making it the second strongest ferroptosis promoter. As for putrescine, only after being transformed to spermine or spermidine again can it exhibit a pro-ferroptotic role. Notably...”

(Discussion: Page 20, line 592)

“...ferroptosis-based treatment. **Specifically, spermine, the strongest ferroptosis promotor in the three polyamine types, which could be obtained from animal-derived foods, could be a promising candidate for tumour radio- and chemo-sensitization.**”

To Reviewer #2:

Comment (1).

• Lack of ferroptosis inhibition experiments in some of important ferroptotic killing experiments. Please add the data.

Answer:

Thanks very much for your comment. It is of great importance to confirm that the treatments specifically (RSL3 or IKE-based assays) or at least partly (radio- or chemotherapy-based assays) induce ferroptosis rather than other types of cell death. Therefore, it is necessary for us to test whether the cell death could be rescued by specific ferroptosis inhibitors including ferrostatin-1 (Fer-1, lipid peroxidation scavenger) and deferoxamine (DFO, iron chelator), but not by apoptosis inhibitor z-VAD(OMe)-FMK or necroptosis inhibitor (necrosulfonamide). In the present study, as the Reviewer kindly suggested, we added the ferroptosis inhibition experiments when validating the ferroptosis-promoting effect of the following agents: arginine, polyamines, WNT-specific agonist SKL2001, and polyamines' radio- and chemo-sensitizing effect. Please find them below.

(Results: Page 4, line 111)

“...arginine-treated cells (Figure S1b). Moreover, arginine-caused RSL3 sensitization could be fully rescued by the ferroptosis inhibitor ferrostatin-1 (Fer-1, lipid peroxidation scavenger) and deferoxamine (DFO, iron chelator), but not by apoptosis inhibitor z-VAD(OMe)-FMK or necroptosis inhibitor (necrosulfonamide), confirming the specific promoting role of arginine in ferroptosis (Figure S1c). These data indicate that arginine enhances cells' vulnerability to ferroptosis.”

(Figure S1c)

(Results: Page 6, line 162)

“...(Figure 2e). The ferroptosis-sensitizing effect of polyamines was fully rescued by Fer-1 and DFO, but not by z-VAD(OMe)-FMK or necrosulfonamide, confirming that polyamine specifically promotes ferroptosis (Figure 2f, S2f). These findings...”

(Figure 2f)

(Figure S2f)

(Results: Page 10, line 293)

“...Consistently, we further observed that SKL2001 and LF3 treatment respectively sensitized and desensitized cells to FINs in a concentration-dependent manner, suggesting WNT signalling pathway impairs cells’ ferroptosis resistance (Figure 4n-o, S4a-b). **SKL2001-caused ferroptosis-sensitization could be fully rescued by Fer-1 and DFO (Figure S4c).** Besides...”

(Revised Figure S4c)

(Results: Page 13, line 367)

“...As expected, exogenous supplementation of polyamines significantly sensitized cells to RT or CDDP, as well as amplified the lipid peroxidation induced by these treatments (Figure 6a-c). Importantly, the radio-/chemo-sensitizing effect of polyamine could be largely abolished by the ferroptosis inhibitor Fer-1 (Figure 6d). Collectively...”

(Figure 6d)

Comment (2).

• The effect differences in some cell viability experiments look not that much. please find the optimum conditions.

Answer:

Thanks very much for your comment. We optimized the condition of some cell viability experiments as the Reviewer kindly suggested and received ideal results. Please find them below.

(Revised Figure 1d)

(Revised Figure 1k)

(Revised Figure S1a)

(Revised Figure S2b)

(Revised Figure 3d)

(Revised Figure S4a)

To Reviewer #3:

Major Points:

Comment (1).

• Polyamines sensitize cells to RSL3 and FIN56. How about FINO₂, which directly oxidizes iron?

Answer:

Thanks very much for your comment. As the Reviewer kindly suggested, we tested this and found that polyamine failed to significantly sensitize cells to FINO₂. This phenomenon could be explained by the mechanism of FINO₂'s ferroptosis inducing property. As demonstrated by Gaschler et al., FINO₂, an endoperoxide, indirectly inhibits GPX4's enzymatic function and directly initiates Fenton reaction by oxidizing Fe²⁺ to Fe³⁺ and generating alkoxy radical inducing lipid peroxidation⁽⁸⁾. That is, when FINO₂ replaced H₂O₂ and induced ferroptosis independent of it, polyamine's ferroptosis-promoting effect afforded by H₂O₂ production was largely abolished. And of course, this finding also further confirmed that the ferroptosis-promoting effect of polyamines is mediated by H₂O₂. Please find this in the Results section.

(Results: Page 6, line 152)

“...metabolites screening results (Figure 1b, 2a, S2a-b). The broad-spectrum sensitizing activity of polyamines was also corroborated in FIN56, another FIN depleting both GPX4 and CoQ, but not in FINO₂, the class IV FIN which directly oxidizes iron (Figure S2c). Interestingly...”

(Revised Figure S2c)

(Results: Page 8, line 230)

“... Δ catalase expression (Figure 3i). Taken together, polyamines' ferroptosis mediating effect is dependent on PAOX/SMOX catalysed H_2O_2 generation. This mechanism explains why polyamines failed to sensitize cells to $FINO_2$, an endoperoxide that directly initiates Fenton reaction by oxidizing Fe^{2+} to Fe^{3+} and generating alkoxyl radical inducing lipid peroxidation. That is, when $FINO_2$ replaced H_2O_2 and induced ferroptosis independent of it, polyamine's ferroptosis-promoting effect afforded by H_2O_2 production was largely abolished. Among the three types of polyamines...”

Comment (2).

- Where does the overloaded iron come from? Does GPX4 KO cause iron overload?

Answer:

Thanks very much for your comment. We consulted several high-quality studies and noted our finding that RSL3-induced GPX4-inhibition indeed causes iron overload was consistent with previous researchers, such as Brown's⁽⁹⁾ and Chen's⁽¹⁰⁾ publication. Multiple theories for ferroptosis-induced iron overload have been proposed yet, such as the release of GSH-bound ferrous^(11, 12), overexpression of TFRC, downregulation of iron exporter SLC40A1⁽¹³⁾, and ferritinophagy. Ferritinophagy refers to an autophagic process in which cellular iron storage proteins like ferritin are degraded, and the liable iron (ferrous) is accumulated. This process was first reported in 2016 by Prof. Daolin Tang and further investigated by Prof. Xuejun Jiang, both of whom are well-known scholars in the area of ferroptosis^(14, 15). The researchers found that ferroptosis-inducing conditions triggered ferritinophagy, while inhibiting autophagy, or ferritinophagy, significantly inhibited ferroptosis initiated by both RSL3 and erastin. These findings suggested that ferritinophagy is essential for ferroptotic cell death⁽¹⁴⁾. Therefore, iron overload induced by GPX4 inhibition is probably due to ferritinophagy. However, the exact mechanism of how ferroptotic signal, especially initiated by GPX4 inhibition or knockout, triggers ferritinophagy still needs further exploration. Please find this information in the Discussion section.

(Discussion: Page 18, line 506)

“Ferrous, also known as Fe^{2+} , is maintained in cells in the form of the liable iron pool, which bounds to low molecular weight compounds such as GSH. Increased cellular liable iron is usually observed during the induction of ferroptosis initiated by either SLC7A11 inhibitor erastin or GPX4 inhibitor RSL3. There are several explanations for this phenomenon. On the one hand, erastin-induced GSH depletion releases the bound Fe^{2+} and mobiles them for Fenton reaction and subsequent ferroptosis. On the other hand, accumulating data suggested that autophagic signals are generated in response to ferroptotic stress and promote the degradation of cellular iron storage protein ferritin, thus leading to iron accumulation. This process is termed as “ferritinophagy”.

However, the exact mechanism of how ferroptosis triggers ferritinophagy still needs further exploration.”

Comment (3).

• WNT has been reported to inhibit ferroptosis (PMID: 35534546). In this aforementioned study, LF3 treatment sensitized cells to Erastin-induced ferroptosis but not H₂O₂-induced apoptosis, which contradicts what the current study showed.

Answer:

Thanks very much for your comment. During the preparation phase of this research, we read the article “Wnt/beta-catenin signalling confers ferroptosis resistance by targeting GPX4 in gastric cancer” (PMID: 35534546) mentioned by the Reviewer. The authors presented a logical and convincing study demonstrating that the WNT/beta-catenin/TCF4 activates the transcription of GPX4 and subsequently inhibits ferroptosis in gastric cancer, thus suggesting a potential therapeutic strategy to enhance chemo-sensitivity for advanced gastric cancer patients. This finding contradicts our result that LF3 desensitized A549 and HT1080 cells to ferroptosis inducers IKE and RSL3. We repeatedly verified our experimental results and received consistent results. Therefore, we believe that the impact of WNT signalling on ferroptosis is context-dependent. In tumour cell lines possessing different genomic characteristics, altering WNT signalling might generate varied effects. We mentioned this in the Discussion section. Please check them below.

(Discussion: Page 18, line 533)

“...mediating this effect. Besides, previous researchers also implied ferroptosis inhibiting property of WNT signalling in gastric cancer and osteoblast differentiation, indicating a context-dependent role of this pathway in the regulation of ferroptosis.”

In the study “Wnt/beta-catenin signaling confers ferroptosis resistance by targeting GPX4 in gastric cancer” (PMID: 35534546), the authors declared that “LF3 treatment selectively enhanced sensitivity to erastin-induced ferroptosis but did not affect actinomycin D-induced autophagy, TNF α -induced necroptosis, or H₂O₂-induced apoptosis”. Currently, H₂O₂ is regarded as a common apoptosis inducer, and numerous mechanisms have been proposed to explain this phenomenon. On the other hand, as a key substrate of the Fenton reaction, a catalytic process in which Fe²⁺ reacts with H₂O₂ to generate Fe³⁺, HO \cdot , and OH $^-$, the role of H₂O₂ in the regulation of ferroptosis has attracted researchers’ attention. In the existence of Fe²⁺, and especially when the anti-oxidative system such as GPX4/GSH and FSP1/CoQ₁₀ are pharmacologically or genetically inhibited, H₂O₂-

(Chen X, et al. Nat Rev Clin Oncol. 2021. PMID: 33514910)

(Dixon S, et al. Mol Cell. 2023. PMID: 36977413)

derived HO· triggers the chain-reaction of lipid peroxidation. The two figures exhibited in the Review published by Chen et al. and Dixon et al. exactly summarized this process^(16, 17).

In our study, when the cells were solely treated with polyamines (10 μM) without ferroptosis inducers, the cells' viability was not significantly altered, suggesting that the H₂O₂ generated from polyamine metabolism in such concentration was insufficient to induce apoptosis. However, when RSL3 or IKE was added to the system, the ferroptosis was significantly amplified by polyamines, which could be rescued by ferroptosis inhibitors like ferrostatin-1 (Fer-1, lipid peroxidation scavenger) and deferoxamine (DFO, iron chelator), but not by apoptosis inhibitor z-VAD(OMe)-FMK or necroptosis inhibitor (necrosulfonamide). Therefore, rather than apoptosis, ferroptosis plays the dominant role in this context.

Comment (4).

• Authors concluded that iron overload as the initial ferroptosis signal resulted in WNT-mediated ODC1 expression, which facilitated the synthesis of polyamine from Arginine. The metabolism of polyamine increased H₂O₂, consequently enhancing ferroptosis via the Fenton reaction. However, this reviewer is not aware of any studies about iron overload as the initial prerequisite ferroptosis signal. References are required.

Answer:

Thanks very much for your comment. Since the iron-chelator, deferoxamine, could completely inhibit ferroptosis, iron overload is regarded as a key event in the initiation and progression of ferroptosis⁽¹⁸⁾. Ferroptosis-induced induced iron-overload has been reported in several high-quality studies^(9, 12, 19). Please find this issue in the following reference:

(1) Brown CW, et al. Prominin2 Drives Ferroptosis Resistance by Stimulating Iron Export. *Dev Cell*. 2019 Dec 2;51(5):575-586.e4⁽⁹⁾

(Results)

“...when either *prominin2* or both *FTH1* and *FTL* were silenced, we observed an increase in the concentration of free iron following RSL3 treatment (Figures 6A, 6B, and S6D). **Additional evidence for *prominin2* regulation of iron levels was obtained by quantifying free iron in control and *prominin2*-expressing MDA-MB-231 cells in response to GPX4 inhibition. *Prominin2* expression in these cells prevented the increase in iron in response to RSL3 that was observed in control cells (Figure 6C). Also, disabling the MVB pathway by decreasing expression of TSG101 significantly increased the concentration of free intracellular iron following GPX4 inhibition (Figure 6D). These results indicate that blocking MVB/exosome-mediated ferritin export results in an accumulation of intracellular iron upon GPX4 inhibition.**”

(Discussion)

“Free iron is essential for the execution of ferroptosis (Dixon et al., 2012, 2014), and an increase in iron import combined with reduced ferritin storage capacity is a characteristic of era-

stin-sensitive cells (Torti et al., 2018; Yang and Stockwell, 2008).”

“...Our results imply that **cytosolic iron increases in response to GPX4 inhibition or ECM detachment, but that this iron can be captured, incorporated into MVBs, and exported from the cell in exosomes if prominin2 expression is sufficient. It could be argued...**”

(2) Aron AT, et al. An Endoperoxide Reactivity-Based FRET Probe for Ratiometric Fluorescence Imaging of Labile Iron Pools in Living Cells. *J Am Chem Soc.* 2016 Nov 2;138(43):14338-14346⁽¹⁹⁾.

(Results)

“Building on the demonstrated ability of FIP-1 to detect endogenous changes in labile Fe(II) in MDA-MB-231 cells (Figure 3), we turned our attention to linking labile iron fluxes to this model for ferroptosis. We observed that MDA-MB-231 cells begin to undergo exponential cell death when treated with 1.25 μM 35MEW28 (a recently reported inducer of ferroptosis)⁶⁶ after 10–12 h. **For labile iron detection, we imaged cells using FIP-1 at various time points after treatment with 35MEW28 (Figure 4). Interestingly, we observed that the Green/FRET ratio increased 2 h after treatment as compared to the vehicle control and the signal further increased over time (Figure 4a, b). To validate that the ratiometric fluorescence response was derived from changes in the labile iron pool, we coincubated cells with 35MEW28 and 100 μM DFO. Confocal microscopy measurements at the 8 h time point no longer revealed an increased Green/FRET ratio (Figure 4c) compared to control (Figure 4a). However, when the cells are cotreated with 35MEW28 and the lipophilic antioxidant Fer-1, which blocks ferroptosis downstream of where we hypothesize a ferrous iron elevation to occur, we observe a Green/FRET ratio that is equivalent to cells treated with 35MEW28 alone (Figure 4d). As such, the data are consistent with the model that Fer-1 does not alter the mobilization of Fe(II) and indicates that the observed change in Green/FRET ratio is not simply an artifact of the process of ferroptosis. Taken together, these imaging results suggest that treatment with 35MEW28 may alter iron homeostasis to increase labile Fe(II) levels, serving as direct evidence that ferroptosis may be altering labile Fe(II) levels.**”

Figure 4. FIP-1 enables direct detection of changes in labile iron pools upon induction of ferroptosis. Confocal microscopy of 10 μM FIP-1 in MDA-MB-231 cells treated with (a) vehicle, (b) 1.25 μM 35MEW28 (see structure above) for 8 hours, (c) 1.25 μM 35MEW28 + 100 μM DFO for 8 hours, and (d) 1.25 μM 35MEW28 + 1 μM Fer-1 for 8 hours. (e-h) Brightfield images of (a-d). (i) Mean Green/FRET ratios of MDA-MB-231 cells treated with ferroptosis-inducing compounds and inhibitors; error bars denote SEM, $n = 3$. Statistical significance was assessed by calculating p -values using one-way ANOVA with the Bonferroni correction in R, $*p < 0.05$. Scale bar = 25 μm .

(3) Stockwell BR. Ferroptosis turns 10: Emerging mechanisms, physiological functions, and therapeutic applications. *Cell.* 2022 Jul 7;185(14):2401-2421.⁽²⁾ (Review)

“Fe(II) is maintained in cells in the form of the labile iron pool, bound to low molecular weight compounds, including GSH (Patel et al., 2019). The depletion of GSH can not only inactivate GPX4 but also mobilize Fe(II) for Fenton chemistry, promoting the propagation of lipid peroxides and ultimately ferroptosis. In addition, iron storage in ferritin requires the formation of the GSH-iron

complex, which is delivered to ferritin via the chaperone poly(rC) binding protein 1 (PCBP1) (Patel et al., 2021). Thus, depletion of GSH promotes the availability of labile iron.”

Notably, the study “A noncanonical function of EIF4E limits ALDH1B1 activity and increases susceptibility to ferroptosis” published in *Nature Communications* by Prof. Daolin Tang’s team has demonstrated the signalling property of iron-overload occurred during ferroptosis, which could be inhibited by iron-chelator deferoxamine⁽¹⁰⁾. As cited below:

(Results)

“Next, we investigated the effect of ferroptosis inducer signaling on the formation of the EIF4E-ALDH1B1 complex. The iron chelator deferoxamine inhibited RSL3-induced formation of the EIF4E-ALDH1B1 complex in Calu-1 cells (Fig. 3j). Compared with RSL3 treatment, 20% FBS and staurosporine did not increase intracellular iron accumulation (Fig. 3k). Furthermore, immunoprecipitation analysis revealed that deferoxamine inhibited RSL3-induced the formation of the ALDH1B1- EIF4E complex in membrane protein extraction (Fig. 3l), suggesting that iron accumulation, **the most important initiation signal of ferroptosis**, is conducive to this process.”

Taken together, this evidence suggested that iron overload occurs during the ferroptosis process, and the accumulated liable iron exhibited signalling properties in the regulation of specific biological processes, which are consistent with our findings. As the Reviewer kindly suggested, we cited the references mentioned above in our manuscript. Please find them below.

(Results: Page 10, line 267)

“...This unexpected finding prompted us to associate ODC1 regulation with the difference between the mechanisms of DFO and Fer-1’s ferroptotic inhibiting function. DFO, an iron chelator, inhibits ferroptosis by blocking liable iron and the corresponding iron-dependent Fenton reaction, whereas Fer-1, an antioxidant, scavenges lipid peroxides, **but does not mitigate iron overload, an important initiation signal of ferroptosis. We confirmed DFO, Fer-1, and H₂O₂’s impact on RSL3-induced iron overload by staining intracellular liable iron with FerroOrange (Figure 4f, S3i).** Similar results obtained from another type of probe have also been reported by Aron et al.”

(Discussion: Page 18, line 520)

“...Recent work has suggested that iron not only supports the function of several vital iron- and haem-containing enzymes as co-factor but also regulates a series of crucial pathways in tumours. **For instance, Chen et al. have proposed an interesting phenomenon that ferroptosis-induced iron accumulation promotes the formation of ALDH1B1-EIF4E complex, leading to ALDH1B1-degradation and ferroptosis-sensitization. Among them, WNT signalling emerges as one**

of the best-known pathways impacted by iron. It has been...”

Comment (5).

- Is the increase in EV production caused by iron overload or GPX4 inhibition by RSL3?

Answer:

Thanks very much for your comment. As the Reviewer kindly suggested, we further tested the EV production level in A549 cells treated by ferric ammonium citrate (FAC) or vehicle using CD63 and TSG101 as EV marker proteins but found that the iron-overload mimicked by FAC treatment did not generate significant alteration. We mentioned this in the Result section. Please find them below.

(Results: Page 12, line 347)

“...cell-derived EVs. Although the polyamine contents of EVs from DMSO- and RSL3-treated cells were nearly identical, ferroptotic cells produced more EVs than healthy cells, as evidenced by the larger amount of EV marker proteins presented in RSL3-treated cells (Figure 5f). However, FAC treatment failed to generate similar effects, suggesting that the increased EV production was not a result of iron overload (Figure S4h).”

(Revised Figure S4h)

The increased EV production induced by RSL3 has been reported by Brown et al. in their study “*Prominin2 Drives Ferroptosis Resistance by Stimulating Iron Export*”, in which the authors illustrated an interesting mechanism that ferroptosis stress induces the expression of Prominin2, which stimulates the formation of ferritin-containing multivesicular bodies/exosomes that transport iron out of the cell, thereby contributing to ferroptosis-resistance. However, in the A549 used in our study, the expression of Prominin2 is almost absent, but RSL3-induced GPX4 inhibition still led to EV overproduction, indicating the existence of other underlying mechanisms. Although a negative result was acquired in the Reviewer-suggested iron-related assay, this finding still brought us closer to the truth. Further exploration of this issue is still warranted in the future.

Comment (6).

- The model that polyamine senses ferroptosis signal and proactively enhances ferroptosis, but at the same time, the cell undergoing cell death also produces more EV to reduce polyamine level and save itself from ferroptosis seems strange. Are these two events happening at the same time? Or do they represent two different stages of the cell under ferroptosis?

Answer:

Thanks very much for your comment. In the viability assay carried out in this research, we

commonly treated A549 cells with 4 μM RSL3 for 8 h, and almost 60-70% of cells were killed in such circumstances. As for the EV-related assays, to provide cells with enough time to generate adequate EVs to be observed, we preferred a lower concentration but longer incubating time by treating A549 cells with 0.5 μM RSL3 for 24 h, and the death rate here was only 20-30%, as shown in our preliminary experiment. We further validated that EVs production began to increase 6 hours after RSL3 treatment and reached peak after 12 hours, represented by the EV specific marker CD63, as shown below.

Therefore, we believe that the two events, the “death signal” RSL3-induced ferroptotic cell death, and the “protect signal”, overproduction of polyamine-containing EV, happen simultaneously, and the “belonging of victory” depends on RSL3’s concentration or the rate of ferroptosis proceeding progress. If the ferroptosis stress is too weak to eliminate the tumour cells in a relatively short period, the cells will have enough time to “deploy defensive forces” and export polyamines through EVs, thus limiting the amplification of ferroptosis in a reversible state. In contrast, if the cells with high-concentration of ferroptosis inducers which completely inhibit GPX4, the rapidly accumulated lipid peroxides will trigger cell death immediately before the polyamines are exported.

Minor Points:

Comment (1).

- Fig 6e is mislabelled.

Answer:

Thanks very much for your comment. We apologize for our carelessness and have revised this mistake.

(Revised Figure 6e)

e

Comment (2).

- The ouroboros symbol in Fig 6j is misleading, as ferroptosis is a terminal cell death process.

Answer:

Thanks very much for your comment. We deleted it as the Reviewer kindly suggested.

Comment (3).

- Some of the English is awkward and hard to understand.

Answer:

Thanks very much for your comment. We had our manuscript reviewed by a native speaker of

English from International Science Editing company as the Reviewer kindly suggested.

To Reviewer #4:

Comment (1).

• What concentration of the metabolites were used in the screen? Are they cytotoxic at that concentration? Are all the metabolites screened cell permeable? Why were cells only treated for 20h in the screen.

Answer:

Thanks very much for your comment. In the metabolite screening assay, A549 cells seeded in 96-well plates were pre-treated with indicated metabolites (50 μ M) for 12 h followed by 4 μ M RSL3 treatment for 8 h. Just as the Reviewer said, some of the metabolites, such as deoxycholic acid sodium salt, skatole, D-erythro-sphingosine, and 2,4-Di-tert-butylphenol exhibited cytotoxicity at such concentration even without the addition of RSL3. However, the confounding effects of the cytotoxicity of metabolites themselves were eliminated in the final screening result because the ratio of metabolite-RSL3 group to metabolite-DMSO group was calculated. Please see the Figure 1b.

(Revised Figure 1b)

(Figure legend, Page 38, line 1186)

b, The metabolite library screening results were exhibited as the relative viability of metabolite-treated cells versus vehicle-treated cells. After cell seeding, the A549 cells were pre-incubated with the metabolites contained in the library (MCE HY-L030) or Vehicle (DMSO, 50 μ M) for 12 hours. Then, the cells were further treated with 4 μ M RSL3 or DMSO for 8 h and the cell viability was measured using CCK8. In the algorithm for the “Relative viability”, “X” represents the pre-treatment of a specific metabolite, and “V” represents the pre-treatment of Vehicle DMSO.”

We referred to a series of high-quality studies of metabolite library screening when designing the screening strategy^(5, 10, 20, 21). In all of these studies, the authors set the final screening concentration at a constant value for all the metabolites (5 μ M in the studies of Conlon et al. and Liu et al., 10 μ M in those of Chen et al. and Oh et al.). In these circumstances, although the actual ferroptosis-related effects of some metabolites were hidden by their own cytotoxicity and the ratio-based algorithm, such information loss is inevitable and acceptable in the high-throughput library screening assay, since the ideal candidates, arginine and its downstream metabolites, were still screened out. In our subsequent studies, we will re-analyze the primary screening data and lower the volume of those cytotoxic metabolites to investigate further whether they could influence cell’s

response to ferroptosis in an appropriate concentration.

Since the metabolite library HY-L030 used in our study contains 889 members, we cannot ensure that all the metabolites are permeable to the membrane of A549 cells. However, those impermeable metabolites would not interfere with the screening result because they could neither enter the cells nor alter cells' metabolic pattern, much less to regulate ferroptosis. We would not select such a metabolite for further exploration because its impermeability would bring the in vitro and in vivo assays with lots of unnecessary trouble, and it would be extremely difficult to investigate their intracellular metabolism and therapeutic significance without exogenously adding the metabolites into the cultural system. Nevertheless, we cannot exclude the possibility that their intracellular metabolism might impact ferroptosis, and we would trace this evidence by focusing on its upstream or downstream metabolites, which are permeable to the membrane.

In the above-mentioned studies, the researchers added metabolites and ferroptosis-inducers simultaneously. However, in the ferroptosis-related research carried out in our group, we preferred to pre-treated the cells with indicated metabolites for several hours (in the range of 2 to 24 h, depending on the experimental type) to make sure the metabolites have enough time to enter the cells and make a difference, especially when the RSL3-treating time is quite brief (8 h). Therefore, we pre-treated the cells with metabolite library for 12 hours before the addition of RSL3.

Comment (2).

• This concentration of polyamines are cytotoxic on the cells regardless of ferroptosis/RSL3 – is this not just an additive effect?

Answer:

Thanks very much for your comment. In most of the in vitro assays carried out in the present research, such as cell viability and lipid peroxidation, we treated A549 or HT1080 cells with 10 μ M polyamines for 4 h before the addition of ferroptosis inducers RSL3 and IKE. Such concentration of polyamine did not significantly decrease cells' viability in RSL3-related assays since the incubation time of polyamine is only 10-12 hours in total (4 hours for polyamine pretreatment and 6-8 hours for RSL3 treatment). In A549-IKE assays, the incubation time of IKE was extended to 48 h, and here, polyamine slightly decreased cells' viability.

Generally, the cytotoxicity of polyamines derives from the copper-dependent serum amine oxidase contained in bovine serum, which catalyzes exogenous polyamines into ammonia and aldehydes. Therefore, to exclude the involvement of this effect when investigating the ferroptosis-sensitizing property, we added aminoguanidine, an inhibitor of the serum amine oxidase, to the cultural media and found that the ferroptosis-sensitizing effect of polyamines still existed (Figure S2g).

(Results: Page 6, line 164)

"...(Figure 2f, S2f). These findings firmly demonstrate the role of endogenous metabolite, polyamines, in promoting lipid peroxidation and associated ferroptotic cell death. Previous researchers have demonstrated that polyamines could be catalysed into ammonia and aldehyde, which are toxic to cells, by the amine oxidase contained in bovine serum. However, on the one hand, at the low concentration (5-10 μ M) commonly used in this study, polyamines did not exert significant negative impact on the cell viability when administered alone; on the other hand, we also ran the experiments with aminoguanidine, an inhibitor of the serum amine oxidase, and found that the ferroptosis-sensitizing effect of polyamines was identical to the results above (Figure S2g).

Therefore, we believed that polyamine's ferroptosis-sensitizing could not be explained by the exogenous catalysis from amine oxidase.”

(Figure S2g)

Besides, since the “relative viability” indicated in all viability assays was calculated as $\text{Viability}_{(\text{RSL3+polyamines})} / \text{Viability}_{(\text{DMSO+polyamines})}$, even though exogenously supplemented polyamines exhibited some cytotoxicity on cells, this confounding effect would be eliminated in the calculation of final results, which means the synergistic cytotoxicity of “polyamine+RSL3” is much stronger than the sum of their separate effects.

Comment (3).

- Need a greater explanation of how the RSL3-A549 cell model works.

Answer:

Thanks very much for your comment. We further explained the working mechanism of RSL3 in the Result section and described the RSL3-A549 cell model used in the metabolite library screening in Figure 1 and the corresponding legend. Please find them below.

(Results: Page 4, line 99)

“To discover metabolites potentially involved in the process of ferroptosis, an RSL3-induced A549 cell ferroptosis model was adopted to screen a library containing 889 human endogenous metabolites (Figure 1a). As one of the most commonly used class II FINS, RSL3 directly inhibits the catalytic activity of GPX4 by covalently binding to the selenocysteine residue of GPX4 through their electrophile chloroacetamide moiety. Screening of this library identified arginine...”

(Revised Figure 1)

(Figure legend, Page 38, line 1184)

“Figure 1. Metabolite library screening links arginine and downstream ornithine to ferroptosis. a, Schematic of the metabolite library screening strategy. A549 cells seeded in 96-well plates were pre-treated with indicated metabolites (50 μM) followed by RSL3 treatment. Cell viability was assessed using CCK8 assay. b, The metabolite library screening results were exhibited as the relative viability of metabolite-treated cells versus vehicle-treated cells. After cell seeding,

the A549 cells were pre-incubated with the metabolites contained in the library (MCE HY-L030) or Vehicle (DMSO, 50 µM) for 12 hours. Then, the cells were further treated with 4 µM RSL3 or DMSO for 8 h and the cell viability was measured using CCK8. In the algorithm for the “Relative viability”, “X” represents the pre-treatment of a specific metabolite, and “V” represents the pre-treatment of Vehicle DMSO.”

Comment (4).

• Fig 2e – the amplification of ferroptotic morphological features upon pre-treatment with polyamines is not obvious from the images – can it be quantified?

Answer:

Thanks very much for your comment. We consulted a series of high-quality ferroptosis-related research in which transmission electron microscopy (TEM) was used to visualize the morphological change of mitochondria in ferroptosis cells, but failed to find a reliable approach to quantify the level of such change as shown below. Typically, ferroptosis is characterized by shrunken dense mitochondria, reduced numbers of mitochondrial cristae, and increased membrane density⁽¹⁾.

(1) Lei G, et al. The role of ferroptosis in ionizing radiation-induced cell death and tumor suppression. *Cell Res.* 2020 Feb;30(2):146-162.⁽²²⁾

(2) Lee H, et al. Energy-stress-mediated AMPK activation inhibits ferroptosis. *Nat Cell Biol.* 2020 Feb;22(2):225-234.⁽²³⁾

(3) Bi G, et al. Retinol Saturase Mediates Retinoid Metabolism to Impair a Ferroptosis Defense System in Cancer Cells. *Cancer Res.* 2023 Jul 14;83(14):2387-2404.⁽²⁴⁾

Therefore, as the Reviewer kindly suggested, we added arrows indicating mitochondria in our TEM images to make the ferroptosis-induced morphological change more understandable for the audiences. As we can see in the image, a single administration of polyamines did not generate any visible change, but RSL3 treatment led to mitochondrial shrinkage and cristae loss, as well as increased membrane density, and polyamine-treatment further amplified these changes. Please find it below.

(Revised Figure 2e)

Comment (5).

- Why pre-treatment in Fig 2.

Answer:

Thanks very much for your comment. When investigating a metabolite or drug's impact on ferroptosis, it is quite common to pretreat the cells with it for several hours before the addition of ferroptosis inducers to provide the metabolites with enough time to enter the cells and make a difference, as suggested in multiple high-quality ferroptosis-related studies(25-27). Especially, in our research, the incubating time of RSL3 is only 8 hours, further emphasizing the importance of pre-treatment process, given that the polyamines need some hours to be absorbed by the cell and metabolized into H₂O₂, and this process must be accomplished before RSL3 kill the cells. In preliminary experiments, we tested several pre-treatment time points and found that 4 hours would be enough for the polyamines to exhibit considerable ferroptosis-sensitizing effect in A549 and HT1080 cells, as shown below.

Comment (6).

• Fig 4C – protein levels should be quantified as the increase in ODC1 expression is not obvious

Answer:

Thanks very much for your comment. We optimized our western blot protocol and quantified the ODC1 expression level (normalized by GAPDH) in Figure 4c using the software ImageJ as the Reviewer kindly suggested. Please check it.

(Revised Figure 4c)

Comment (7).

• In Fig 4C Fer-1 also reduced ODC1 expression in comparison to DMSO – therefore not convinced iron overload alone is the reason for ODC1 overexpression.

Answer:

Thanks very much for your comment. In Figure 4b and 4c, we noticed that Fer-1 also slightly inhibited RSL3-induced ODC1 upregulation (both mRNA and protein levels) compared with the vehicle group, although not as significant as DFO. As an antioxidant, Fer-1 inhibits ferroptosis by scavenging lipid peroxides rather than blocking liable iron and the corresponding iron-dependent Fenton reaction.

Multiple theories for ferroptosis-induced iron overload have been proposed yet, such as the release of GSH-bound ferrous^(11, 12), overexpression of TFRC, downregulation of iron exporter SLC40A1⁽¹³⁾, and ferritinophagy, an autophagic process in which cellular iron storage proteins like ferritin are degraded and the liable iron (ferrous) are accumulated^(14, 15). Therefore, iron overload could be triggered by several different events that occurred during the initiation, propagation, and termination of ferroptosis process. Meanwhile, as we can see in the vehicle group (without DFO or RSL3) exhibited in Figure 4b-c, incubation with 0.2 μM RSL3 led to a higher iron level compared to 0.1 μM group, suggesting the extent of iron overload was also partly determined by the severity of ferroptosis.

Together, although Fer-1 does not directly chelate iron as DFO does, the level of RSL3-induced iron accumulation might also be slightly suppressed by the absence of lipid peroxidation and ferroptotic cell death in Fer-1 group compared with the vehicle group. And of course, as the Reviewer kindly suggested, we cannot exclude the existence of other unknown mechanisms

accounting for ferroptosis-induced ODC1 expression. We mentioned these in the revised manuscript. Please find them in the revised manuscript.

(Results, Page 10, line 267)

“This unexpected finding prompted us to associate ODC1 regulation with the difference between the mechanisms of DFO and Fer-1’s ferroptotic inhibiting function. DFO, an iron chelator, inhibits ferroptosis by blocking liable iron and the corresponding iron-dependent Fenton reaction, whereas Fer-1, an antioxidant, scavenges lipid peroxides, cannot completely mitigate iron overload as DFO does, an important initiation signal of ferroptosis. We confirmed DFO, Fer-1, and H₂O₂’s impact on RSL3-induced iron overload by staining intracellular liable iron with FerroOrange (Figure 4f, S3i). Similar results...”

(Discussion, Page 19, line 535)

“...regulation of ferroptosis. Besides, Fer-1 also slightly abrogated RSL3-induced ODC1 upregulation. On the one hand, although Fer-1 does not directly chelate iron as DFO does, the level of RSL3-induced iron accumulation might also be slightly suppressed by the absence of lipid peroxidation and ferroptotic cell death in Fer-1 group. On the other hand, we cannot exclude the existence of other unknown mechanisms accounting for this phenomenon.”

Comment (8).

- Fig 4l-m – these panels jump about a bit in the figure, they do not follow a logical order.

Answer:

Thanks very much for your comment. We rearranged these figures into a logical order as the Reviewer kindly suggested.

Comment (9).

- Not clear how specific the effect is on Wnt/Myc – as this is the only pathway tested here. Myc is regulated by other pathways other than Wnt signalling.

Answer:

Thanks very much for your comment. It has been reported that iron activated the WNT signalling pathway, and MYC, one of the most important WNT downstream target genes, serves as a direct transcription factor of ODC1⁽²⁸⁾. Therefore, in the present research we hypothesized and validated that iron overload stimulates ODC1 expression through the WNT/MYC signalling pathway. We first confirmed RSL3/FAC-induced MYC expression and WNT activation-induced ODC1 expression and ferroptosis sensitization (Figure 4k-n). Furthermore, we found that RSL3/FAC induce ODC1 expression was largely abrogated by WNT antagonist LF3 (Figure 4s). Considering that the oncogene MYC could be regulated by several biological pathways, we further investigated specific regulating effect of WNT/MYC signalling as the Reviewer kindly suggested. On the one hand, as shown in revised Figure 4m, we validated that WNT agonist SKL2001 and antagonist LF3 significantly promoted and inhibited MYC expression, respectively. On the other hand, block WNT signalling by LF3 largely abrogated RSL3/FAC-induced MYC expression (revised Figure 4s). Consistently, siRNA-mediated MYC knockdown decreased ODC1 mRNA level and blocked RSL3/FAC-induced ODC1 upregulation (Figure S4f-g). Therefore, these findings indicated the dominant role of WNT signalling in the proposed iron-MYC/ODC1 pathway. However, considering that WNT antagonist LF3 could only largely, but not completely, abrogate RSL3/FAC-

triggered MYC/ODC1 induction, suggesting the existence of other unknown mechanisms accounting for this phenomenon, although they are not the dominant factor. We mentioned these in the Results section. Please find them below.

(Results, Page 10, line 281)

“Iron not only serves as an essential nutrient that facilitates cell proliferation but also participates in several canonical signalling pathways. Notably, the link between iron and WNT signalling has been implicated in several cancer-related research. Generally, in cells with activated WNT signalling, β -catenin enters the nucleus and promotes T cell factor (TCF)-lymphoid enhancer factor (LEF)-dependent transcription of a series of downstream target genes, such as MYC, whose direct transcriptive targets include ODC1. Thus, it is reasonable to speculate that iron overload stimulates ODC1 expression through the WNT/MYC signalling pathway. To verify this hypothesis, we first treated A549 and HT1080 cells with FAC and RSL3 to induce iron overload and detected the activation level of WNT signalling using MYC as a specific indicator. As exhibited in Figure 4j-k, both of them significantly increased the mRNA and protein levels of MYC, whereas DFO supplementation completely abolished this alteration. Furthermore, we noticed that treating cells with WNT-specific agonist SKL2001 substantially upregulated ODC1 and MYC levels, while WNT inhibitor LF3 generated the opposite result (Figure 4l-m). Consistently, we further observed that SKL2001 and LF3 treatment respectively sensitized and desensitized cells to FINs in a concentration-dependent manner, suggesting WNT signalling pathway impairs cells' ferroptosis resistance (Figure 4n-o, S4a-b). SKL2001 caused ferroptosis-sensitization could be fully rescued by Fer-1 and DFO (Figure S4c). Besides, to ascertain whether MYC directly activates ODC1's transcription, we took advantage of the ENCODE database and confirmed the enrichment of MYC at the promoter region of ODC1 gene (Figure S4c). Multiple potential MYC binding sites (BS) were also identified through JASPAR (Figure S4d). In agreement with these predictions and previous reports, our CHIP assay revealed significant enrichment of MYC to BS2 and BS3 (Figure 4p). To further confirm the binding specificity, a luciferase reporter gene assay was performed using plasmids containing ODC1 promoter region (WT) and introduced mutations in the MYC binding sites (MUT). As shown in Figure 4q, both FAC and SKL2001 treatment resulted in extensive transcriptional activation, while the iron chelator DFO led to inhibition, as reflected by the luciferase intensity in cells transfected with WT but not the all-MUT plasmid. Single mutation at the two binding sites also mitigated these changes to varying degrees. siRNA-mediated MYC knockdown also decreased ODC1 mRNA level and blocked RSL3/FAC induced ODC1 upregulation (Figure S4e-f). These data demonstrate that MYC directly activates ODC1 transcription. Moreover, pre-treatment with WNT inhibitor LF3 largely, but not completely abrogated FAC or RSL3-induced MYC/ODC1 upregulation, suggesting the existence of other unknown mechanisms accounting for this phenomenon (Figure 4r-s), although they are not the dominant factor.”

(Revised Figure 4k-s)

(Revised Figure S4f-g)

Comment (10).

• The effects of addition of the media from ferroptotic cells vs normal cells in Fig 5c-d are significant but minimal – how biologically relevant are these small changes?

Answer:

Thanks very much for your comment. In Figure 5d, the relative viability of the two groups indicated by the Reviewer are 0.394 ± 0.027 (Plate A) and 0.487 ± 0.022 (Plate B), respectively. Although the difference in absolute values is not that big, compared with plate B, ferroptotic cell-derived medium led to a reduction of 20% in relative viability.

Theoretically, in real tumour tissue, or even a layer of cells attached to the bottom of the culture dish, the tumour cells are close to each other and could easily uptake the extracellular vehicles released from adjacent cells suffering from ferroptotic stress. In contrast, in the cell model depicted in Figure 5, we used culture dishes with a diameter of 6 cm and added 4 mL medium to each dish. So, the polyamines-containing extracellular vehicles released from ferroptotic cells are diluted in the medium, thus leading to a relatively low concentration and preventing us from observing a larger difference. Further in vivo validation of the polyamine-containing EVs-mediated ferroptosis amplification is still warranted in subsequent research.

Comment (11).

• It is important to show the specificity of the samples for EVs in Fig 5f – need to blot with a

specific EV marker

Answer:

Thanks very much for your comment. For the selection of EV marker proteins, we consulted the definitive guideline in this area: Minimal information for studies of extracellular vesicles 2018 (MISEV2018) ⁽²⁹⁾, as well as several extracellular vesicles related high-quality studies^(9, 30-32). Generally, it is recommended that three positive protein markers of EVs, including at least one transmembrane or GPI-anchored proteins protein (CD63, CD81, CD9), one cytosolic protein (TSG101, Alix), and one negative protein marker associated to other cellular compartments (secretory GM130 or calnexin, mitochondrial TOMM20, and nuclear Lamin A). Therefore, we adopted CD63, CD81, TSG101, and GM130 for the determination of EV's specificity as the Reviewer kindly suggested. Please find revised Figure 5f.

(Revised Figure 5f)

Comment (12).

• Again the effects in Fig 5g are minimal and difficult to understand whether this would have biological significance.

Answer:

Thanks very much for your comment. We optimized the EV isolation/purification protocols and reperformed the cell viability assays as the Reviewer kindly suggested and received a better result. Please find them below.

(Revised Figure 5g)

Reference in Response Letter

1. Lei G, Zhuang L, Gan B. Targeting ferroptosis as a vulnerability in cancer. Nature Reviews Cancer. 2022.
2. Stockwell BR. Ferroptosis turns 10: Emerging mechanisms, physiological functions, and therapeutic applications. Cell. 2022;185(14):2401-21.

3. Tang D, Chen X, Kang R, Kroemer G. Ferroptosis: molecular mechanisms and health implications. *Cell Res.* 2021;31(2):107-25.
4. Zhang F, Hu G, Chen X, Zhang L, Guo L, Li C, Zhao H, Cui Z, Guo X, Sun F, et al. Excessive branched-chain amino acid accumulation restricts mesenchymal stem cell-based therapy efficacy in myocardial infarction. *Signal Transduction and Targeted Therapy.* 2022;7(1):171.
5. Conlon M, Poltorack CD, Forcina GC, Armenta DA, Mallais M, Perez MA, Wells A, Kahanu A, Magtanong L, Watts JL, et al. A compendium of kinetic modulatory profiles identifies ferroptosis regulators. *Nat Chem Biol.* 2021;17(6):665-74.
6. Rodriguez PC, Ochoa AC, Al-Khami AA. Arginine Metabolism in Myeloid Cells Shapes Innate and Adaptive Immunity. *Frontiers In Immunology.* 2017;8:93.
7. Bronte V, Zanovello P. Regulation of immune responses by L-arginine metabolism. *Nat Rev Immunol.* 2005;5(8):641-54.
8. Gaschler MM, Andia AA, Liu H, Csuka JM, Hurlocker B, Vaiana CA, Heindel DW, Zuckerman DS, Bos PH, Reznik E, et al. FINO2 initiates ferroptosis through GPX4 inactivation and iron oxidation. *Nat Chem Biol.* 2018;14(5):507-15.
9. Brown CW, Amante JJ, Chhoy P, Elaimy AL, Liu H, Zhu LJ, Baer CE, Dixon SJ, Mercurio AM. Prominin2 Drives Ferroptosis Resistance by Stimulating Iron Export. *Dev Cell.* 2019;51(5).
10. Chen X, Huang J, Yu C, Liu J, Gao W, Li J, Song X, Zhou Z, Li C, Xie Y, et al. A noncanonical function of EIF4E limits ALDH1B1 activity and increases susceptibility to ferroptosis. *Nature Communications.* 2022;13(1):6318.
11. Patel SJ, Frey AG, Palenchar DJ, Achar S, Bullough KZ, Vashisht A, Wohlschlegel JA, Philpott CC. A PCBP1-BolA2 chaperone complex delivers iron for cytosolic [2Fe-2S] cluster assembly. *Nat Chem Biol.* 2019;15(9):872-81.
12. Patel SJ, Protchenko O, Shakoury-Elizeh M, Baratz E, Jadhav S, Philpott CC. The iron chaperone and nucleic acid-binding activities of poly(rC)-binding protein 1 are separable and independently essential. *Proc Natl Acad Sci USA.* 2021;118(25).
13. Song X, Xie Y, Kang R, Hou W, Sun X, Epperly MW, Greenberger JS, Tang D. FANCD2 protects against bone marrow injury from ferroptosis. *Biochem Biophys Res Commun.* 2016;480(3):443-9.
14. Gao M, Monian P, Pan Q, Zhang W, Xiang J, Jiang X. Ferroptosis is an autophagic cell death process. *Cell Research.* 2016;26(9):1021-32.
15. Hou W, Xie Y, Song X, Sun X, Lotze MT, Zeh HJ, Kang R, Tang D. Autophagy promotes ferroptosis by degradation of ferritin. *Autophagy.* 2016;12(8):1425-8.
16. Chen X, Kang R, Kroemer G, Tang D. Broadening horizons: the role of ferroptosis in cancer. *Nat Rev Clin Oncol.* 2021.
17. Dixon SJ, Pratt DA. Ferroptosis: A flexible constellation of related biochemical mechanisms. *Molecular Cell.* 2023;83(7):1030-42.
18. Dixon SJ, Lemberg KM, Lamprecht MR, Skouta R, Zaitsev EM, Gleason CE, Patel DN, Bauer AJ, Cantley AM, Yang WS, et al. Ferroptosis: an iron-dependent form of nonapoptotic cell death. *Cell.* 2012;149(5):1060-72.
19. Aron AT, Loehr MO, Bogena J, Chang CJ. An Endoperoxide Reactivity-Based FRET Probe for Ratiometric Fluorescence Imaging of Labile Iron Pools in Living Cells. *J Am Chem Soc.* 2016;138(43):14338-46.
20. Oh M, Jang SY, Lee J-Y, Kim JW, Jung Y, Kim J, Seo J, Han T-S, Jang E, Son HY, et al. The lipoprotein-associated phospholipase A2 inhibitor Darapladib sensitises cancer cells to ferroptosis by

- remodelling lipid metabolism. *Nature Communications*. 2023;14(1):5728.
21. Liu D, Liang C-H, Huang B, Zhuang X, Cui W, Yang L, Yang Y, Zhang Y, Fu X, Zhang X, et al. Tryptophan Metabolism Acts as a New Anti-Ferroptotic Pathway to Mediate Tumor Growth. *Adv Sci (Weinh)*. 2023:e2204006.
 22. Lei G, Zhang Y, Koppula P, Liu X, Zhang J, Lin SH, Ajani JA, Xiao Q, Liao Z, Wang H, et al. The role of ferroptosis in ionizing radiation-induced cell death and tumor suppression. *Cell Res*. 2020;30(2):146-62.
 23. Lee H, Zandkarimi F, Zhang Y, Meena JK, Kim J, Zhuang L, Tyagi S, Ma L, Westbrook TF, Steinberg GR, et al. Energy-stress-mediated AMPK activation inhibits ferroptosis. *Nat Cell Biol*. 2020;22(2):225-34.
 24. Bi G, Liang J, Shan G, Bian Y, Chen Z, Huang Y, Lu T, Li M, Besskaya V, Zhao M, et al. Retinol saturase mediates retinoid metabolism to impair a ferroptosis defense system in cancer cells. *Cancer Research*. 2023.
 25. Koppula P, Lei G, Zhang Y, Yan Y, Mao C, Kondiparthi L, Shi J, Liu X, Horbath A, Das M, et al. A targetable CoQ-FSP1 axis drives ferroptosis- and radiation-resistance in KEAP1 inactive lung cancers. *Nature Communications*. 2022;13(1):2206.
 26. Mishima E, Ito J, Wu Z, Nakamura T, Wahida A, Doll S, Tonnus W, Nepachalovich P, Eggenhofer E, Aldrovandi M, et al. A non-canonical vitamin K cycle is a potent ferroptosis suppressor. *Nature*. 2022;608(7924):778-83.
 27. Bersuker K, Hendricks JM, Li Z, Magtanong L, Ford B, Tang PH, Roberts MA, Tong B, Maimone TJ, Zoncu R, et al. The CoQ oxidoreductase FSP1 acts parallel to GPX4 to inhibit ferroptosis. *Nature*. 2019;575(7784):688-92.
 28. Holbert CE, Cullen MT, Casero RA, Stewart TM. Polyamines in cancer: integrating organismal metabolism and antitumour immunity. *Nature Reviews Cancer*. 2022;22(8):467-80.
 29. Théry C, Witwer KW, Aikawa E, Alcaraz MJ, Anderson JD, Andriantsitohaina R, Antoniou A, Arab T, Archer F, Atkin-Smith GK, et al. Minimal information for studies of extracellular vesicles 2018 (MISEV2018): a position statement of the International Society for Extracellular Vesicles and update of the MISEV2014 guidelines. *J Extracell Vesicles*. 2018;7(1):1535750.
 30. Liang W, Sagar S, Ravindran R, Najor RH, Quiles JM, Chi L, Diao RY, Woodall BP, Leon LJ, Zumaya E, et al. Mitochondria are secreted in extracellular vesicles when lysosomal function is impaired. *Nature Communications*. 2023;14(1):5031.
 31. Liu D-A, Tao K, Wu B, Yu Z, Szczepaniak M, Rames M, Yang C, Svitkina T, Zhu Y, Xu F, et al. A phosphoinositide switch mediates exocyst recruitment to multivesicular endosomes for exosome secretion. *Nature Communications*. 2023;14(1):6883.
 32. Kim B, Kang Y-T, Mendelson FE, Hayes JM, Savelieff MG, Nagrath S, Feldman EL. Palmitate and glucose increase amyloid precursor protein in extracellular vesicles: Missing link between metabolic syndrome and Alzheimer's disease. *J Extracell Vesicles*. 2023;12(11):e12340.

Reviewers' Comments:

Reviewer #1:

Remarks to the Author:

The authors have carefully addressed the concerns with substantial details and new data added which meets the standard for acceptance.

Reviewer #2:

Remarks to the Author:

The authors did extra experiments, the revision looks much better. But the revision did not fully address my concerns. The key conclusion "polyamine supplementation also sensitizes cancer cells or xenograft tumors to radiotherapy or chemotherapy through inducing ferroptosis" is mostly based on the combination usage of IKE and spermine (Figure 6). Thus, although most ferroptosis inhibition experiments were finished in RSL3-induced system, ferroptosis inhibition experiments were still needed to be checked in IKE-induced system. In addition, most of ferroptotic killing and rescue experiments, the effects are so mild.

Reviewer #3:

Remarks to the Author:

The authors have adequately addressed most of my questions.

Regarding if iron overload is THE initial ferroptosis signal, while iron overload can be an initiating ferroptotic signal, as shown in many studies cited by the authors, no evidence, as far as this reviewer is aware, has been published to support that this is universally accurate or true.

Reviewer #4:

None

Response to Reviewer Comments

Dear Editors and Reviewers,

We appreciate your valuable comments and suggestions for our manuscript titled “Polyamine-mediated ferroptosis amplification acts as a targetable vulnerability in cancer” (Manuscript number: NCOMMS-23-37087A). We are grateful for the detailed comments and suggestions provided by each Reviewer, and we believe that their input has greatly improved our manuscript. In this letter, we include their comments verbatim in blue, followed by our comments and revisions. We use quotation marks and *Italic* to mark the sentences cited from the manuscript and use **Yellow Background** for the revised messages. Although this makes a rather lengthy letter, we believe that it will provide you and the Reviewers with the best explanation of the changes that we have made.

REVIEWER COMMENTS

Reviewer #1 (Remarks to the Author):

The authors have carefully addressed the concerns with substantial details and new data added which meets the standard for acceptance.

Reviewer #2 (Remarks to the Author):

The authors did extra experiments, the revision looks much better. But the revision did not fully address my concerns. The key conclusion “polyamine supplementation also sensitizes cancer cells or xenograft tumors to radiotherapy or chemotherapy through inducing ferroptosis” is mostly based on the combination usage of IKE and spermine (Figure 6). Thus, although most ferroptosis inhibition experiments were finished in RSL3-induced system, ferroptosis inhibition experiments were still needed to be checked in IKE-induced system. In addition, most of ferroptotic killing and rescue experiments, the effects are so mild.

Reviewer #3 (Remarks to the Author):

The authors have adequately addressed most of my questions.

Regarding if iron overload is THE initial ferroptosis signal, while iron overload can be an initiating ferroptotic signal, as shown in many studies cited by the authors, no evidence, as far as this reviewer is aware, has been published to support that this is universally accurate or true.

To Reviewer #1:

Comment (1).

• The authors have carefully addressed the concerns with substantial details and new data added which meets the standard for acceptance.

Answer:

Thanks very much for your recognition. Your kind suggestions have greatly improved our manuscript.

To Reviewer #2:

Comment (1).

• The authors did extra experiments, the revision looks much better. But the revision did not fully address my concerns. The key conclusion “polyamine supplementation also sensitizes cancer cells or xenograft tumors to radiotherapy or chemotherapy through inducing ferroptosis” is mostly

based on the combination usage of IKE and spermine (Figure 6). Thus, although most ferroptosis inhibition experiments were finished in RSL3-induced system, ferroptosis inhibition experiments were still needed to be checked in IKE-induced system. In addition, most of ferroptotic killing and rescue experiments, the effects are so mild.

Answer:

Thanks very much for your comment. Since the Ethics Committee of our institute only approved the use of one type of ferroptosis inducer in nude mice xenograft assays to reduce the number of experimental animals sacrificed, we preferred IKE, an erastin analog with improved potency and metabolic stability that is suitable for animal studies as suggested by several high-quality studies (1-6). In the preparation of this manuscript, most ferroptotic killing experiments, or cytotoxicity tests, were conducted in both RSL3 and IKE-induced systems in multiple cell lines. However, considering the length limit of submission, we only exhibited part of them in the initial version. As the Reviewer kindly suggested, we added these IKE-related results into the revised manuscript, especially in the demonstration of arginine and polyamines' ferroptosis-promoting effect, as well as the roles of key-enzymes of polyamine metabolism in this process. We believe that the data obtained from two types of ferroptosis inducers targeting different mechanisms (RSL3 blocks GPX4, while IKE blocks SLC7A11) would make our conclusion more convincing. Please check these data below.

Besides, in the preliminary assays of this research, we adopted the optimal experimental conditions (such as the dose and time of drug treatment) that could best reflect spermine's ferroptosis-promoting effect, and the condition was then used in the whole research to guarantee the consistency. Although this condition might not be the best one for some other assays, statistically significant differences were observed, suggesting the robustness of our findings. In future research we would further optimize our experimental protocols to overcome this problem as the Reviewer kindly suggested. We believe the Reviewer's valuable comments would greatly improve our manuscript. Thank you very much for your helping.

(Results: Page 4, line 105)

“We further validated this finding in both A549 and HT1080 cells, as well as three non-small cell lung cancer cell lines including H1299, H23, and PC9, by deleting or exogenously supplementing arginine into the culture medium (Figure 1c-d, S1a). Since...”

(Revised Figure 1c)

(Revised Figure S1a)

(Results: Page 4, line 110)

“Upregulation of prostaglandin-endoperoxide synthase 2 (PTGS2, a biomarker of ferroptosis mRNA was also detected in arginine-treated cells (Figure S1b). Moreover, arginine-caused RSL3 and IKE sensitization could be fully rescued by the ferroptosis inhibitor ferrostatin-1 (Fer-1, lipid peroxidation scavenger) and deferoxamine (DFO, iron chelator), but not by apoptosis inhibitor z-VAD(OMe)-FMK or necroptosis inhibitor (necrosulfonamide), confirming the specific promoting role of arginine in ferroptosis (Figure S1c). These data...”

(Revised Figure S1b)

(Revised Figure S1c)

(Results: Page 5, line 134)

“To understand whether arginine impairs cells' resistance to ferroptosis by converting to ornithine, we performed cytotoxicity assay and observed that exogenously supplemented ornithine generated similar ferroptosis sensitizing effect to arginine, whereas the by-product urea failed to do so. Additionally, citrulline also slightly promoted ferroptosis (Figure 1h, S1d-e). This phenomenon...”

(Revised Figure S1d)

(Revised Figure S1e)

(Results: Page 5, line 139)

“Furthermore, CRISPR/Cas9-mediated ARG2 knockout (KO) not only led to a detectable accumulation of arginine and decreased formation of ornithine but also inhibited FINs-induced cell death, as well as abolished the ferroptosis-promoting effect conferred by arginine supplementation (Figure 1i-k, S1f). These findings...”

(Revised Figure S1f)

(Results: Page 6, line 149)

“Notably, proline did not affect cells’ response to FINs, whereas polyamines displayed stronger ferroptosis sensitizing effect than arginine even in a low concentration without markedly impacting cell’s viability when solely administered (spermine > spermidine > putrescine), which was consistent with our initial metabolites screening results (Figure 1b, 2a, S2a-b). The broad-spectrum...”

(Revised Figure 2a)

(Revised Figure S2a)

(Revised Figure S2b)

(Results: Page 6, line 157)

“Besides, pre-treatment with polyamine transport inhibitor AMXT-1501 enhanced cells’ resistance to RSL3/IKE and blocked polyamines’ ferroptosis sensitizing effect (Figure 2c-d, S2e).

Moreover...”

(Revised Figure 2c)

(Revised Figure S2e)

(Results: Page 6, line 162)

“The ferroptosis-sensitizing effect of polyamines was fully rescued by Fer-1 and DFO, but not by z-VAD(OMe)-FMK or necrosulfonamide, confirming that polyamine specifically promotes ferroptosis (Figure 2f, S2f). These findings...”

(Revised Figure S2f)

(Results: Page 7, line 178)

“In agreement with the CTRP results, the depletion of ODC1 not only protected cells from FINs-induced cell death but also completely abrogated arginine’s pro-ferroptotic effect (Figure 2h-i, S2h). Sensitivity to FINs in ODC1-KO cells was restored by its re-expression, thus excluding the off-target effects of CRISPR/Cas9 mediated ODC1 knockout (Figure 2j-k, S2i). Meanwhile, pharmacological inhibition of ODC1 using difluoromethylornithine (DFMO) generated similar effect with ODC1 knockout (Figure 2l, S2j). Supportively...”

(Revised Figure S2h)

(Revised Figure S2i)

(Revised Figure S2j)

(Results: Page 8, line 207)

“To understand whether polyamine sensitizes cells to ferroptosis via its metabolic process, we designed a series of siRNAs targeting these genes and found that knockdown of these polyamine metabolism-related genes all more or less protected A549 and HT1080 from FINs-induced cell death, among which knockdown of PAOX and SMOX exhibited most potent protective effect (Figure 3a, S3d-e). Considering that...”

(Revised Figure S3e)

(Results: Page 8, line 216)

“To verify this hypothesis, we generated PAOX/SMOX double-knockout cell lines (Figure 3b). As expected, PAOX/SMOX-KO completely abrogated polyamines-induced ferroptosis sensitization (Figure 3c, S3f). Furthermore, we depleted intracellular polyamines by simultaneously targeting their synthesis by DFMO and uptake by AMXT-1501. In these polyamine-depleted cells, PAOX/SMOX-KO failed to confer resistance to RSL3/IKE any more (Figure 3d, S3g). These results...”

(Revised Figure S3f)

(Revised Figure S3g)

(Results: Page 8, line 228)

“The expressed Δ catalase significantly desensitized cells to RSL3 and IKE, and restored polyamine-treated cells’ resistance (Figure 3h, S3i). Meanwhile, the ferroptosis-inhibiting effect conferred by PAOX/SMOX-KO was also abrogated by Δ catalase expression (Figure 3i, S3j). Taken together...”

(Revised Figure S3i)

(Revised Figure S3j)

(Results: Page 10, line 293)

“Consistently, we further observed that SKL2001 and LF3 treatment respectively sensitized and desensitized cells to FINs in a concentration-dependent manner, suggesting WNT signalling pathway impairs cells’ ferroptosis resistance (Figure 4n-o, S4a-b). SKL2001...”

(Figure S4a)

To Reviewer #3:

Comment (1).

• The authors have adequately addressed most of my questions. Regarding if iron overload is THE initial ferroptosis signal, while iron overload can be an initiating ferroptotic signal, as shown in many studies cited by the authors, no evidence, as far as this reviewer is aware, has been published to support that this is universally accurate or true.

Answer:

Thanks very much for your recognition. After review of published studies and discussion with my colleagues, I agree with the Reviewer's view that using the term "initial ferroptosis signal" to describe "iron overload" is not precise enough since the initiation of ferroptosis is quite a complicated process involving multiple factors like ROS, PUFA-PL, and iron. We revised our manuscript by replacing "initial" with "important" or just deleting it as the Reviewer kindly suggested. Please check them below.

(Introduction: Page 2, line 37)

"Notably, the expression of ODC1, the critical enzyme catalysing polyamine synthesis, was significantly activated by the ferroptosis signal—iron overload—through WNT/MYC signalling, as well as the subsequent elevated polyamine synthesis in ferroptosis process, thus forming a ferroptosis-iron overload-WNT/MYC-ODC1-polyamine-H₂O₂ positive feedback loop that amplifies ferroptosis level. Meanwhile..."

(Introduction: Page 4, line 90)

"Polyamine synthesis is induced by the important ferroptosis signal—iron overload through WNT/MYC signalling, thus forming a positive feedback axis that amplifies ferroptosis. Consistently..."

(Results: Page 11, line 314)

"Combined this conclusion with the pro-ferroptotic role of ODC1 and corresponding polyamine synthesis mentioned above, these findings demonstrate that ODC1 sensed the important ferroptosis signal—iron overload in a WNT/MYC dependent manner, and then catalysed the synthesis of polyamine, a potent ferroptosis mediator, thus forming a positive feedback loop consisting of ferroptosis-iron-WNT/MYC-ODC1-polyamine-ferroptosis, which may function as a core ferroptosis amplifier."

(Results: Page 15, line 430)

“Mechanically, the important ferroptosis signal—iron overload—triggers the transcription of ODC1, the critical enzyme catalysing polyamine synthesis, through WNT/MYC signalling, thus forming a ferroptosis-iron overload-WNT/MYC/ODC1-polyamine-H₂O₂ positive feedback axis that amplifies ferroptosis. Moreover...”

Reference

1. Bi G, Liang J, Shan G, Bian Y, Chen Z, Huang Y, Lu T, Li M, Besskaya V, Zhao M, et al. Retinol Saturase Mediates Retinoid Metabolism to Impair a Ferroptosis Defense System in Cancer Cells. *Cancer Research*. 2023;83(14):2387-404.
2. Zhang Y, Tan H, Daniels JD, Zandkarimi F, Liu H, Brown LM, Uchida K, O'Connor OA, Stockwell BR. Imidazole Ketone Erastin Induces Ferroptosis and Slows Tumor Growth in a Mouse Lymphoma Model. *Cell Chem Biol*. 2019;26(5).
3. Zhang Y, Swanda RV, Nie L, Liu X, Wang C, Lee H, Lei G, Mao C, Koppula P, Cheng W, et al. mTORC1 couples cyst(e)ine availability with GPX4 protein synthesis and ferroptosis regulation. *Nature Communications*. 2021;12(1):1589.
4. Lee H, Horbath A, Kondiparthi L, Meena JK, Lei G, Dasgupta S, Liu X, Zhuang L, Koppula P, Li M, et al. Cell cycle arrest induces lipid droplet formation and confers ferroptosis resistance. *Nature Communications*. 2024;15(1):79.
5. Chen X, Huang J, Yu C, Liu J, Gao W, Li J, Song X, Zhou Z, Li C, Xie Y, et al. A noncanonical function of EIF4E limits ALDH1B1 activity and increases susceptibility to ferroptosis. *Nature Communications*. 2022;13(1):6318.
6. Anandhan A, Dodson M, Shakya A, Chen J, Liu P, Wei Y, Tan H, Wang Q, Jiang Z, Yang K, et al. NRF2 controls iron homeostasis and ferroptosis through HERC2 and VAMP8. *Sci Adv*. 2023;9(5):eade9585.

Reviewers' Comments:

Reviewer #2:

Remarks to the Author:

No further comments